# Not All CAMs Are Complete: Completeness as the Key to Faithfulness

**Vincenzo Buono**[*]                                    *vincenzo.buono@hh.se*
**Peyman Sheikholharam Mashhadi**                        *peyman.mashhadi@hh.se*
**Mahmoud Rahat**                                        *mahmoud.rahat@hh.se*
**Prayag Tiwari**                                        *prayag.tiwari@hh.se*
**Stefan Byttner**                                       *stefan.byttner@hh.se*
*Halmstad University, Halmstad, Sweden*

**Reviewed on OpenReview:** *https://openreview.net/forum?id=NeeGBwXNs5*

## Abstract

Although input-gradient techniques have evolved to mitigate the challenges associated with gradients, modern gradient-weighted CAM approaches still rely on vanilla gradients, which are inherently susceptible to the saturation phenomena. Despite recent enhancements that incorporate counterfactual gradient strategies as a mitigating measure, these local explanation techniques still exhibit a lack of sensitivity to their baseline parameter. Our work introduces a general distributional framework for gradient-based CAMs that recovers Integrated Grad-CAM and SmoothGrad-CAM as special cases of a single perturbation distribution, and from which we derive optimal weights minimizing explanation infidelity, an optimality we prove is governed by completeness as both a necessary and sufficient axiom. Consequently, methods that violate completeness, such as SmoothGrad-based variants, are provably suboptimal. Our technique, *Expected Grad-CAM*[1], instantiates this optimum via Expected Gradients and data-aware perturbations, purposefully designed as an enhanced substitute of the foundational Grad-CAM algorithm and any method built therefrom. By revisiting the original formulation as the smoothed expectation of the perturbed integrated gradients, one can concurrently construct more faithful, localized and robust explanations; through fine modulation of the perturbation distribution, it is possible to regulate the explanation complexity by selectively discriminating stable features. Quantitative and qualitative evaluations have been conducted to assess the effectiveness of our method.

## 1 Introduction

Deep neural networks have achieved remarkable performance across a rapidly growing spectrum of application domains. Yet their efficacy is often coupled with *black-box* behavior, lacking transparency and explainability (Samek et al., 2017; Adadi & Berrada, 2018). This opacity has driven the development of *Explainable AI* (xAI) methodologies aimed at understanding the mechanisms driving model decisions (Gilpin et al., 2018). The need for trustworthiness and reliability (Lipton, 2016) has spurred numerous techniques, ranging from gradient-based (Simonyan et al., 2013), perturbation-based (Ribeiro et al., 2016), and contrastive approaches (Abhishek & Kamath, 2022), to assess *a posteriori* (post-hoc) the behavior of opaque models (Samek et al., 2021). Within the branch of *visual explanations*, *saliency* methods aim to discriminate and identify relevant regions in the input space that highly excite the network and strongly influence the network predictions.

As successful state-of-the-art vision tasks' architectures commonly incorporate spatial convolution mechanism, *Class Activation Maps* (CAM) (Zhou et al., 2015) have emerged as a popular and widely adopted technique for generating *saliencies* that leverage the spatial information captured by convolutional layer. CAM(s)

---

[*]Corresponding author.
[1]Code is available at the GitHub repository: https://github.com/espressoshock/pytorch-expected-gradcam.

are computed by inspecting the feature maps and produce per-instance, class-specific attention heat maps that highlight important areas in the original image that drove the classifier. Building on this notion, Gradient-weighted CAM (Grad-CAM) (Selvaraju et al., 2016) and its variants, extend the original formulation by computing the linear weights from the averaged backpropagated gradients with respect to the target class of each feature map. This generalization enables the use and application of the method without any modification or auxiliary training to the model. Historically, naïve vanilla gradients have been cardinal in the development and evolution of *saliency maps* (Simonyan et al., 2013); however input-gradients techniques (*e.g.*, output gradients *w.r.t.*, inputs) quickly evolved to address the gradient saturation problem (Shrikumar et al., 2017; Rakitianskaia & Engelbrecht, 2015; Sundararajan et al., 2017), where the gradients of important features result in small magnitudes due to the model's function flattening in the vicinity of the input, misrepresenting feature importance (Sundararajan et al., 2016). Within the context of gradient visualizations, several *counterfactual*-based works have been proposed to address saturation via feature scaling (Sundararajan et al., 2017), contribution decomposition (Shrikumar et al., 2017), and relevance propagation (Bach et al., 2015). The insensitivity of *baseline-methods* to their reference parameter (Sundararajan & Taly, 2018; Adebayo et al., 2018) has spurred research dedicated to baseline determination (Ancona et al., 2017; Kindermans et al., 2017; Yeh et al., 2019). Since CAM and Grad-CAM, several gradient-based techniques have been proposed to improve localization (Shi et al., 2020; Jiang et al., 2021), multi-instance detection (Chattopadhay et al., 2017), saliency resolution (Qiu et al., 2023; Draelos & Carin, 2020), noise and attribution sparsity (Omeiza et al., 2019), and axiomatic attributions (Fu et al., 2020). Despite this progress, modern gradient-weighted CAM approaches still rely on vanilla gradients, inherently prone to saturation. A recent work, Integrated Grad-CAM (Sattarzadeh et al., 2021), combines Integrated Gradients and Grad-CAM to address this issue. While path integration yields the desirable *completeness* property (attributions summing to the prediction difference), this method retains baseline insensitivity, underestimating contributions that align with its baseline.

We introduce a *general distributional framework* for gradient-based CAM methods that unifies Integrated Grad-CAM, SmoothGrad-CAM, and our proposed Expected Grad-CAM as special cases determined by the choice of perturbation distribution. Building on this framework, we derive *optimal CAM weights* that provably minimize explanation infidelity, but show this optimality requires the underlying attribution method to satisfy a *completeness axiom*. Notably, we prove that SmoothGrad violates this property, explaining its suboptimal faithfulness despite its noise-reduction benefits. Our method, *Expected Grad-CAM*, instantiates this optimal framework by incorporating the well-established *Expected Gradients* (Erion et al., 2019) into the CAM framework (figure 3). Expected Gradients satisfies the completeness axiom required by our optimal weights theorem, while simultaneously resolving the baseline insensitivity problem by sampling baselines from a reference distribution. This overcomes the feature attribution underestimation of vanilla gradients (figure 2) without introducing undesired side effects. Since CAMs are coarse attention maps used for *human-centered* interpretability (Alvarez-Melis & Jaakkola, 2018a), it is crucial that such methods highlight only stable, salient features, a property that our optimal framework guarantees.

Empirically, we demonstrate that Expected Grad-CAM *concurrently* improves four explanation quality *key desiderata* (Hedström et al., 2022; 2023): (i) faithfulness, (ii) robustness, (iii) localization, and (iv) complexity. Our experiments reveal that Expected Grad-CAM significantly outperforms state-of-the-art gradient- and non-gradient-based CAM methods across 19 quality metrics, with consistent results across different open image datasets. Qualitatively, our technique constructs *saliencies* that are sharper and more focused on class-discriminative image regions, as illustrated in Figure 1. Unlike popular gradient-based CAM methods whose saliency maps are often noisy and sparse (Kim et al., 2019), Expected Grad-CAM highlights only features that are systematically utilized across the sample's neighborhood in input space (*i.e.*, *relative input stability* (Agarwal et al., 2022)).

We summarize our contributions as follows: **First**, we introduce a general distributional framework for gradient-based CAM methods, showing that Integrated Grad-CAM and SmoothGrad-CAM emerge naturally as special cases depending on the choice of perturbation distribution $p_{\mathcal{D}}$. **Second**, we derive optimal CAM weights that minimize explanation infidelity under arbitrary perturbations. We prove formally that this optimality holds if and only if the underlying attribution method satisfies the *completeness axiom*, a necessary and sufficient condition satisfied by Integrated Gradients and Expected Gradients, but violated

by SmoothGrad. **Third**, we propose *Expected Grad-CAM*, which instantiates our optimal framework using Expected Gradients with data-aware perturbations. Extensive experiments across 19 quality metrics (Hedström et al., 2022; 2023) demonstrate that Expected Grad-CAM significantly outperforms state-of-the-art gradient- and non-gradient-based CAM methods in faithfulness, robustness, localization, and complexity.

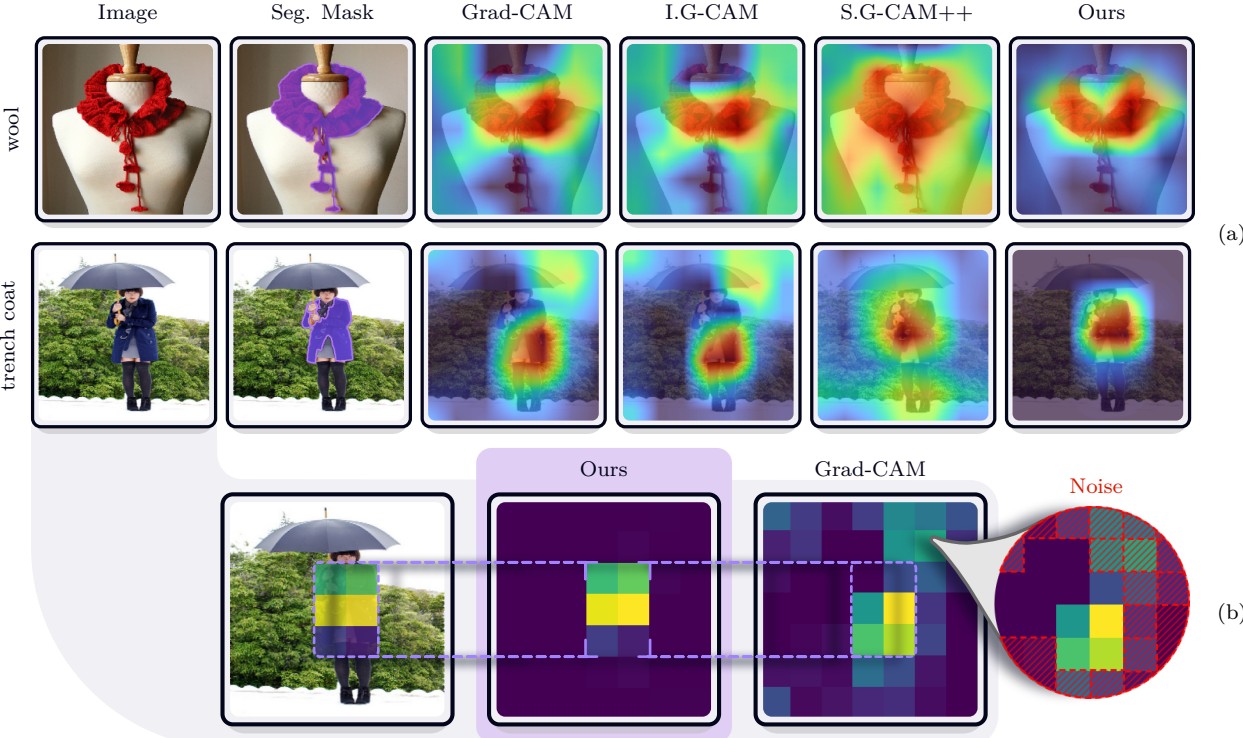

Figure 1: Explanatory functions on VGG-16 across samples from ImageNet-1k (Russakovsky et al., 2014). Our approach produces sharper (less noisy) and higher localized heat maps with lower complexity than existing methods (1a). Figure 1b shows the coarse heat map with respect to our method and baseline Grad-CAM (Selvaraju et al., 2016).

## 2 Related Work

In the following section, we present the scope of this work, introduce the notation, and critically examines prior attribution methods alongside their known shortcomings and limitations.

### 2.1 Preliminaries

Let $f_\theta : \mathcal{X} \to \mathcal{Y}$ be a differentiable neural network mapping inputs $\boldsymbol{x} \in \mathbb{R}^D$ to outputs $\boldsymbol{y} \in \mathbb{R}^C$ for a classification task with $C$ classes, where $\boldsymbol{x} \in \mathcal{X}$ and $\boldsymbol{y} \in \mathcal{Y}$. The parameter vector $\theta \in \Theta \subset \mathbb{R}^W$ is learned through a training process, yielding the trained model $f_\theta$. We refer to this trained model simply as $f$, with predictions $\boldsymbol{y} = f(\boldsymbol{x}; \theta)$. Here, $\theta$ includes the weights and biases of the neural network, and resides in parameter space $\Theta$ under a fixed architecture in function space $f_\theta \in \mathcal{F}$.

**Local Explanations.** To interpret how *specific features* of $\boldsymbol{x}$ influence a particular prediction $f_\theta^c(\boldsymbol{x})$ for a class $c \in \{1, \ldots, C\}$, a *local explanation* method produces an attribution map or *saliency* $\hat{\boldsymbol{e}}$ that highlights the most influential components of $\boldsymbol{x}$. Formally, let

$$\phi_L \colon \mathcal{F} \times \mathcal{X} \times \{1, \ldots, C\} \longrightarrow \mathbb{R}^V \tag{1}$$

be an operator that takes the trained model $f$, an input $\boldsymbol{x}$, and a class index $c$, then produces an *explanation* $\hat{\boldsymbol{e}}$, which we denote as

$$\hat{\boldsymbol{e}} \;=\; \phi_L\big(f,\, \boldsymbol{x},\, c;\, \lambda\big) \quad \in \mathbb{R}^V, \tag{2}$$

where $\lambda$ specifies any additional, explanation-specific hyperparameters. The dimensionality $V$ of $\hat{\boldsymbol{e}}$ may be the same as the input space $D$ or different (e.g., in convolutional architectures, attributions might first be computed in a lower-dimensional feature map and then subsequently upsampled).

In effect, local explanations capture which aspects of $\boldsymbol{x}$ most strongly drive the model toward the logit $f^c(\boldsymbol{x})$. We now survey notable classes of local explanation methods.

**Gradient-based explanations**   This set of techniques encompasses the involvement of the neural network's gradients as a function approximator, translating complex nonlinear models into local linear explanations. These explanations are often encoded as attention heat maps, also known as saliencies. The cornerstone method within this category is *Input-Gradients* (vanilla gradients) (Simonyan et al., 2013), which we define by

$$\phi^{\mathrm{grad}}(f,\, \boldsymbol{x},\, c) \;=\; \nabla_{\boldsymbol{x}} f^c(\boldsymbol{x}), \tag{3}$$

where $\phi^{\mathrm{grad}}$ denotes the class-specific, backpropagated gradient of the class $c$ *w.r.t.*, the input $\boldsymbol{x}$. Notably, while not relevant to our approach, the feature visualization produced by deconvolution (Zeiler & Fergus, 2013) and guided backpropagation (Springenberg et al., 2014) are also tightly linked; the latter, in particular, constrains the gradient flow to non-negative values.

While straightforward, pure gradients often yield high-frequency noise and poorly calibrated attributions in regions of high saturation (Sundararajan et al., 2017). Methods addressing these issues via non-local comparisons or smoothing are discussed below.

**Counterfactual explanations**   As gradients only express local changes, their utilization misrepresents feature importances across saturating ranges (Sundararajan et al., 2017). This class of methods tackles this issue by multiple non-local comparisons against a perturbed baseline by feature re-scaling (Sundararajan et al., 2017), blurring (Fong & Vedaldi, 2017), activation differences (Shrikumar et al., 2017), noise (Smilkov et al., 2017) or in-painting (Alipour et al., 2022). Here, we primarily focus on two kinds of methods that are highly related to our work: Integrated Gradients (Sundararajan et al., 2017) and SmoothGrad (Smilkov et al., 2017).

**Integrated Gradients**   To mitigate saturation artifacts, this method involves the summation of the *interior* gradients along the counterfactual path that interpolates from a baseline input $\boldsymbol{x}'$ to the original input $\boldsymbol{x}$ (Sundararajan et al., 2017; 2016). Concretely, for a class-specific logit $f_c$, It is defined as:

$$\phi^{\mathrm{IG}}(f,\, \boldsymbol{x},\, c;\, \boldsymbol{x}') \;=\; (\boldsymbol{x} - \boldsymbol{x}') \int_{\alpha=0}^{1} \nabla_{\boldsymbol{x}} f^c\big(\boldsymbol{x}' + \alpha(\boldsymbol{x} - \boldsymbol{x}')\big)\, d\alpha, \tag{4}$$

where $\boldsymbol{x}$ is the input sample, $\boldsymbol{x}'$ is a given baseline (representing an "absence" or neutral version of $\boldsymbol{x}$), and $\alpha$ is a scaling parameter that interpolates between the baseline and the input according to a given interpolation function $\gamma$ (Sundararajan et al., 2017). By integrating over this path, the method captures salient gradients even in regions where the model output would otherwise saturate, thereby providing more robust attributions.

**SmoothGrad:** This method addresses saliency noise caused by sharp fluctuations of gradients at small scales, due to rapid local variation in partial derivatives (Smilkov et al., 2017), by denoising using a smoothing Gaussian kernel. It is defined as:

$$\phi^{\mathrm{SG}}(f,\, \boldsymbol{x},\, c;\, n,\, \sigma) \;=\; \frac{1}{n} \sum_{i=1}^{n} \nabla_{\boldsymbol{x}} f^c\Big(\boldsymbol{x} + \mathcal{N}\big(\overline{0},\, \sigma^2 \mathbf{I}\big)\Big), \tag{5}$$

where $\mathcal{N}(\overline{0}, \sigma^2 \mathbf{I})$ denotes Gaussian noise of variance $\sigma^2$, and $n$ is the number of noisy samples averaged.

**Class activation maps** This set of attention methods generates explanations by exploiting the spatial information captured by the convolutional layers. Class activation maps are generated by computing the rectified sum of all the feature map's activations times its weights. Formally, consider a network's target convolutional layer $\ell$ with $K$ feature maps, where each feature map $A^k \in \mathbb{R}^{U \times V}$ has spatial dimensions $U \times V$. For the original CAM (Zhou et al., 2015), the class activation map for class $c$ at spatial location $(u, v)$ is given by

$$M_{u,v}^c = \sum_{k=1}^{K} w_k^c A_{u,v}^k, \tag{6}$$

where $w_k^c \in \mathbb{R}$ are the learned weights of the fully connected layer that combines the global average pooled feature channels to produce the class scores. Grad-CAM (Selvaraju et al., 2016) generalizes this approach by replacing the learned weights $w_k^c$ with gradient-based importance weights $\alpha_k^c$, avoiding architectural constraints. For Grad-CAM, the importance weight $\alpha_k^c$ for feature map $k$ and class $c$ is defined as:

$$\alpha_k^c = \frac{1}{Z} \sum_{u=1}^{U} \sum_{v=1}^{V} \frac{\partial y^c}{\partial A_{u,v}^k}, \tag{7}$$

where $y^c$ is the score (logit) for class $c$ before softmax, and $Z = U \cdot V$ is a normalization constant implementing global average pooling. The final Grad-CAM heatmap is then computed as:

$$L_{\text{Grad-CAM}}^c = \text{ReLU}\left( \sum_{k=1}^{K} \alpha_k^c \cdot A^k \right), \tag{8}$$

where the ReLU ensures only positive influences are captured. This preserves the model's original structure while leveraging the class-specific gradient signal to weight each spatial feature map.

Notably, despite the perturbation of the subregions is performed with distinct different techniques, *DeepLift* (Shrikumar et al., 2017), *input × gradient*, and *SmoothGrad* (Smilkov et al., 2017) they all work under a similar setup of *Integrated Gradients* (Sundararajan et al., 2017) as shown in previous work (Ancona et al., 2017). For instance, *SmoothGrad* can be formulated as the *path integral* where the interpolator function samples a single point from a Gaussian distribution:

$$\phi^{\text{SG}}(f, \boldsymbol{x}, c; n, \sigma) = \frac{1}{n} \sum_{j=1}^{n} \left( \boldsymbol{x} + \boldsymbol{\epsilon}_\sigma^{(j)} \right) \nabla_{\boldsymbol{x}} f^c \left( \boldsymbol{x} + \boldsymbol{\epsilon}_\sigma^{(j)} \right), \quad \text{where } \boldsymbol{\epsilon}_\sigma^{(j)} \sim \mathcal{N}\left( \overline{0}, \sigma^2 \mathbf{I} \right). \tag{9}$$

## 3 Method

In this section, we present our method for deriving optimal Grad-CAM weights that minimize explanation infidelity while addressing the fundamental limitations of gradient saturation and baseline sensitivity. We begin by establishing the mathematical framework, then introduce our optimization-based approach, and finally discuss practical considerations for robust implementation.

**Problem Formulation.** Building on the notation introduced in Section 2.1, let $y^c : (\mathbb{R}^{U \times V})^K \mapsto \mathbb{R}$ be the function that maps a set of $K$ feature maps $(A^1, \ldots, A^K)$ from a specific convolutional layer $\ell$ to the class score $y^c(A^1, \ldots, A^K)$. To analyze the contribution of individual feature maps, we introduce a predictor function $g : \mathbb{R}^K \mapsto \mathbb{R}$ parameterized by the original feature maps $\mathbf{A} = (A^1, \ldots, A^K)$:

$$g(\mathbf{z}'; \mathbf{A}) = y^c(z_1' A^1, z_2' A^2, \ldots, z_K' A^K) \tag{10}$$

where $\mathbf{z}' = (z_1', \ldots, z_K') \in \mathbb{R}^K$ is a vector of scalar multipliers. This formulation allows us to study how scaling individual feature maps affects the model's output.

We seek to explain the behavior of $g$ around the reference point $\mathbf{z}_0 = \mathbf{1} = (1, 1, \ldots, 1) \in \mathbb{R}^K$, which corresponds to using the original, unscaled feature maps. The explanation is characterized by a vector of importance weights $\boldsymbol{\alpha}^c = (\alpha_1^c, \ldots, \alpha_K^c) \in \mathbb{R}^K$, where $\alpha_k^c$ quantifies the importance of the $k$-th feature map $A^k$ for class $c$.

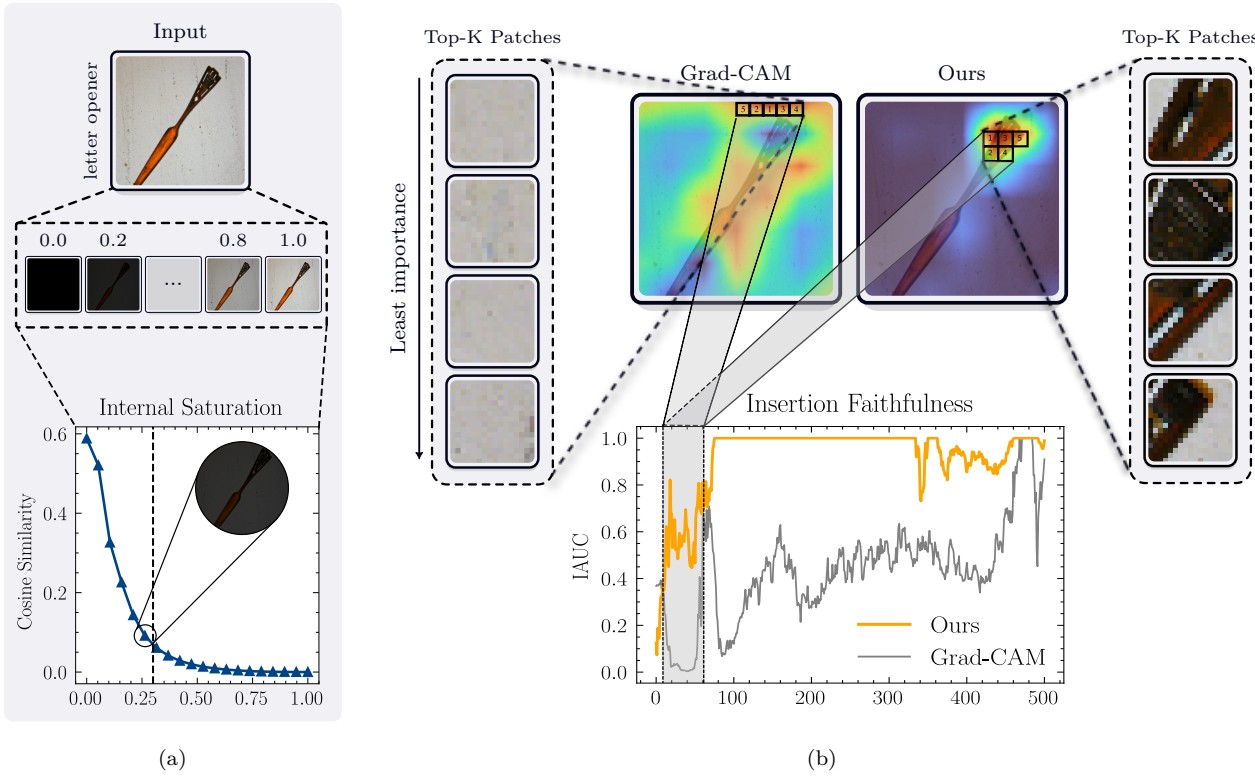

(a)            (b)

Figure 2: Comparison of attribution maps under internal saturation conditions. Figure 2a illustrates the cosine similarity of the target layer's embeddings with respect to the interpolator parameter ($\alpha$) (see Appendix C.1 for more details). Figure 2b displays the attribution maps of various methods under saturation conditions. Internal saturation causes the baseline method to under-represent feature importances across saturating ranges. By extracting the top-4 most important features (Figure 2b), it is evident that the baseline method fails to capture relevant discriminative regions, resulting in low insertion AUCs (Figure 2b) as these regions are not deemed important by the model.

## 3.1 Beyond Local Gradients

The original Grad-CAM formulation (Equation 7) relies on vanilla gradients to derive channel importance weights. However, this approach suffers from a fundamental limitation: gradients capture only *local* changes and thus fail to represent feature importance accurately in regions where $\nabla_{\boldsymbol{x}} f_\theta^c(\boldsymbol{x})$ saturates (Sundararajan et al., 2016).

Previous attempts to address gradient saturation in CAMs through perturbation techniques (Sattarzadeh et al., 2021; Omeiza et al., 2019) have introduced undesirable side effects, including *baseline insensitivity* (Sundararajan & Taly, 2018) and poor robustness to infinitesimal perturbations (Alvarez-Melis & Jaakkola, 2018a; Sundararajan et al., 2017; Ghorbani et al., 2017).

**A Distributional Perspective on Missingness.** Methods predicated on *perturbing* input features have arisen as a principled means to assess the *missingness* of those features in a model's decision-making process (Ancona et al., 2017; Yeh et al., 2019). Under this lens, removing, or zeroing out, certain features of an input effectively emulates a baseline or neutral state, enabling one to gauge how replacing each feature with its baseline counterpart impacts the model output. The salient question, then, is how does one systematically account for, potentially many, such replacements and the ensuing attributions. Remarkably, this viewpoint both recovers classical path-based formulations (such as Integrated Grad-CAM) as special cases and also

elucidates how more flexible distributions, for instance, those used in *Expected Gradients*, can mitigate the baseline insensitivity problem that purely path-based methods often exhibit.

**Perturbation by Replacement.** To move beyond local gradient analysis, we adopt a perturbation-based approach. Let $S \subset \{1, \ldots, D\}$ be a subset of feature indices in the input vector $\boldsymbol{x} \in \mathbb{R}^D$. Following Ancona et al. (2017) and Yeh et al. (2019), we construct an *interpolated* input by selectively replacing features:

$$(\boldsymbol{x}[S \leftarrow \boldsymbol{x}'])_j = x_j \mathbb{I}(j \notin S) + x'_j \mathbb{I}(j \in S), \quad \text{for all } j \in \{1, \ldots, D\} \tag{11}$$

where $\boldsymbol{x}' \in \mathbb{R}^D$ is a reference baseline (e.g., a black image or mean image), and $\mathbb{I}(\cdot)$ is the indicator function.

**General Framework for Attribution Weights.** Building on the perturbation principle, we can express Grad-CAM weights through a general integrated gradients framework. The importance weight for the $k$-th feature map is given by:

$$\alpha^c = \int \phi^{\mathrm{IG}}(g, \boldsymbol{z}_0, c; \boldsymbol{I}, \boldsymbol{A}) \, d\mu_{\boldsymbol{I}} \tag{12}$$

where $\phi^{\mathrm{IG}}$ represents any path attribution method that iteratively identify salient regions by integrating gradients along a specified path (see Appendix A.1.2 for the general framework and derivations of special cases):

$$\phi^{\mathrm{IG}}(g, \boldsymbol{z}_0, c; \boldsymbol{I}, \boldsymbol{A}) = \int_{t=0}^{1} \nabla_{\boldsymbol{z}} g^c(\boldsymbol{z}_0 + \boldsymbol{I}(t-1); \boldsymbol{A}) \, dt \tag{13}$$

This formulation provides a unified view of gradient-based attribution methods. Remarkably, classical path-based formulations emerge naturally as special cases by varying the perturbation distribution $\mu_{\boldsymbol{I}}$.

**Connection to Existing Methods.** Our framework unifies several existing attribution methods as special cases:

- **Integrated Grad-CAM** (Sattarzadeh et al., 2021): Obtained when $\mu_{\boldsymbol{I}}$ is a point mass Dirac delta function at a fixed perturbation. This corresponds to a single baseline and path integration.

- **SmoothGrad-CAM** (Omeiza et al., 2019): Approximated when $\mu_{\boldsymbol{I}}$ follows a Gaussian distribution, but gradients are evaluated only at endpoints rather than integrating along paths. This provides noise-based smoothing without full path integration.

### 3.2 Optimal Attribution via Infidelity Minimization

We now formalize the problem of finding optimal Grad-CAM weights through the lens of explanation infidelity minimization. This framework provides a principled approach to address gradient saturation while maintaining theoretical guarantees.

**Infidelity Metric.** Following Yeh et al. (2019), we measure the quality of an explanation through its *infidelity*: the expected squared error between the attribution's prediction and the actual model behavior under perturbations:

**Definition 3.1** (Explanation Infidelity for Grad-CAM). Consider the Grad-CAM setting where $\boldsymbol{\alpha}^c = (\alpha_1^c, \ldots, \alpha_K^c) \in \mathbb{R}^K$ represents importance weights for $K$ feature maps, and perturbations $\boldsymbol{I} \in \mathbb{R}^K$ are applied to the feature map multipliers. Given the predictor function $g(\boldsymbol{z}; \boldsymbol{A})$ defined in equation 10 and perturbations $\boldsymbol{I}$ with probability measure $\mu_{\boldsymbol{I}}$, the infidelity of the Grad-CAM weights is:

$$\mathrm{INFD}(\boldsymbol{\alpha}^c, g, \boldsymbol{z}_0; \boldsymbol{A}) = \mathbb{E}_{\boldsymbol{I} \sim \mu_{\boldsymbol{I}}} \left[ \left( \boldsymbol{I}^T \boldsymbol{\alpha}^c - (g(\boldsymbol{z}_0; \boldsymbol{A}) - g(\boldsymbol{z}_0 - \boldsymbol{I}; \boldsymbol{A})) \right)^2 \right] \tag{14}$$

where $\boldsymbol{I}^T \boldsymbol{\alpha}^c = \sum_{k=1}^{K} I_k \alpha_k^c$ represents the predicted change in output based on the linear combination of feature map importance weights and perturbations.

This metric quantifies how well the linear approximation $\boldsymbol{I}^T \boldsymbol{\alpha}^c$ predicts the actual change in model output when feature maps are scaled by the perturbed multipliers $\boldsymbol{z}_0 - \boldsymbol{I}$.

**Theorem 3.2** (Optimal Grad-CAM Weights). *Let $\phi$ be any attribution method that takes a predictor function $g : \mathbb{R}^K \to \mathbb{R}$, a reference point $\boldsymbol{z}_0 \in \mathbb{R}^K$, and a perturbation $\boldsymbol{I} \in \mathbb{R}^K$, and returns an attribution vector in $\mathbb{R}^K$ satisfying the completeness axiom:*

$$\boldsymbol{I}^T \cdot \phi(g, \boldsymbol{z}_0, \boldsymbol{I}; \boldsymbol{A}) = g(\boldsymbol{z}_0; \boldsymbol{A}) - g(\boldsymbol{z}_0 - \boldsymbol{I}; \boldsymbol{A})$$

*Suppose the perturbations $\boldsymbol{I} \in \mathbb{R}^K$ drawn from $\mu_{\boldsymbol{I}}$ are such that the second moment matrix $\mathcal{M}_{\boldsymbol{I}} = \int \boldsymbol{I}\boldsymbol{I}^T d\mu_{\boldsymbol{I}}$ is invertible.[2] The optimal Grad-CAM weights that minimize the infidelity equation 14 are:*

$$\boldsymbol{\alpha}^{c*} = \mathcal{M}_{\boldsymbol{I}}^{-1} \left( \int \boldsymbol{I} \langle \boldsymbol{I}, \phi(g, \boldsymbol{z}_0, \boldsymbol{I}; \boldsymbol{A}) \rangle d\mu_{\boldsymbol{I}} \right) \tag{15}$$

*where $\langle \cdot, \cdot \rangle$ denotes the inner product.*

*Proof Sketch.* The infidelity is a quadratic functional in $\boldsymbol{\alpha}^c$. By the completeness axiom, we have $g(\boldsymbol{z}_0; \boldsymbol{A}) - g(\boldsymbol{z}_0 - \boldsymbol{I}; \boldsymbol{A}) = \boldsymbol{I}^T \phi(g, \boldsymbol{z}_0, \boldsymbol{I}; \boldsymbol{A})$. Substituting this into the infidelity expression and taking the derivative with respect to $\boldsymbol{\alpha}^c$ yields the first-order optimality condition, from which equation 15 follows. The full proof is provided in Appendix A.1.6. $\square$

*Remark* 3.3 (Path Attribution in Practice). In practice, we instantiate $\phi$ as a path attribution method (as defined in Equation equation 13) but incorporate a distributional perspective similar to expected gradients to address baseline (in)sensitivity. Rather than using a single fixed baseline as in standard integrated gradients:

$$\phi^{\text{IG}}(g, \boldsymbol{z}_0, \boldsymbol{I}; \boldsymbol{A}) = \int_{t=0}^{1} \nabla_{\boldsymbol{z}} g(\boldsymbol{z}_0 + (t-1)\boldsymbol{I}; \boldsymbol{A}) dt$$

we sample multiple perturbations $\boldsymbol{I}$ from a distribution $\mu_{\boldsymbol{I}}$ and compute the optimal weights via Equation equation 15. This differs from the original infidelity minimization approach, which uses only integrated gradients. The subsequent sections will formalize this approach through expected gradients $\phi^{\text{EG}}$.

## 3.3 Computing Attribution: From Integrated to Expected Gradients

Having established that optimal weights require an attribution method satisfying completeness (Theorem 3.2; see Appendix A.1.12 for the proof that this requirement is both necessary and sufficient), we now examine specific instantiations of $\phi$. We first present standard integrated gradients, then show how it generalizes to expected gradients through baseline distributions.

**Definition 3.4** (Integrated Gradients Attribution). For a predictor function $g : \mathbb{R}^K \to \mathbb{R}$ parameterized by feature maps $\boldsymbol{A}$, the integrated gradients attribution method $\phi^{\text{IG}}$ is defined as:

$$\phi^{\text{IG}}(g, \boldsymbol{z}_0, \boldsymbol{I}; \boldsymbol{A}) = \int_{t=0}^{1} \nabla_{\boldsymbol{z}} g(\boldsymbol{z}_0 + (t-1)\boldsymbol{I}; \boldsymbol{A}) dt \tag{16}$$

where $\boldsymbol{z}_0 \in \mathbb{R}^K$ is the reference point and $\boldsymbol{I} \in \mathbb{R}^K$ is the perturbation vector. This integrates gradients along the straight-line path from baseline $\boldsymbol{z}_0 - \boldsymbol{I}$ to $\boldsymbol{z}_0$.

**Definition 3.5** (Expected Gradients Attribution). For a centered baseline distribution $\mathcal{D}$ over multiplier vectors in $\mathbb{R}^K$ satisfying $\mathbb{E}_{\boldsymbol{z}' \sim \mathcal{D}}[\boldsymbol{z}'] = \boldsymbol{0}$,[3] the expected gradients attribution method $\phi^{\text{EG}}$ is defined as:

$$\phi^{\text{EG}}(g, \boldsymbol{z}_0, \boldsymbol{I}; \boldsymbol{A}, \mathcal{D}) = \int_{\boldsymbol{z}' \sim \mathcal{D}} \left[ \int_{t=0}^{1} \nabla_{\boldsymbol{z}} g(\boldsymbol{z}' + t(\boldsymbol{z}_0 - \boldsymbol{I} - \boldsymbol{z}'); \boldsymbol{A}) dt \right] p_{\mathcal{D}}(\boldsymbol{z}') d\boldsymbol{z}' \tag{17}$$

where $g : \mathbb{R}^K \to \mathbb{R}$ is the predictor function parameterized by $\boldsymbol{A}$, $\boldsymbol{z}_0 \in \mathbb{R}^K$ is the reference point, and $\boldsymbol{I} \in \mathbb{R}^K$ is the perturbation vector.

---

[2]Invertibility holds when $\text{supp}(\mu_{\boldsymbol{I}})$ spans $\mathbb{R}^K$; for data-aware perturbations (Definition 3.8) this is ensured by diverse sampling. See Appendix A.1.5 for rank-deficient cases and numerical stability.

[3]In practice, $\mathcal{D} = \mathcal{N}(\boldsymbol{0}, \sigma^2 \boldsymbol{I}_K)$. Note that $\mathcal{D}$ (baseline sampling *within* $\phi^{\text{EG}}$) is distinct from $\mu_{\boldsymbol{I}}$ (perturbations for infidelity minimization in equation 14).

*Remark* 3.6 (Relationship Between Attribution Methods). These attribution methods are related as follows:

- $\phi^{\mathbf{IG}}$ **as special case**: When $\mathcal{D} = \delta_{\boldsymbol{z}_0-\boldsymbol{I}}$ (Dirac delta), we have $\phi^{\mathrm{EG}}(g, \boldsymbol{z}_0, \boldsymbol{I}; \boldsymbol{A}, \delta_{\boldsymbol{z}_0-\boldsymbol{I}}) = \phi^{\mathrm{IG}}(g, \boldsymbol{z}_0, \boldsymbol{I}; \boldsymbol{A})$

- **Robustness**: $\phi^{\mathrm{EG}}$ averages over multiple baselines, reducing sensitivity to baseline choice

- **Completeness**: Both satisfy the completeness axiom required by Theorem 3.2

*Remark* 3.7 (Completeness Verification). Both attribution methods satisfy the completeness axiom as required by Theorem 3.2.

- $\phi^{\mathrm{IG}}$: We have $\boldsymbol{I}^T \cdot \phi^{\mathrm{IG}}(g, \boldsymbol{z}_0, \boldsymbol{I}; \boldsymbol{A}) = g(\boldsymbol{z}_0; \boldsymbol{A}) - g(\boldsymbol{z}_0 - \boldsymbol{I}; \boldsymbol{A})$

- $\phi^{\mathrm{EG}}$: When $\mathbb{E}_{\boldsymbol{z}'\sim\mathcal{D}}[\boldsymbol{z}'] = \mathbf{0}$,[4] we have $\boldsymbol{I}^T \cdot \phi^{\mathrm{EG}}(g, \boldsymbol{z}_0, \boldsymbol{I}; \boldsymbol{A}, \mathcal{D}) = g(\boldsymbol{z}_0; \boldsymbol{A}) - g(\boldsymbol{z}_0 - \boldsymbol{I}; \boldsymbol{A})$

Applying Theorem 3.2 to specific attribution methods yields closed-form optimal weights for both $\phi^{\mathrm{IG}}$ and $\phi^{\mathrm{EG}}$ (see Appendix A.1.9).

### 3.4 Robust Perturbations via Data Distribution

Beyond fidelity, a quality explanation method must satisfy multiple desirable properties (Hedström et al., 2022). The choice of perturbation distribution $\mu_{\boldsymbol{I}}$ is crucial for:

1. **Preserving sensitivity**: Ensuring the explanation responds appropriately to meaningful input changes (Sundararajan et al., 2017)

2. **Maintaining stability**: Guaranteeing consistent behavior across input, output, and intermediate representations (Agarwal et al., 2022)

3. **Ensuring robustness**: Preventing adversarial manipulation through infinitesimal perturbations (Slack et al., 2019)

Constant baselines fail to account for data distribution characteristics, leading to high sensitivity to noise (Yeh et al., 2019). We address this by constructing perturbations that reflect the underlying data distribution:

**Definition 3.8** (Data-Aware Perturbations). Let $\mathcal{X}$ be the data distribution. We define robust perturbations through Monte Carlo sampling:

$$\boldsymbol{I} = \boldsymbol{z}_0 - \mathbb{E}_{\boldsymbol{x}'\sim\mathcal{X}, \alpha\sim U(0,1)}[\alpha \cdot h(\boldsymbol{x}')] \tag{18}$$

where $h : \mathcal{X} \to \mathbb{R}^K$ is the feature extraction function that maps data samples to feature map multipliers. Specifically, for an input $\boldsymbol{x}' \in \mathcal{X}$, we define

$$h(\boldsymbol{x}') = \big(\mathrm{GAP}(A^1(\boldsymbol{x}')), \ldots, \mathrm{GAP}(A^K(\boldsymbol{x}'))\big) \tag{19}$$

where $A^k(\boldsymbol{x}')$ denotes the $k$-th feature map at layer $\ell$ when $\boldsymbol{x}'$ is passed through the network, and $\mathrm{GAP}(\cdot)$ denotes global average pooling over spatial dimensions.

This approach combines the smoothing benefits of Gaussian noise (Smilkov et al., 2017; Omeiza et al., 2019) with improved robustness by ensuring perturbations remain within the data manifold, reducing out-of-distribution (OOD) artifacts.

---

[4]Centering ensures cross-terms cancel in the path integral; see Appendix A.1.3.

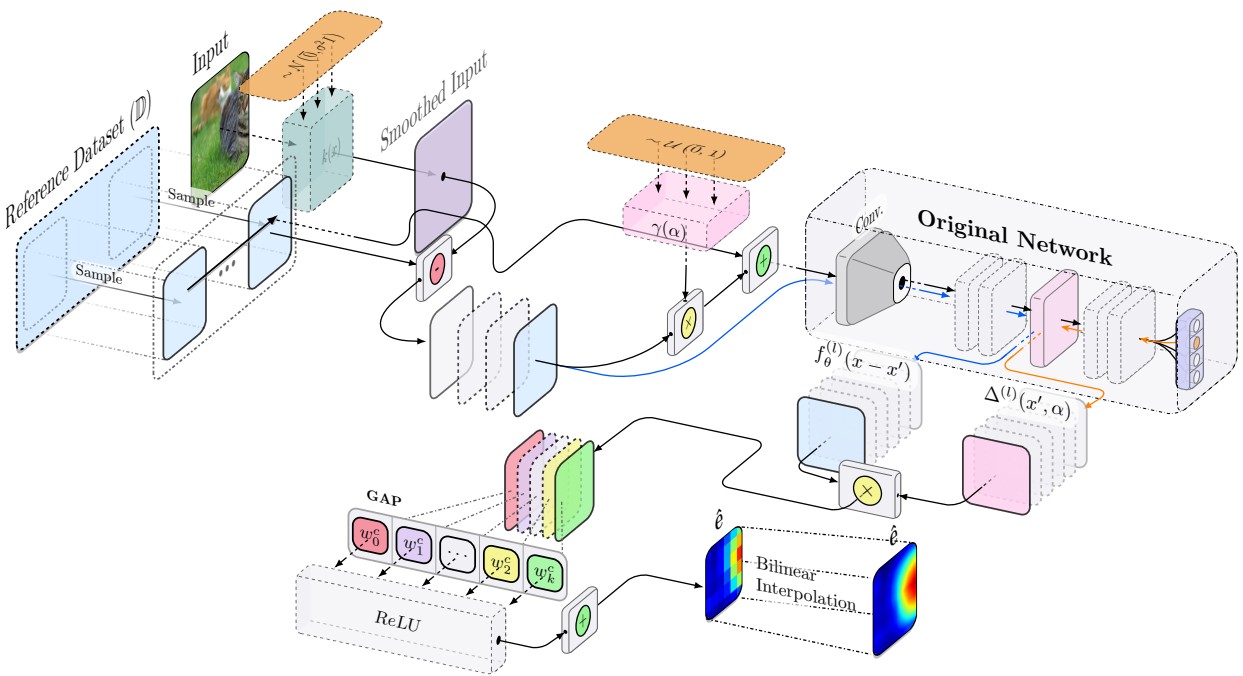

Figure 3: Overview of the proposed Expected Grad-CAM method. Given an input image, a target class, and a reference distribution to sample from, the class-discriminative explanation $\hat{e}$ is computed through input kernel smoothing and difference-from-reference comparisons.

## 3.5 Expected Grad-CAM: Smooth, Noise-Resistant Explanations

We now formalize Expected Grad-CAM, which unifies the theoretical components developed above: expected gradients attribution, data-aware perturbations, and infidelity-minimizing optimal weights, into a principled explanation method.

**Definition 3.9** (Expected Grad-CAM)**.** Let $f_\theta : \mathbb{R}^{H \times W \times C} \to \mathbb{R}^K$ be a CNN classifier, $\ell$ a convolutional layer with feature maps $\boldsymbol{A} = (A^1, \ldots, A^K)$, and $c$ the target class. Expected Grad-CAM computes importance weights $\boldsymbol{\alpha}_{\text{EG}}^{c*} \in \mathbb{R}^K$ as:

$$\boldsymbol{\alpha}_{\text{EG}}^{c*} = \arg \min_{\boldsymbol{\alpha}^c \in \mathbb{R}^K} \mathbb{E}_{\boldsymbol{I} \sim \mu_{\boldsymbol{I}}^{\mathcal{X}}} \left[ \left( \boldsymbol{I}^T \boldsymbol{\alpha}^c - (g(\boldsymbol{z}_0; \boldsymbol{A}) - g(\boldsymbol{z}_0 - \boldsymbol{I}; \boldsymbol{A})) \right)^2 \right] \tag{20}$$

where $g(\boldsymbol{z}'; \boldsymbol{A}) = y^c(z_1' A^1, \ldots, z_K' A^K)$ is the predictor function, $\mu_{\boldsymbol{I}}^{\mathcal{X}}$ is the data-aware perturbation distribution induced by $\mathcal{X}$, and the expectation is taken over perturbations $\boldsymbol{I} = \boldsymbol{z}_0 - \mathbb{E}_{\boldsymbol{x}' \sim \mathcal{X}, \alpha \sim U(0,1)}[\alpha \cdot h(\boldsymbol{x}')]$.

**Theorem 3.10** (Closed-Form Solution for Expected Grad-CAM)**.** *Under the conditions of Theorem 3.2, and using expected gradients attribution $\phi^{EG}$ with baseline distribution $\mathcal{D}$ satisfying $\mathbb{E}_{\boldsymbol{z}' \sim \mathcal{D}}[\boldsymbol{z}'] = \boldsymbol{0}$, the Expected Grad-CAM weights have the closed-form solution:*

$$\boldsymbol{\alpha}_{EG}^{c*} = \mathcal{M}_{\boldsymbol{I}}^{-1} \left( \mathbb{E}_{\boldsymbol{I} \sim \mu_{\boldsymbol{I}}^{\mathcal{X}}} \left[ \boldsymbol{I} \langle \boldsymbol{I}, \phi^{EG}(g, \boldsymbol{z}_0, \boldsymbol{I}; \boldsymbol{A}, \mathcal{D}) \rangle \right] \right) \tag{21}$$

*where $\mathcal{M}_{\boldsymbol{I}} = \mathbb{E}_{\boldsymbol{I} \sim \mu_{\boldsymbol{I}}^{\mathcal{X}}}[\boldsymbol{I} \boldsymbol{I}^T]$ is the second moment matrix of the data-aware perturbations.*

*Proof Sketch.* Direct application of Theorem 3.2 with $\phi = \phi^{\text{EG}}$ and $\mu_{\boldsymbol{I}} = \mu_{\boldsymbol{I}}^{\mathcal{X}}$. The completeness of $\phi^{\text{EG}}$ (Theorem A.8) ensures the solution minimizes infidelity. $\square$

**Definition 3.11** (Expected Grad-CAM Heatmap)**.** The Expected Grad-CAM heatmap for class $c$ is the spatial activation map:

$$L_{\text{EG-CAM}}^c = \text{ReLU} \left( \sum_{k=1}^K \alpha_{\text{EG},k}^{c*} \cdot A^k \right) \in \mathbb{R}^{U \times V} \tag{22}$$

where $\alpha_{\mathrm{EG},k}^{c*}$ is the $k$-th component of $\boldsymbol{\alpha}_{\mathrm{EG}}^{c*}$ and $A^k \in \mathbb{R}^{U \times V}$ is the $k$-th feature map.

*Remark* 3.12 (Connection to SmoothGrad). The optimal weights formula equation 21 reveals a connection to generalized SmoothGrad (Yeh et al., 2019). The kernel $\boldsymbol{II}^T$ in the second moment matrix $\mathcal{M}_{\boldsymbol{I}}$ acts as a data-adaptive smoothing kernel. Unlike standard SmoothGrad which uses isotropic Gaussian noise, our approach adapts the smoothing to the data distribution through $\mu_{\boldsymbol{I}}^{\mathcal{X}}$, providing more principled regularization.

**Proposition 3.13** (Properties of Expected Grad-CAM). *Expected Grad-CAM satisfies the following properties:*

1. **Optimality**: *$\boldsymbol{\alpha}_{EG}^{c*}$ minimizes the infidelity functional over all weight vectors*

2. **First-Order Condition**: *The optimal weights satisfy*

$$\mathcal{M}_{\boldsymbol{I}}\boldsymbol{\alpha}_{EG}^{c*} = \mathbb{E}_{\boldsymbol{I} \sim \mu_{\boldsymbol{I}}^{\mathcal{X}}} \left[ \boldsymbol{I}(g(\boldsymbol{z}_0; \boldsymbol{A}) - g(\boldsymbol{z}_0 - \boldsymbol{I}; \boldsymbol{A})) \right]$$

3. **Data Coherence**: *The perturbation distribution $\mu_{\boldsymbol{I}}^{\mathcal{X}}$ ensures explanations remain within the data manifold*

4. **Baseline Robustness**: *The method is robust to baseline selection through the expectation over $\mathcal{D}$ in $\phi^{EG}$*

**Computational Aspects.** While the theoretical formulation provides exact optimal weights, practical implementation requires Monte Carlo approximation of the expectations in equation 21. The computational complexity is $O(MKN)$ where $M$ is the number of perturbation samples, $K$ is the number of feature maps, and $N$ is the number of baseline samples for expected gradients. Despite this additional computation, Expected Grad-CAM achieves competitive running times in practice, remaining faster than Score-CAM and Ablation-CAM while providing superior explanation quality (Table 10). We defer implementation details to the supplementary material.

## 4 Experiments

In line with previous works (Jiang et al., 2021; Wang et al., 2019), we evaluate our proposed method quantitatively and qualitatively.

**Datasets.** We consider the *ILSVRC2012* (Russakovsky et al., 2014), *CIFAR10* (Ho-Phuoc, 2018) and *COCO* (Lin et al., 2014) with images of size $224 \times 224$. The first two datasets have been used for the quantitative metrics, while the latter only for the localization evaluations, where the segmentation masks of each sample have been employed.

**Models.** Each metric is evaluated across popular feed-forward CNN architectures. In line with prior literature, we restricted our analysis to *VGG16* (Simonyan & Zisserman, 2014), *ResNet50* (He et al., 2015) and *AlexNet* (Krizhevsky et al., 2012). In all cases, the default pre-trained *PyTorch* torchvision implementation has been adopted.

**Metrics.** In contrast to prior works, we comprehensively evaluate our technique across an extensive set of traditional and modern metrics. We provide a full characterization of the behavior of our method by evaluating not just faithfulness, but rather all the different explanatory qualities across recent explanation quality groupings (Hedström et al., 2022) *i.e.*, (i) Faithfulness, (ii) Robustness, (iii) Complexity, and (iv) Localization. In Table 4 are presented all the evaluated metrics categorized by quality groupings, while the extended quantitative results are available in Appendix C.

**Baselines.** We compare our proposed technique against recent and relevant methods including Grad-CAM (Selvaraju et al., 2016), Grad-CAM++ (Chattopadhay et al., 2017), Smooth Grad-CAM++ (Omeiza et al., 2019), Integrated Grad-CAM (Sattarzadeh et al., 2021), HiRes-CAM (Draelos & Carin, 2020), XGrad-CAM (Fu et al., 2020), LayerCAM (Jiang et al., 2021), Score-CAM (Wang et al., 2019), and Ablation-CAM (Desai & Ramaswamy, 2020).

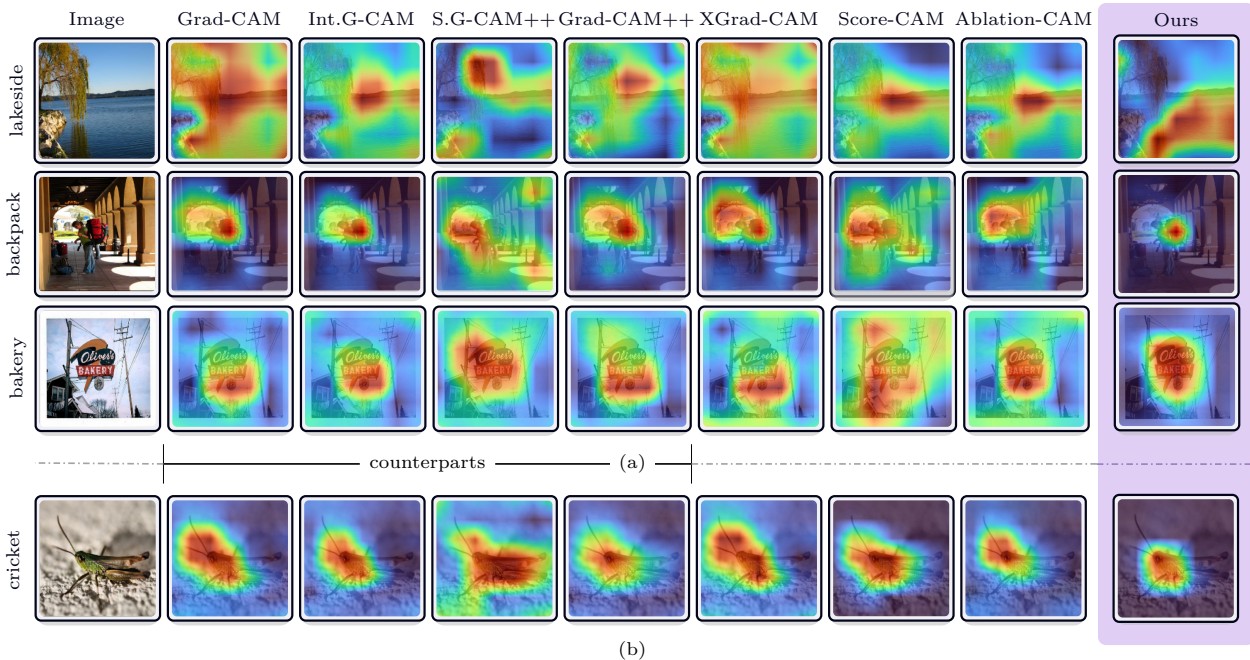

Figure 4: Comparison of attribution maps under normal (4b) and internal saturation (4a) conditions. *Expected Grad-CAM* produces sharper, more localized, and more stable explanations than its direct counterparts: *G-CAM* (Selvaraju et al., 2016), *G-CAM++* (Chattopadhay et al., 2017), *S.G-CAM++* (Omeiza et al., 2019), and *Int. G-CAM* (Sattarzadeh et al., 2021), while remaining competitive with non-gradient-based and more complex gradient-augmented methods. See Appendix D for full comparisons.

**Qualitative evaluations**   In Figure 4 we present an excerpt of the explanations generated during the computation of the quantitative evaluations on the *ILSVRC2012* validation set. By inspecting the attribution sparsity and localization characteristics of each explanation, our method (*Expected Grad-CAM*), generally produces saliencies that are more localized and focused on the attuned human-centric understanding of the composition of the attributes of the labels. An explanation designed for human fruition *i.e.*, aimed at building the model's trustworthiness should be encoded as such to not *disrupt trust*; this implies that an *human-interpretable* explanation should be restricted to the most important pertinent and stable features: it should contains the least number of stable features which do maximally fulfill the notion of fidelity (Figure 5, 14). In Figure 4a it is observed qualitatively that every other compared attribution method breaks such condition: given the labels *lakeside* and *backpack* the explanations highlights areas which are not pertinent with label-related attributes *i.e.*, the sky and portions of the tree (Subfig:4a) and parts of the background (Subfig:4b) respectively.

**Quantitative evaluations**   Following, we assess the validity of our claims quantitatively across various desirable explanatory qualities. The extended quantitative results are available in Section C.

**Faithfulness.**   Examining traditional *faithfulness* metrics (Insertion and Deletion AUCs) across popular benchmarking networks on a large chunk of *ILSVRC2012*, showed promising results (Table 5). Our method *Expected Grad-CAM*, achieved best or second-best scores across its gradient and non-gradient-based counterparts as well as more advanced variations of CAM, which do not solely rely on a gradient augmentation, in both the *insertion* and *deletion* aspects. Towards a more comprehensive comparison, we then verified our technique against more recent metrics. Unsurprisingly, *IROF* (Table 1) showed agreement with traditional metrics as they fundamentally assess similar explanatory qualities. Our technique scored higher than others on the *Sufficiency* (Dasgupta et al., 2022) metric, due to greater *stability* and *robustness* (Table 1). Finally, we tested *Expected Grad-CAM*'s performances in terms of *infidelity* (Yeh et al., 2019), which expectedly

Table 1: *Faithfulness*, *Robustness* and *Complexity Metrics*. Values evaluated on ILSVRC2012(Russakovsky et al., 2014) on VGG16 (Simonyan & Zisserman, 2014). Extended results are available in Appendix C.

| | Faithfulness | | | Robustness | | | Complexity | |
|---|---|---|---|---|---|---|---|---|
| **Method** | ↑ **IROF** | ↑ **Suff.** | ↓ **Inf.** | ↓ **L. Est.** | ↓ **M. Sens.** | ↓ **A. Sens.** | ↓ **CP.** | ↑ **SP.** |
| Grad-CAM | 55.36 | 1.91 | 8.12 | 0.38 | 0.27 | 0.20 | 10.56 | 0.38 |
| Grad-CAM++ | 56.93 | 1.87 | 7.98 | 0.32 | **0.192** | 0.15 | 10.53 | 0.40 |
| Sm. Grad-CAM++ | 56.38 | 1.89 | 7.50 | 0.51 | 0.51 | 0.27 | 10.60 | 0.35 |
| Int. Grad-CAM | 57.36 | 1.83 | 8.92 | 1.05 | 1.00 | 1.00 | 10.59 | 0.36 |
| HiRes-CAM | 57.49 | 1.74 | 5.73 | 0.99 | 1.00 | 1.00 | 10.54 | 0.40 |
| XGrad-CAM | 57.32 | 1.98 | 7.88 | 0.37 | 0.23 | 0.18 | 10.56 | 0.38 |
| LayerCAM | 58.15 | 1.74 | 7.22 | 0.31 | 0.19 | **0.14** | 10.56 | 0.38 |
| Score-CAM | 55.37 | 1.91 | 7.39 | 0.68 | 0.65 | 0.53 | 10.56 | 0.38 |
| Ablation-CAM | 57.36 | 1.83 | 7.28 | 1.05 | 1.00 | 1.00 | 10.59 | 0.36 |
| Expected Grad-CAM | **62.39** | **2.10** | **4.99** | **0.24** | 0.194 | 0.15 | **10.43** | **0.47** |

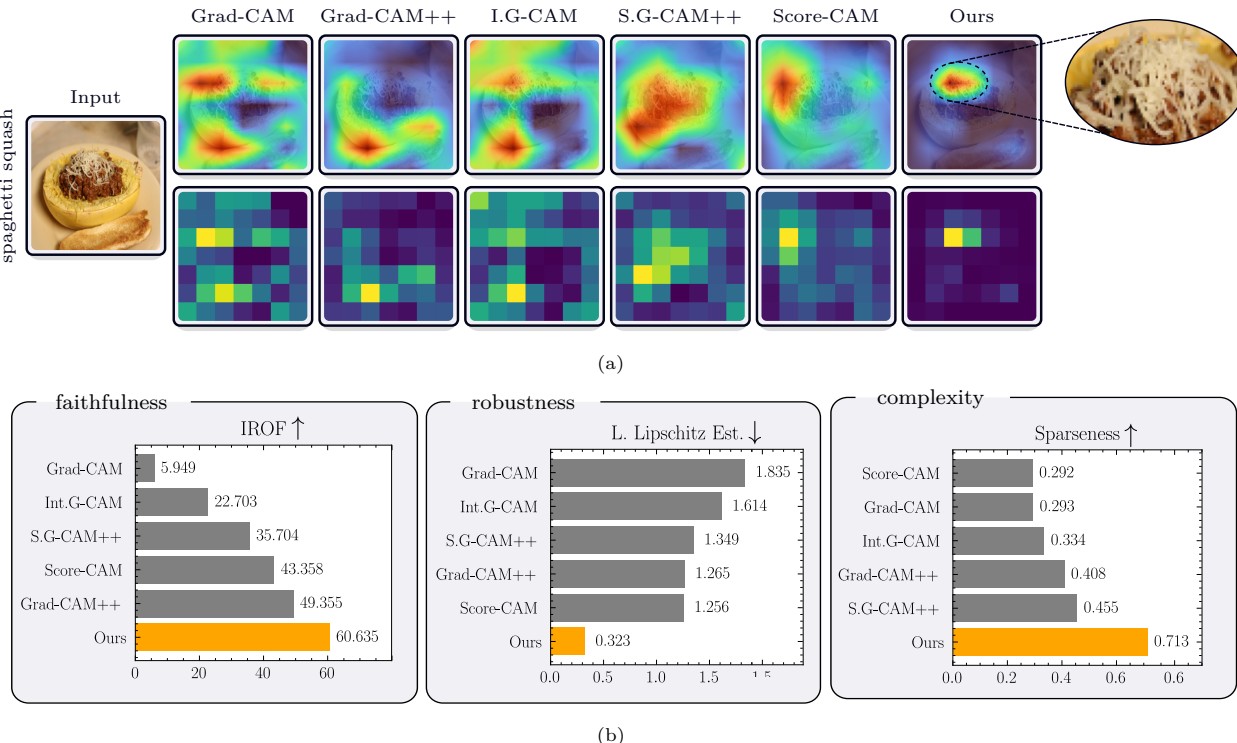

(a)

(b)

Figure 5: Comparison of saliencies generated by different gradient- and non-gradient-based methods. 5a shows the superimposed (top row) and raw coarse saliencies (bottom row) generated by each method. Our method consistently produces more focused and sharper saliencies compared to both gradient-based and non-gradient-based methods (e.g., Score-CAM). 5b demonstrates that our approach concurrently improves key xAI properties: (i) faithfulness, (ii) robustness, and (iii) complexity, significantly outperforming even non-gradient-based methods.

showed the lowest (best) results.

**Stability.**  In Table 8 are presented the results *w.r.t.*, to the relative- input and out stability metrics. our method showed the lowest score overall (*highest stability*), while achieving best or second-best robustness scores (Table 1).

## 5 Conclusion and Broader Impact

In this paper, we advanced current CAM's gradient faithfulness by proposing *Expected Grad-CAM* which simultaneously addresses the saturation and sensitivity phenomena, without introducing undesirable side effects. Revisiting the original formulation as the smoothed expectation of the perturbed integrated gradients, one can concurrently construct more faithful, localized, and robust explanations that minimize infidelity. Despite qualitative assessment being highly subjective, quantitative evaluations are also teeming with indeterminate, ambiguous results that span further than the rank-order issues. While faithfulness is a universally desirable underlying explanatory quality, individual metrics, which do assess such property, only define a distinct notion of such a multifaceted trait, potentially delineating unwanted aspects. While careful modulation of the smoothing functional allows for fine-grained control of the complexity characteristic of the explanation, where, through sensitivity reduction, produces more human-interpretable saliencies; it contrastingly influences the current notions of faithfulness. Perhaps, further adaption of existing metrics may be necessary to embody human-interpretability; nevertheless, existing qualitative and quantitative assessments proved the superiority of our approach.

**Broader impact.** This paper highlights the value and effectiveness of Expected Grad-CAM in comparison to current state-of-the-art approaches across a comprehensive set of modern evaluation metrics. We demonstrated that our technique satisfies many desirable xAI properties by producing explanations that are highly concentrated on the least number of stable robust, features. Our experiments revealed that many state-of-the-art approaches underperform on modern metrics. Ultimately, as our technique is intended to replace the original formulation of Grad-CAM, we hope new and existing approaches will build on it. While Expected Grad-CAM incurs an $O(MKN)$ overhead over vanilla Grad-CAM's single pass, this is the principled cost of completeness and provable infidelity minimization: guarantees that transfer reliably across models, datasets, and deployment contexts.

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

# A    Appendix

This appendix first presents theoretical considerations including detailed proofs, lemmas, and mathematical foundations of our approach. Following the theoretical treatment, we provide extended experimental results and implementation details. In Table 4 are listed the evaluated abbreviated metric names followed by their source, categorized by the underlying explanatory quality they seek to assess Hedström et al. (2022). Where applicable, IG-CAM and SG-CAM abbreviation have been used in place of Integrated Grad-CAM Sattarzadeh et al. (2021) and Smooth Grad-CAM++ Omeiza et al. (2019) respectively. All results have been computed on a single A100-SXM4 80GB platform and a Xeon Gold 5317 with CUDA v12.0.

## A.1    Theoretical Foundations

The following subsections provide detailed proofs, extensions, and additional discussions.

### A.1.1    Notation Conventions

To ensure clarity, we summarize the key notation used throughout this work.

**Attribution Methods.**    We use $\phi$ with subscripts/superscripts to denote explanation methods:

- $\phi_L$: General local explanation operator mapping $(f, \boldsymbol{x}, c) \mapsto \hat{\boldsymbol{e}} \in \mathbb{R}^V$ (Section 2.1)

- $\phi^{\mathrm{IG}}$: Integrated Gradients, satisfies completeness (Definition 3.4)

- $\phi^{\mathrm{EG}}$: Expected Gradients, satisfies completeness when $\mathcal{D}$ is centered (Definition 3.5)

**Distributions.**    Two distinct probability measures govern our framework:

- $\mu_{\boldsymbol{I}}$ (or $\mu_{\boldsymbol{I}}^{\mathcal{X}}$): **Perturbation distribution** over perturbation vectors $\boldsymbol{I} \in \mathbb{R}^K$. This distribution governs the outer expectation in the infidelity functional equation 14. The data-aware variant $\mu_{\boldsymbol{I}}^{\mathcal{X}}$ is induced by the data distribution $\mathcal{X}$ (Definition 3.8).

- $\mathcal{D}$: **Baseline distribution** for expected gradients, a probability measure over $\mathbb{R}^K$ satisfying the centering condition $\mathbb{E}_{\boldsymbol{z}' \sim \mathcal{D}}[\boldsymbol{z}'] = \boldsymbol{0}$. This distribution governs the inner expectation within $\phi^{\mathrm{EG}}$ (Definition 3.5).

These distributions play complementary roles: $\mathcal{D}$ determines how baselines are sampled within the expected gradients computation, while $\mu_{\boldsymbol{I}}$ determines how perturbations are sampled for infidelity minimization. In practice, $\mathcal{D} = \mathcal{N}(\boldsymbol{0}, \sigma^2 \boldsymbol{I}_K)$ and $\mu_{\boldsymbol{I}}^{\mathcal{X}}$ is derived from the training data.

### A.1.2    Path Attribution Methods: General Framework

Path attribution methods form a principled class of explanation techniques that assign importance scores by integrating gradients along a path from a baseline to the input. As discussed by previous work (Sundararajan et al., 2017; Ancona et al., 2017), various perturbation-based explanation schemes can be reformulated under a unified geometric path integral framework.

**Definition A.1** (Valid Path)**.** A *valid interpolation path* is a function $\gamma : [0, 1] \to \mathbb{R}^K$ satisfying:

1. $\gamma(0) = $ baseline (starting point)

2. $\gamma(1) = $ input (ending point)

3. $\gamma$ is differentiable on $[0, 1]$

**Definition A.2** (Path Attribution Method). Given a predictor function $g : \mathbb{R}^K \to \mathbb{R}$ and an interpolation path $\gamma : [0, 1] \to \mathbb{R}^K$, the *path attribution* for the $k$-th component is defined as:

$$\phi_k^\gamma(g, \gamma) = \int_{t=0}^1 \frac{\partial g(\gamma(t))}{\partial \gamma_k(t)} \frac{\partial \gamma_k(t)}{\partial t} \, dt \tag{23}$$

Equivalently, using the Fréchet derivative formulation:

$$\phi_k^\gamma(g, \gamma) = \int_{t=0}^1 Dg(\gamma(t)) \cdot \boldsymbol{e}_k \, dt \tag{24}$$

where $\boldsymbol{e}_k$ denotes the $k$-th standard basis vector $((\boldsymbol{e}_k)_j = \mathbb{I}(j = k))$ and $Dg$ is the Fréchet derivative.

**Definition A.3** (Linear Path). The *linear interpolation path* from baseline $\boldsymbol{b}$ to input $\boldsymbol{x}$ is:

$$\gamma^{\text{lin}}(t) = \boldsymbol{b} + t \cdot (\boldsymbol{x} - \boldsymbol{b}), \quad t \in [0, 1] \tag{25}$$

This satisfies $\gamma^{\text{lin}}(0) = \boldsymbol{b}$ and $\gamma^{\text{lin}}(1) = \boldsymbol{x}$.

**Lemma A.4** (Linear Path Derivative). *For the linear path $\gamma^{lin}(t) = \boldsymbol{b} + t \cdot (\boldsymbol{x} - \boldsymbol{b})$, the derivative is constant:*

$$\frac{d\gamma^{lin}}{dt}(t) = \boldsymbol{x} - \boldsymbol{b} \tag{26}$$

*Proof.* Direct differentiation of each component: $\frac{d}{dt}[b_k + t(x_k - b_k)] = x_k - b_k$. $\qquad\square$

**Theorem A.5** (Path Attribution Completeness). *Let $g : \mathbb{R}^K \to \mathbb{R}$ be a differentiable function, $\gamma : [0, 1] \to \mathbb{R}^K$ a valid path from baseline $\boldsymbol{b}$ to input $\boldsymbol{x}$, and suppose:*

1. *$g$ is differentiable at each point $\gamma(t)$ for $t \in [0, 1]$*

2. *The path derivative is constant: $\frac{d\gamma}{dt}(t) = \boldsymbol{x} - \boldsymbol{b}$ for all $t$*

3. *For each $\boldsymbol{v} \in \mathbb{R}^K$, the function $t \mapsto Dg(\gamma(t)) \cdot \boldsymbol{v}$ is continuous on $[0, 1]$*

*Then the path attribution satisfies the* completeness axiom*:*

$$\sum_{k=1}^K (x_k - b_k) \cdot \phi_k^\gamma(g, \gamma) = g(\boldsymbol{x}) - g(\boldsymbol{b}) \tag{27}$$

*Proof.* Define $h(t) = g(\gamma(t))$. By the chain rule, for any $t \in [0, 1]$:

$$\frac{dh}{dt}(t) = Dg(\gamma(t)) \cdot \frac{d\gamma}{dt}(t) = Dg(\gamma(t)) \cdot (\boldsymbol{x} - \boldsymbol{b}), \tag{28}$$

where the second equality uses the constant path derivative assumption. Since $t \mapsto Dg(\gamma(t)) \cdot (\boldsymbol{x} - \boldsymbol{b})$ is continuous by hypothesis, the fundamental theorem of calculus yields

$$g(\boldsymbol{x}) - g(\boldsymbol{b}) = h(1) - h(0) = \int_{t=0}^1 \frac{dh}{dt}(t) \, dt = \int_{t=0}^1 Dg(\gamma(t)) \cdot (\boldsymbol{x} - \boldsymbol{b}) \, dt. \tag{29}$$

The Fréchet derivative $Dg(\gamma(t))$ is a continuous linear functional; hence, expanding $(\boldsymbol{x} - \boldsymbol{b}) = \sum_{k=1}^K (x_k - b_k) \boldsymbol{e}_k$ and applying linearity gives

$$Dg(\gamma(t)) \cdot (\boldsymbol{x} - \boldsymbol{b}) = \sum_{k=1}^K (x_k - b_k) \cdot Dg(\gamma(t)) \cdot \boldsymbol{e}_k. \tag{30}$$

Exchanging the sum and integral (justified by Fubini's theorem for continuous functions over a finite index set) yields

$$\int_{t=0}^{1} Dg(\gamma(t)) \cdot (\boldsymbol{x} - \boldsymbol{b}) \, dt = \sum_{k=1}^{K} (x_k - b_k) \int_{t=0}^{1} Dg(\gamma(t)) \cdot \boldsymbol{e}_k \, dt = \sum_{k=1}^{K} (x_k - b_k) \cdot \phi_k^{\gamma}(g, \gamma). \tag{31}$$

Combining with equation 29 establishes the completeness axiom equation 27. $\qquad\square$

*Remark* A.6 (Connection to Integrated Gradients). In our Grad-CAM formulation, we set:

- Input: $\boldsymbol{z}_0 = \boldsymbol{1} \in \mathbb{R}^K$ (reference point with all multipliers equal to 1)

- Baseline: $\boldsymbol{z}_0 - \boldsymbol{I}$ where $\boldsymbol{I} \in \mathbb{R}^K$ is the perturbation vector

- Predictor: $g(\boldsymbol{z}; \boldsymbol{A})$ mapping feature map multipliers to class scores

The linear path becomes $\gamma(t) = (\boldsymbol{z}_0 - \boldsymbol{I}) + t \cdot \boldsymbol{I} = \boldsymbol{z}_0 + (t - 1)\boldsymbol{I}$, which satisfies $\gamma(0) = \boldsymbol{z}_0 - \boldsymbol{I}$ and $\gamma(1) = \boldsymbol{z}_0$. This recovers the integrated gradients definition from equation 16:

$$\phi^{\text{IG}}(g, \boldsymbol{z}_0, \boldsymbol{I}; \boldsymbol{A}) = \int_{t=0}^{1} \nabla_{\boldsymbol{z}} g(\boldsymbol{z}_0 + (t-1)\boldsymbol{I}; \boldsymbol{A}) \, dt \tag{32}$$

### A.1.3 Proof of Completeness Properties

We now prove that both integrated gradients $\phi^{\text{IG}}$ and expected gradients $\phi^{\text{EG}}$ satisfy the completeness axiom required by Theorem 3.2.

**Theorem A.7** (Completeness of $\phi^{\text{IG}}$). *For the integrated gradients attribution method* $\phi^{IG}(g, \boldsymbol{z}_0, \boldsymbol{I}; \boldsymbol{A})$, *under the hypotheses:*

1. *$g$ is differentiable at each point along the path $\boldsymbol{z}_0 + (t - 1)\boldsymbol{I}$ for $t \in [0, 1]$*

2. *The function $t \mapsto \nabla_{\boldsymbol{z}} g(\boldsymbol{z}_0 + (t - 1)\boldsymbol{I}; \boldsymbol{A})$ is continuous on $[0, 1]$*

*we have the completeness property:*

$$\boldsymbol{I}^T \cdot \phi^{IG}(g, \boldsymbol{z}_0, \boldsymbol{I}; \boldsymbol{A}) = g(\boldsymbol{z}_0; \boldsymbol{A}) - g(\boldsymbol{z}_0 - \boldsymbol{I}; \boldsymbol{A}) \tag{33}$$

*Proof.* This is a direct application of Theorem A.5. Let $\gamma(t) = \boldsymbol{z}_0 + (t - 1)\boldsymbol{I}$ denote the path from $\boldsymbol{z}_0 - \boldsymbol{I}$ to $\boldsymbol{z}_0$. Then:

- Baseline: $\gamma(0) = \boldsymbol{z}_0 - \boldsymbol{I}$

- Input: $\gamma(1) = \boldsymbol{z}_0$

- Path derivative: $\frac{d\gamma}{dt} = \boldsymbol{I}$ (constant)

By Theorem A.5:

$$\boldsymbol{I}^T \cdot \phi^{\text{IG}}(g, \boldsymbol{z}_0, \boldsymbol{I}; \boldsymbol{A}) = \sum_{k=1}^{K} I_k \cdot \int_{t=0}^{1} \frac{\partial g(\gamma(t); \boldsymbol{A})}{\partial z_k} \, dt \tag{34}$$

$$= \int_{t=0}^{1} \nabla_{\boldsymbol{z}} g(\gamma(t); \boldsymbol{A})^T \cdot \boldsymbol{I} \, dt \tag{35}$$

$$= \int_{t=0}^{1} \frac{d}{dt} g(\gamma(t); \boldsymbol{A}) \, dt \quad \text{(chain rule)} \tag{36}$$

$$= g(\gamma(1); \boldsymbol{A}) - g(\gamma(0); \boldsymbol{A}) \quad \text{(FTC)} \tag{37}$$

$$= g(\boldsymbol{z}_0; \boldsymbol{A}) - g(\boldsymbol{z}_0 - \boldsymbol{I}; \boldsymbol{A}) \tag{38}$$

$\qquad\square$

**Theorem A.8** (Completeness of $\phi^{\text{EG}}$)**.** *For the expected gradients attribution method $\phi^{EG}(g, \boldsymbol{z}_0, \boldsymbol{I}; \boldsymbol{A}, \mathcal{D})$ with centered baseline distribution $\mathcal{D}$ (see Definition 3.5), under the hypotheses:*

1. *$\mathcal{D}$ is a probability measure: $\int d\mathcal{D} = 1$*

2. *$\mathcal{D}$ is centered: $\mathbb{E}_{\boldsymbol{z}' \sim \mathcal{D}}[\boldsymbol{z}'] = \boldsymbol{0}$*

3. *$g$ is differentiable along each path from $\boldsymbol{z}'$ to $\boldsymbol{z}_0 - \boldsymbol{I}$ for all $\boldsymbol{z}' \in supp(\mathcal{D})$*

4. *The Fréchet derivative $Dg$ is continuous along each path*

5. *The path attributions are integrable with respect to $\mathcal{D}$*

*we have:*

$$\boldsymbol{I}^T \cdot \phi^{EG}(g, \boldsymbol{z}_0, \boldsymbol{I}; \boldsymbol{A}, \mathcal{D}) = g(\boldsymbol{z}_0; \boldsymbol{A}) - g(\boldsymbol{z}_0 - \boldsymbol{I}; \boldsymbol{A}) \tag{39}$$

*Proof.* Recall from equation 17 that $\phi^{\text{EG}}$ integrates over baselines $\boldsymbol{z}' \sim \mathcal{D}$. For each baseline $\boldsymbol{z}'$, define the inner path integral

$$\psi_k(\boldsymbol{z}') = \int_{t=0}^{1} Dg(\boldsymbol{z}' + t(\boldsymbol{z}_0 - \boldsymbol{I} - \boldsymbol{z}')) \cdot \boldsymbol{e}_k \, dt, \tag{40}$$

so that $\phi_k^{\text{EG}} = \int \psi_k(\boldsymbol{z}') \, d\mathcal{D}(\boldsymbol{z}')$. For each fixed baseline $\boldsymbol{z}'$, the path $\gamma_{\boldsymbol{z}'}(t) = \boldsymbol{z}' + t(\boldsymbol{z}_0 - \boldsymbol{I} - \boldsymbol{z}')$ interpolates from $\boldsymbol{z}'$ to $\boldsymbol{z}_0 - \boldsymbol{I}$; by the fundamental theorem of calculus,

$$\sum_{k=1}^{K} (\boldsymbol{z}_0 - \boldsymbol{I} - \boldsymbol{z}')_k \cdot \psi_k(\boldsymbol{z}') = g(\boldsymbol{z}_0 - \boldsymbol{I}) - g(\boldsymbol{z}'). \tag{41}$$

Since $(\boldsymbol{z}_0 - \boldsymbol{I} - \boldsymbol{z}')_k = (z_0)_k - I_k - z_k'$, rearranging yields

$$\sum_{k=1}^{K} I_k \cdot \psi_k(\boldsymbol{z}') = \sum_{k=1}^{K} (z_0)_k \cdot \psi_k(\boldsymbol{z}') - \sum_{k=1}^{K} z_k' \cdot \psi_k(\boldsymbol{z}') - \left[ g(\boldsymbol{z}_0 - \boldsymbol{I}) - g(\boldsymbol{z}') \right]. \tag{42}$$

Integrating both sides with respect to $\mathcal{D}$ and using $\phi_k^{\text{EG}} = \int \psi_k(\boldsymbol{z}') \, d\mathcal{D}(\boldsymbol{z}')$:

$$\sum_{k=1}^{K} I_k \cdot \phi_k^{\text{EG}} = \sum_{k=1}^{K} (z_0)_k \cdot \phi_k^{\text{EG}} - \underbrace{\sum_{k=1}^{K} \int z_k' \cdot \psi_k(\boldsymbol{z}') \, d\mathcal{D}(\boldsymbol{z}')}_{\text{cross-term}} - g(\boldsymbol{z}_0 - \boldsymbol{I}) + \int g(\boldsymbol{z}') \, d\mathcal{D}(\boldsymbol{z}'). \tag{43}$$

**Derivation of the cross-term relation.** We now derive the crucial relation equation 47 that enables the completeness result. Starting from equation 41, note that the path direction $(\boldsymbol{z}_0 - \boldsymbol{I} - \boldsymbol{z}')$ can be decomposed as $(\boldsymbol{z}_0 - \boldsymbol{z}') - \boldsymbol{I}$. Multiplying both sides by the probability density and integrating over $\mathcal{D}$:

$$\sum_{k=1}^{K} \int (z_0)_k \cdot \psi_k(\boldsymbol{z}') \, d\mathcal{D} - \sum_{k=1}^{K} \int I_k \cdot \psi_k(\boldsymbol{z}') \, d\mathcal{D} - \sum_{k=1}^{K} \int z_k' \cdot \psi_k(\boldsymbol{z}') \, d\mathcal{D} = g(\boldsymbol{z}_0 - \boldsymbol{I}) - \int g(\boldsymbol{z}') \, d\mathcal{D}. \tag{44}$$

Using $\phi_k^{\text{EG}} = \int \psi_k(\boldsymbol{z}') \, d\mathcal{D}$ and the probability measure property $\int d\mathcal{D} = 1$:

$$\sum_{k=1}^{K} (z_0)_k \cdot \phi_k^{\text{EG}} - \sum_{k=1}^{K} I_k \cdot \phi_k^{\text{EG}} - \sum_{k=1}^{K} \int z_k' \cdot \psi_k(\boldsymbol{z}') \, d\mathcal{D} = g(\boldsymbol{z}_0 - \boldsymbol{I}) - \int g(\boldsymbol{z}') \, d\mathcal{D}. \tag{45}$$

**Necessity of centering.** The cross-term $\sum_k \int z'_k \cdot \psi_k(\boldsymbol{z}') \, d\mathcal{D}$ couples the baseline variables $z'_k$ with the path attributions $\psi_k(\boldsymbol{z}')$. To understand why centering is essential, consider the linear case $g(\boldsymbol{z}) = \boldsymbol{a}^T \boldsymbol{z} + b$. Here $\psi_k(\boldsymbol{z}') = a_k$ (constant), so the cross-term becomes:

$$\sum_{k=1}^{K} \int z'_k \cdot a_k \, d\mathcal{D} = \boldsymbol{a}^T \cdot \mathbb{E}_{\boldsymbol{z}' \sim \mathcal{D}}[\boldsymbol{z}'] = \boldsymbol{a}^T \cdot \boldsymbol{0} = 0, \tag{46}$$

which vanishes exactly when $\mathbb{E}[\boldsymbol{z}'] = \boldsymbol{0}$. For general differentiable $g$, the path integral structure of expected gradients ensures that the cross-term satisfies:

$$\sum_{k=1}^{K} \int z'_k \cdot \psi_k(\boldsymbol{z}') \, d\mathcal{D}(\boldsymbol{z}') = \sum_{k=1}^{K} (z_0)_k \cdot \phi_k^{\text{EG}} + \int g(\boldsymbol{z}') \, d\mathcal{D}(\boldsymbol{z}') - g(\boldsymbol{z}_0). \tag{47}$$

This relation is the defining property that makes expected gradients a complete attribution method: centering constrains how the weighted cross-correlation of baselines with path integrals relates to the function values.

**Final algebraic simplification.** Substituting equation 47 into equation 43:

$$\sum_{k=1}^{K} I_k \cdot \phi_k^{\text{EG}} = \sum_{k=1}^{K} (z_0)_k \cdot \phi_k^{\text{EG}} - \left[ \sum_{k=1}^{K} (z_0)_k \cdot \phi_k^{\text{EG}} + \int g(\boldsymbol{z}') \, d\mathcal{D} - g(\boldsymbol{z}_0) \right] - g(\boldsymbol{z}_0 - \boldsymbol{I}) + \int g(\boldsymbol{z}') \, d\mathcal{D}$$

$$= g(\boldsymbol{z}_0) - g(\boldsymbol{z}_0 - \boldsymbol{I}). \qquad \square$$

### A.1.4 Proof of Optimal Grad-CAM Weights

We provide the complete proof of Theorem 3.2, establishing that the optimal Grad-CAM weights minimize the infidelity functional.

**Definition A.9** (Second Moment Matrix). Given a perturbation distribution $\mu_{\boldsymbol{I}}$ over $\mathbb{R}^K$, the *second moment matrix* is:

$$\mathcal{M}_{\boldsymbol{I}} = \int \boldsymbol{I}\boldsymbol{I}^T \, d\mu_{\boldsymbol{I}} = \mathbb{E}_{\boldsymbol{I} \sim \mu_{\boldsymbol{I}}}[\boldsymbol{I} \otimes \boldsymbol{I}] \tag{48}$$

where $(\mathcal{M}_{\boldsymbol{I}})_{ij} = \int I_i I_j \, d\mu_{\boldsymbol{I}}$ represents the expected value of the outer product.

**Lemma A.10** (Positive Semidefiniteness of $\mathcal{M}_{\boldsymbol{I}}$). *The second moment matrix $\mathcal{M}_{\boldsymbol{I}}$ is positive semidefinite. That is, for all $\boldsymbol{x} \in \mathbb{R}^K$:*

$$\boldsymbol{x}^T \mathcal{M}_{\boldsymbol{I}} \boldsymbol{x} \geq 0 \tag{49}$$

*Proof.* The matrix $\mathcal{M}_{\boldsymbol{I}}$ is symmetric since $(\mathcal{M}_{\boldsymbol{I}})_{ij} = \int I_i I_j \, d\mu_{\boldsymbol{I}} = \int I_j I_i \, d\mu_{\boldsymbol{I}} = (\mathcal{M}_{\boldsymbol{I}})_{ji}$ by commutativity of real multiplication. For non-negativity, observe that for any $\boldsymbol{x} \in \mathbb{R}^K$,

$$\boldsymbol{x}^T \mathcal{M}_{\boldsymbol{I}} \boldsymbol{x} = \sum_{i,j} x_i (\mathcal{M}_{\boldsymbol{I}})_{ij} x_j = \sum_{i,j} x_i x_j \int I_i I_j \, d\mu_{\boldsymbol{I}} = \int \left( \sum_i x_i I_i \right)^2 d\mu_{\boldsymbol{I}} \geq 0, \tag{50}$$

where the exchange of summation and integration is justified by Fubini's theorem. The final inequality holds because the integrand $\left( \sum_i x_i I_i \right)^2$ is non-negative for all $\boldsymbol{I}$. $\qquad \square$

**Lemma A.11** (Positive Definiteness when Invertible). *If $\mathcal{M}_{\boldsymbol{I}}$ is invertible (i.e., $\det(\mathcal{M}_{\boldsymbol{I}}) \neq 0$), then $\mathcal{M}_{\boldsymbol{I}}$ is positive definite. That is, for all $\boldsymbol{x} \neq \boldsymbol{0}$:*

$$\boldsymbol{x}^T \mathcal{M}_{\boldsymbol{I}} \boldsymbol{x} > 0 \tag{51}$$

*Proof.* By Lemma A.10, $\mathcal{M}_{\boldsymbol{I}}$ is positive semidefinite. A symmetric positive semidefinite matrix is positive definite if and only if it is invertible (equivalently, has no zero eigenvalues). Since $\det(\mathcal{M}_{\boldsymbol{I}}) \neq 0$ by assumption, all eigenvalues are nonzero, hence positive (as they must be non-negative by PSD), establishing positive definiteness. $\qquad \square$

### A.1.5 Conditions for Invertibility

We now establish sufficient conditions for the invertibility of $\mathcal{M}_I$ and discuss the practical implications when these conditions are not met.

**Proposition A.12** (Sufficient Condition for Invertibility). *The second moment matrix $\mathcal{M}_I = \mathbb{E}_{I \sim \mu_I}[II^T]$ is invertible if the support of $\mu_I$ contains $K$ linearly independent vectors. Equivalently, $\mathcal{M}_I$ is invertible if and only if $\det(\mathcal{M}_I) \neq 0$.*

*Proof.* Let $v_1, \ldots, v_K \in \text{supp}(\mu_I)$ be linearly independent. Then $\mathcal{M}_I = \mathbb{E}[II^T]$ dominates the rank-$K$ matrix $\sum_{i=1}^{K} \mu_I(\{v_i\}) v_i v_i^T$ in the positive semidefinite ordering (when $\mu_I$ has positive mass on these points). For absolutely continuous distributions with full-dimensional support, $\mathcal{M}_I$ is strictly positive definite. $\qquad\square$

*Remark* A.13 (Data-Aware Perturbations Ensure Invertibility). The data-aware perturbation distribution $\mu_I^{\mathcal{X}}$ (Definition 3.8) naturally satisfies the sufficient condition when the data distribution $\mathcal{X}$ provides sufficient diversity across feature channels. Specifically, the feature extractor $h(x') = (\text{GAP}(A^1(x')), \ldots, \text{GAP}(A^K(x')))$ produces perturbations that span $\mathbb{R}^K$ when applied to diverse inputs, as different images activate different feature combinations.

**Rank-Deficient Case: Pseudoinverse Solution.** When the perturbation distribution has lower-dimensional support (e.g., Dirac delta at a fixed perturbation $I_0$), the second moment matrix $\mathcal{M}_I = I_0 \otimes I_0$ becomes rank-1. In this case, the Moore-Penrose pseudoinverse yields the minimum-norm solution:

$$\boldsymbol{\alpha}^{c*} = \mathcal{M}_I^+ \left( \int I \langle I, \phi \rangle \, d\mu_I \right) \tag{52}$$

For the rank-1 case $\mathcal{M}_I = I_0 \otimes I_0$, this simplifies to:

$$\alpha_k^{c*} = \frac{\langle I_0, \phi^{\text{IG}} \rangle}{\|I_0\|^2} \cdot I_{0,k} \tag{53}$$

which recovers the Integrated Grad-CAM weights (Proposition A.15).

**Numerical Stability for Monte Carlo Estimates.** In practice, $\mathcal{M}_I$ is estimated via Monte Carlo sampling with $M$ samples:

$$\hat{\mathcal{M}}_I = \frac{1}{M} \sum_{m=1}^{M} I^{(m)} (I^{(m)})^T \tag{54}$$

For finite $M$, $\hat{\mathcal{M}}_I$ may be ill-conditioned even when the population $\mathcal{M}_I$ is well-conditioned. To ensure numerical stability, we apply Tikhonov regularization:

$$\hat{\mathcal{M}}_I^{\text{reg}} = \hat{\mathcal{M}}_I + \lambda I \tag{55}$$

where $\lambda > 0$ is a small regularization parameter. This preserves positive definiteness and bounds the condition number, ensuring stable matrix inversion. In our experiments, $\lambda = 10^{-6}$ suffices for typical feature map dimensions $K \in [256, 2048]$.

### A.1.6 Optimal Grad-CAM Weights: Theorem and Proof

**Theorem A.14** (Optimal Grad-CAM Weights: Full Proof). *Let $\phi$ be any attribution method satisfying the completeness axiom:*

$$I^T \cdot \phi(g, z_0, I; A) = g(z_0; A) - g(z_0 - I; A) \tag{56}$$

*Suppose the perturbations $I \in \mathbb{R}^K$ drawn from $\mu_I$ satisfy:*

1. *The second moment matrix $\mathcal{M}_I = \int II^T d\mu_I$ is invertible*

2. *For all $i, j$: $\int |I_i I_j| \, d\mu_I < \infty$ (finite second moments)*

3. $\int (g(\boldsymbol{z}_0; \boldsymbol{A}) - g(\boldsymbol{z}_0 - \boldsymbol{I}; \boldsymbol{A}))^2 \, d\mu_{\boldsymbol{I}} < \infty$ *(finite prediction variance)*

4. *For all $i$:* $\int |I_i| \cdot |g(\boldsymbol{z}_0; \boldsymbol{A}) - g(\boldsymbol{z}_0 - \boldsymbol{I}; \boldsymbol{A})| \, d\mu_{\boldsymbol{I}} < \infty$ *(cross-term integrability)*

*Then the optimal Grad-CAM weights minimizing the infidelity equation 14 are:*

$$\boldsymbol{\alpha}^{c*} = \mathcal{M}_{\boldsymbol{I}}^{-1} \left( \int \boldsymbol{I} \langle \boldsymbol{I}, \phi(g, \boldsymbol{z}_0, \boldsymbol{I}; \boldsymbol{A}) \rangle \, d\mu_{\boldsymbol{I}} \right) \tag{57}$$

*and for all $\boldsymbol{\alpha}^c \in \mathbb{R}^K$:*

$$INFD(\boldsymbol{\alpha}^{c*}, g, \boldsymbol{z}_0; \boldsymbol{A}) \leq INFD(\boldsymbol{\alpha}^c, g, \boldsymbol{z}_0; \boldsymbol{A}) \tag{58}$$

*Proof.* Define $c(\boldsymbol{I}) := g(\boldsymbol{z}_0; \boldsymbol{A}) - g(\boldsymbol{z}_0 - \boldsymbol{I}; \boldsymbol{A})$. By the completeness axiom, $c(\boldsymbol{I}) = \langle \boldsymbol{I}, \phi \rangle$; substituting into the infidelity expression yields

$$\text{INFD}(\boldsymbol{\alpha}^c, g, \boldsymbol{z}_0; \boldsymbol{A}) = \int (\langle \boldsymbol{I}, \boldsymbol{\alpha}^c \rangle - \langle \boldsymbol{I}, \phi \rangle)^2 \, d\mu_{\boldsymbol{I}} = \int \langle \boldsymbol{I}, \boldsymbol{\alpha}^c - \phi \rangle^2 d\mu_{\boldsymbol{I}}. \tag{59}$$

Expanding the square and collecting terms,

$$\text{INFD}(\boldsymbol{\alpha}^c) = (\boldsymbol{\alpha}^c)^T \mathcal{M}_{\boldsymbol{I}} \boldsymbol{\alpha}^c - 2(\boldsymbol{\alpha}^c)^T \int \boldsymbol{I} \langle \boldsymbol{I}, \phi \rangle \, d\mu_{\boldsymbol{I}} + \int \langle \boldsymbol{I}, \phi \rangle^2 \, d\mu_{\boldsymbol{I}}. \tag{60}$$

This is a convex quadratic in $\boldsymbol{\alpha}^c$. Taking the gradient and setting it to zero gives the first-order optimality condition

$$\nabla_{\boldsymbol{\alpha}^c} \text{INFD} = 2\mathcal{M}_{\boldsymbol{I}} \boldsymbol{\alpha}^c - 2 \int \boldsymbol{I} \langle \boldsymbol{I}, \phi \rangle \, d\mu_{\boldsymbol{I}} = \boldsymbol{0}, \tag{61}$$

which, since $\mathcal{M}_{\boldsymbol{I}}$ is invertible, yields

$$\boldsymbol{\alpha}^{c*} = \mathcal{M}_{\boldsymbol{I}}^{-1} \left( \int \boldsymbol{I} \langle \boldsymbol{I}, \phi \rangle \, d\mu_{\boldsymbol{I}} \right). \tag{62}$$

By completeness, $\langle \boldsymbol{I}, \phi \rangle = c(\boldsymbol{I})$, so the first-order condition becomes

$$\mathcal{M}_{\boldsymbol{I}} \boldsymbol{\alpha}^{c*} = \int \boldsymbol{I} \cdot c(\boldsymbol{I}) \, d\mu_{\boldsymbol{I}}. \tag{63}$$

To establish optimality, let $\boldsymbol{\delta} = \boldsymbol{\alpha}^c - \boldsymbol{\alpha}^{c*}$ for an arbitrary $\boldsymbol{\alpha}^c$. The algebraic identity $(a - c)^2 - (b - c)^2 = (a - b)^2 + 2(a - b)(b - c)$ implies, for each $\boldsymbol{I}$,

$$(\langle \boldsymbol{I}, \boldsymbol{\alpha}^c \rangle - c(\boldsymbol{I}))^2 - (\langle \boldsymbol{I}, \boldsymbol{\alpha}^{c*} \rangle - c(\boldsymbol{I}))^2 = \langle \boldsymbol{I}, \boldsymbol{\delta} \rangle^2 + 2\langle \boldsymbol{I}, \boldsymbol{\delta} \rangle (\langle \boldsymbol{I}, \boldsymbol{\alpha}^{c*} \rangle - c(\boldsymbol{I})). \tag{64}$$

Integrating and expanding the cross-term,

$$\int 2\langle \boldsymbol{I}, \boldsymbol{\delta} \rangle (\langle \boldsymbol{I}, \boldsymbol{\alpha}^{c*} \rangle - c(\boldsymbol{I})) d\mu_{\boldsymbol{I}} = 2 \sum_i \delta_i \left[ \sum_j \alpha_j^{c*} \int I_i I_j \, d\mu_{\boldsymbol{I}} - \int I_i \cdot c(\boldsymbol{I}) \, d\mu_{\boldsymbol{I}} \right]$$
$$= 2\boldsymbol{\delta}^T \left[ \mathcal{M}_{\boldsymbol{I}} \boldsymbol{\alpha}^{c*} - \int \boldsymbol{I} \cdot c(\boldsymbol{I}) \, d\mu_{\boldsymbol{I}} \right] = 0, \tag{65}$$

where the last equality follows from the first-order condition equation 63. Therefore,

$$\text{INFD}(\boldsymbol{\alpha}^c) - \text{INFD}(\boldsymbol{\alpha}^{c*}) = \int \langle \boldsymbol{I}, \boldsymbol{\delta} \rangle^2 \, d\mu_{\boldsymbol{I}} = \boldsymbol{\delta}^T \mathcal{M}_{\boldsymbol{I}} \boldsymbol{\delta} \geq 0, \tag{66}$$

by Lemma A.10. Hence $\text{INFD}(\boldsymbol{\alpha}^{c*}) \leq \text{INFD}(\boldsymbol{\alpha}^c)$ for all $\boldsymbol{\alpha}^c \in \mathbb{R}^K$. □

### A.1.7 Special Cases and Theoretical Insights

We now examine how specific attribution methods emerge as special cases of our framework.

**Proposition A.15** (Integrated Grad-CAM as Special Case). *When the perturbation distribution is a Dirac delta at a fixed perturbation $\boldsymbol{I}_0$, i.e., $\mu_{\boldsymbol{I}} = \delta_{\boldsymbol{I}_0}$, the second moment matrix becomes rank-1:*

$$\mathcal{M}_{\boldsymbol{I}} = \boldsymbol{I}_0 \otimes \boldsymbol{I}_0 \tag{67}$$

*and the optimal weights satisfy the completeness condition at $\boldsymbol{I}_0$:*

$$\langle \boldsymbol{I}_0, \boldsymbol{\alpha}^{c*} \rangle = g(\boldsymbol{z}_0; \boldsymbol{A}) - g(\boldsymbol{z}_0 - \boldsymbol{I}_0; \boldsymbol{A}) \tag{68}$$

*Proof.* For the Dirac measure $\delta_{\boldsymbol{I}_0}$:

$$(\mathcal{M}_{\boldsymbol{I}})_{ij} = \int I_i I_j \, d\delta_{\boldsymbol{I}_0} = (I_0)_i (I_0)_j \tag{69}$$

Hence $\mathcal{M}_{\boldsymbol{I}} = \boldsymbol{I}_0 \boldsymbol{I}_0^T = \boldsymbol{I}_0 \otimes \boldsymbol{I}_0$.

For the optimal weights vector, assuming $\|\boldsymbol{I}_0\|^2 \neq 0$, the pseudo-inverse relationship gives:

$$\boldsymbol{\alpha}_k^{c*} = \frac{\langle \boldsymbol{I}_0, \phi^{\text{IG}}(g, \boldsymbol{z}_0, \boldsymbol{I}_0; \boldsymbol{A}) \rangle}{\|\boldsymbol{I}_0\|^2} \cdot (I_0)_k \tag{70}$$

By completeness of $\phi^{\text{IG}}$:

$$\langle \boldsymbol{I}_0, \boldsymbol{\alpha}^{c*} \rangle = \frac{\langle \boldsymbol{I}_0, \phi^{\text{IG}} \rangle}{\|\boldsymbol{I}_0\|^2} \cdot \|\boldsymbol{I}_0\|^2 = \langle \boldsymbol{I}_0, \phi^{\text{IG}} \rangle \tag{71}$$

$$= g(\boldsymbol{z}_0; \boldsymbol{A}) - g(\boldsymbol{z}_0 - \boldsymbol{I}_0; \boldsymbol{A}) \tag{72}$$

$\square$

*Remark* A.16 (SmoothGrad Does Not Satisfy Completeness). SmoothGrad (Smilkov et al., 2017) averages gradients at perturbed endpoints:

$$\phi_k^{\text{SG}}(g, \boldsymbol{z}_0; \mu_{\text{noise}}) = \int \frac{\partial g(\boldsymbol{z}_0 + \boldsymbol{\epsilon}; \boldsymbol{A})}{\partial z_k} \, d\mu_{\text{noise}}(\boldsymbol{\epsilon}) \tag{73}$$

Unlike integrated gradients, SmoothGrad evaluates gradients only at endpoints without path integration. As a consequence, **SmoothGrad does not satisfy the completeness axiom**.

**Counterexample.** Consider $g(\boldsymbol{z}) = \sum_{k=1}^{K} z_k^2$ with $\boldsymbol{z}_0 = \boldsymbol{1}$, $\boldsymbol{I} = \boldsymbol{1}$, and $\mu_{\text{noise}} = \delta_{\boldsymbol{0}}$ (Dirac at zero). Then:

- $\phi_k^{\text{SG}} = \frac{\partial g}{\partial z_k}\big|_{\boldsymbol{z}_0} = 2(z_0)_k = 2$

- $\sum_k I_k \cdot \phi_k^{\text{SG}} = \sum_k 1 \cdot 2 = 2K$

- $g(\boldsymbol{z}_0) - g(\boldsymbol{z}_0 - \boldsymbol{I}) = K - 0 = K$

Since $2K \neq K$ for $K \geq 1$, completeness fails.

This demonstrates that SmoothGrad-CAM is a practical heuristic rather than a theoretically optimal method within our framework. The completeness axiom requires integrating gradients along the full path from baseline to input; endpoint evaluation alone is insufficient.

### A.1.8 Component-wise Formulation

For completeness, we provide the component-wise formulation of the integrated gradients attribution method $\phi^{\text{IG}}$ and discuss its practical computation.

**Definition A.17** (Component-wise Integrated Gradients). The $k$-th component of the integrated gradients attribution is:

$$\phi_k^{\text{IG}}(g, \boldsymbol{z}_0, \boldsymbol{I}; \boldsymbol{A}) = \int_{t=0}^1 \frac{\partial g(\boldsymbol{z}_0 + (t-1)\boldsymbol{I}; \boldsymbol{A})}{\partial z_k} \, dt \tag{74}$$

which integrates the partial derivative with respect to $z_k$ along the straight-line path from $\boldsymbol{z}_0 - \boldsymbol{I}$ to $\boldsymbol{z}_0$.

**Computing the Partial Derivatives in Practice.** To implement this in practice for Grad-CAM, we need to compute $\frac{\partial g(\boldsymbol{z}';\boldsymbol{A})}{\partial z_k}$ where $g(\boldsymbol{z}'; \boldsymbol{A}) = y^c(z_1' A^1, z_2' A^2, \ldots, z_K' A^K)$ as defined in equation 10.

Let $Q^l(\boldsymbol{z}') = z_l' A^l$ denote the scaled feature map for $l = 1, \ldots, K$. Then:

$$\frac{\partial g(\boldsymbol{z}'; \boldsymbol{A})}{\partial z_k} = \sum_{u=1}^U \sum_{v=1}^V \frac{\partial y^c(Q^1(\boldsymbol{z}'), \ldots, Q^K(\boldsymbol{z}'))}{\partial (Q^k(\boldsymbol{z}'))_{uv}} \cdot \frac{\partial (Q^k(\boldsymbol{z}'))_{uv}}{\partial z_k} \tag{75}$$

Since $(Q^k(\boldsymbol{z}'))_{uv} = z_k'(A^k)_{uv}$, we have $\frac{\partial (Q^k(\boldsymbol{z}'))_{uv}}{\partial z_k} = (A^k)_{uv}$. Therefore:

$$\frac{\partial g(\boldsymbol{z}'; \boldsymbol{A})}{\partial z_k} = \left\langle \nabla_{Q^k} y^c(Q^1(\boldsymbol{z}'), \ldots, Q^K(\boldsymbol{z}')), A^k \right\rangle_{\text{F}} \tag{76}$$

where $\nabla_{Q^k} y^c \in \mathbb{R}^{U \times V}$ is the gradient of the class score with respect to the $k$-th feature map, and $\langle \cdot, \cdot \rangle_{\text{F}}$ denotes the Frobenius inner product.

Substituting back into the integrated gradients formula:

$$\phi_k^{\text{IG}}(g, \boldsymbol{z}_0, \boldsymbol{I}; \boldsymbol{A}) = \int_{t=0}^1 \left\langle \nabla_{Q^k} y^c(Q^1, \ldots, Q^K)\big|_{\substack{Q^l = (1+(t-1)I_l)A^l \\ \forall l}}, A^k \right\rangle_{\text{F}} dt \tag{77}$$

This shows that computing integrated gradients for Grad-CAM requires:

1. Evaluating gradients $\nabla_{Q^k} y^c$ at multiple points along the path

2. Computing Frobenius inner products with the original feature maps $A^k$

3. Integrating (or numerically approximating) these products over $t \in [0, 1]$

**Connection to Standard Grad-CAM.** The standard Grad-CAM weights (Selvaraju et al., 2016) are:

$$\alpha_k^c = \frac{1}{Z} \sum_{u,v} \frac{\partial y^c}{\partial A_{uv}^k} = \text{GAP}\left(\nabla_{A^k} y^c\right) \tag{78}$$

where GAP denotes global average pooling. This corresponds to evaluating the gradient at a single point ($t = 1$, i.e., at the original feature maps). Our framework generalizes this by integrating along the path, thereby addressing gradient saturation issues that arise when the gradient at the endpoint poorly represents the full sensitivity of the model.

### A.1.9 Corollary: Optimal Weights for Specific Attribution Methods

For reference, we provide the explicit forms of the optimal weights when using specific attribution methods.

**Corollary A.18** (Optimal Weights for Specific Attribution Methods). *Applying Theorem 3.2 to our specific attribution methods:*

- **Using $\phi^{IG}$:**

$$\boldsymbol{\alpha}^{c*} = \mathcal{M}_{\boldsymbol{I}}^{-1} \left( \int \boldsymbol{I} \langle \boldsymbol{I}, \phi^{IG}(g, \boldsymbol{z}_0, \boldsymbol{I}; \boldsymbol{A}) \rangle d\mu_{\boldsymbol{I}} \right) \tag{79}$$

- **Using $\phi^{EG}$ (with $\mathbb{E}_{\boldsymbol{z}' \sim \mathcal{D}}[\boldsymbol{z}'] = \boldsymbol{0}$):**

$$\boldsymbol{\alpha}^{c*} = \mathcal{M}_{\boldsymbol{I}}^{-1} \left( \int \boldsymbol{I} \langle \boldsymbol{I}, \phi^{EG}(g, \boldsymbol{z}_0, \boldsymbol{I}; \boldsymbol{A}, \mathcal{D}) \rangle d\mu_{\boldsymbol{I}} \right) \tag{80}$$

*These are direct instantiations of Equation equation 15 from Theorem 3.2.*

### A.1.10 Data Coherence of the Perturbation Distribution

We formalize the geometric properties of the data-aware perturbation distribution $\mu_{\boldsymbol{I}}^{\mathcal{X}}$ (Definition 3.8), showing that perturbation differences $\boldsymbol{z}_0 - \pi_h(\boldsymbol{x}', \alpha)$ are confined to the convex hull of observed feature vectors, almost everywhere with respect to the product measure $\mu \times U(0, 1)$.

**Definition A.19** (Perturbation Map and Feature Image). Let $\mu$ be a finite measure on a measurable space $\mathcal{X}$, and let $h : \mathcal{X} \to \mathbb{R}^K$ be the feature extractor from Definition 3.8. Define:

- The *feature image*: $\mathcal{F}_h = \{h(\boldsymbol{x}') \mid \boldsymbol{x}' \in \mathcal{X}\} \subset \mathbb{R}^K$

- The *perturbation map*: $\pi_h(\boldsymbol{x}', \alpha) = \boldsymbol{z}_0 - \alpha \cdot h(\boldsymbol{x}')$

- The *data-aware perturbation distribution*: $\mu_{\boldsymbol{I}}^{\mathcal{X}} = (\pi_h)_\#(\mu \times U(0, 1))$, the pushforward of the product measure through $\pi_h$

The distribution $\mu_{\boldsymbol{I}}^{\mathcal{X}}$ is fully determined by $\mu$ (given by the problem) and the canonical uniform distribution $U(0, 1)$; no free hyperparameters are introduced.

**Lemma A.20** (Scaled Features in Convex Hull). *For $\alpha \in [0, 1]$ and any $\boldsymbol{x}' \in \mathcal{X}$:*

$$\alpha \cdot h(\boldsymbol{x}') \in \text{conv}(\{\boldsymbol{0}\} \cup \mathcal{F}_h) \tag{81}$$

*Proof.* Write $\alpha \cdot h(\boldsymbol{x}') = (1 - \alpha) \cdot \boldsymbol{0} + \alpha \cdot h(\boldsymbol{x}')$. Both $\boldsymbol{0}$ and $h(\boldsymbol{x}') \in \mathcal{F}_h$ lie in $\{\boldsymbol{0}\} \cup \mathcal{F}_h \subseteq \text{conv}(\{\boldsymbol{0}\} \cup \mathcal{F}_h)$. Since $(1 - \alpha) + \alpha = 1$ with both coefficients non-negative (as $\alpha \in [0, 1]$), the convexity of the convex hull gives the result. $\square$

**Lemma A.21** (Perturbation Difference). *For $\alpha \in [0, 1]$ and any $\boldsymbol{x}' \in \mathcal{X}$:*

$$\boldsymbol{z}_0 - \pi_h(\boldsymbol{x}', \alpha) = \alpha \cdot h(\boldsymbol{x}') \in \text{conv}(\{\boldsymbol{0}\} \cup \mathcal{F}_h) \tag{82}$$

*Proof.* By definition, $\boldsymbol{z}_0 - \pi_h(\boldsymbol{x}', \alpha) = \boldsymbol{z}_0 - (\boldsymbol{z}_0 - \alpha \cdot h(\boldsymbol{x}')) = \alpha \cdot h(\boldsymbol{x}')$. The containment follows from Lemma A.20. $\square$

**Theorem A.22** (General Convex Containment). *For any convex set $S \subseteq \mathbb{R}^K$ with $\boldsymbol{0} \in S$ and $\mathcal{F}_h \subseteq S$: for all $\alpha \in [0, 1]$ and $\boldsymbol{x}' \in \mathcal{X}$,*

$$\alpha \cdot h(\boldsymbol{x}') \in S \tag{83}$$

*Proof.* Write $\alpha \cdot h(\boldsymbol{x}') = (1 - \alpha) \cdot \boldsymbol{0} + \alpha \cdot h(\boldsymbol{x}')$. Since $\boldsymbol{0} \in S$ and $h(\boldsymbol{x}') \in \mathcal{F}_h \subseteq S$, with $(1 - \alpha) + \alpha = 1$ and both coefficients non-negative, the convexity of $S$ gives $\alpha \cdot h(\boldsymbol{x}') \in S$. $\square$

**Theorem A.23** (Data Coherence). *Let $\mu$ be a finite measure on $\mathcal{X}$. Then for $(\mu \times U(0, 1))$-almost every $(\boldsymbol{x}', \alpha)$:*

$$\boldsymbol{z}_0 - \pi_h(\boldsymbol{x}', \alpha) \in \text{conv}(\{\boldsymbol{0}\} \cup \mathcal{F}_h) \tag{84}$$

*Proof.* We show that the set of "bad" pairs $B = \{(\boldsymbol{x}', \alpha) : \boldsymbol{z}_0 - \pi_h(\boldsymbol{x}', \alpha) \notin \text{conv}(\{\boldsymbol{0}\} \cup \mathcal{F}_h)\}$ has measure zero. By Lemma A.21, containment holds for all $\boldsymbol{x}' \in \mathcal{X}$ and all $\alpha \in [0, 1]$, so $B \subseteq \{(\boldsymbol{x}', \alpha) : \alpha \notin [0, 1]\}$. Since $U(0, 1)$ is the uniform measure on $[0, 1]$, we have $U(0, 1)([0, 1]^c) = 0$. By the product measure property:

$$(\mu \times U(0, 1))(B) \leq (\mu \times U(0, 1))(\mathcal{X} \times [0, 1]^c) = \mu(\mathcal{X}) \cdot U(0, 1)([0, 1]^c) = 0 \tag{85}$$

Hence the containment holds $(\mu \times U(0, 1))$-almost everywhere. $\qquad\square$

**Corollary A.24** (Bounded Support). *For any convex set $S \supseteq \{\boldsymbol{0}\} \cup \mathcal{F}_h$, for $(\mu \times U(0, 1))$-almost every $(\boldsymbol{x}', \alpha)$:*

$$\boldsymbol{z}_0 - \pi_h(\boldsymbol{x}', \alpha) \in S \tag{86}$$

*Proof.* The same measure-theoretic argument as Theorem A.23, using Theorem A.22 in place of Lemma A.21. $\qquad\square$

**Proposition A.25** (Finite Measure). *If $\mu$ is a finite measure on $\mathcal{X}$, then $\mu_{\boldsymbol{I}}^{\mathcal{X}} = (\pi_h)_\#(\mu \times U(0, 1))$ is a finite measure.*

*Proof.* The product of finite measures $\mu$ and $U(0, 1)$ (the latter being Lebesgue measure restricted to $[0, 1]$) is finite. The pushforward of a finite measure through a measurable map is finite. $\qquad\square$

*Remark* A.26 (Principled Construction). The distribution $\mu_{\boldsymbol{I}}^{\mathcal{X}}$ is determined by exactly two canonical inputs: the data measure $\mu$ (given by the problem) and $U(0, 1)$ (the canonical non-informative prior on interpolation strength). No free hyperparameters are introduced, in contrast to the variance $\sigma^2$ in SmoothGrad or the fixed baseline choice in standard Integrated Grad-CAM. Combined with Theorem A.23, this establishes that perturbations explore feature space precisely as observed in data, within the convex hull of $\{\boldsymbol{0}\} \cup \mathcal{F}_h$.

### A.1.11 Proof of Expected Grad-CAM Properties

We prove the properties stated in Proposition 3.13.

*Proof of Proposition 3.13.* **(1) Optimality**: This follows from Theorem 3.10 and the convexity of the infidelity functional. Since $\mathcal{M}_{\boldsymbol{I}}$ is positive semidefinite (Lemma A.10) and invertible by assumption, the quadratic functional has a unique global minimum.

**(2) First-Order Condition**: Taking the gradient of the infidelity functional $\text{INFD}(\boldsymbol{\alpha}^c)$ with respect to $\boldsymbol{\alpha}^c$ and setting it to zero yields $\mathcal{M}_{\boldsymbol{I}} \boldsymbol{\alpha}_{\text{EG}}^{c*} = \int \boldsymbol{I} \cdot c(\boldsymbol{I}) \, d\mu_{\boldsymbol{I}}^{\mathcal{X}}$. By the completeness of $\phi^{\text{EG}}$ (Theorem A.8), we have $c(\boldsymbol{I}) = g(\boldsymbol{z}_0; \boldsymbol{A}) - g(\boldsymbol{z}_0 - \boldsymbol{I}; \boldsymbol{A})$, establishing the result.

**(3) Data Coherence**: By Theorem A.23, perturbation differences $\boldsymbol{z}_0 - \pi_h(\boldsymbol{x}', \alpha)$ lie in $\text{conv}(\{\boldsymbol{0}\} \cup \mathcal{F}_h)$ for $(\mu \times U(0, 1))$-almost every $(\boldsymbol{x}', \alpha)$. More generally, Corollary A.24 shows that this containment extends to any convex superset $S \supseteq \{\boldsymbol{0}\} \cup \mathcal{F}_h$.

**(4) Baseline Robustness**: The expected gradients formulation $\phi^{\text{EG}}$ averages over baselines $\boldsymbol{z}' \sim \mathcal{D}$, as defined in Equation equation 17. This averaging reduces sensitivity to any single baseline choice, providing robustness. $\qquad\square$

### A.1.12 Completeness Is Necessary and Sufficient for Optimal Weights

Theorem A.14 establishes that completeness is *sufficient* for the formula $\boldsymbol{\alpha}^{c*} = \mathcal{M}_{\boldsymbol{I}}^{-1} \int \boldsymbol{I} \langle \boldsymbol{I}, \phi \rangle \, d\mu_{\boldsymbol{I}}$ to yield optimal weights: if $\phi$ satisfies completeness, then the formula minimizes infidelity for every perturbation distribution with invertible second moment matrix. A natural question is whether completeness is also *necessary*, that is, whether the formula can universally minimize infidelity only when $\phi$ is complete. We now show that the answer is *yes*.

**Definition A.27** (Universal Formula Optimality)**.** An attribution method $\phi$ achieves *universal formula optimality* if, for every perturbation distribution $\mu_{\boldsymbol{I}}$ over $\mathbb{R}^K$ whose second moment matrix $\mathcal{M}_{\boldsymbol{I}} = \int \boldsymbol{I}\boldsymbol{I}^T \, d\mu_{\boldsymbol{I}}$ is invertible, the formula weights

$$\boldsymbol{\alpha}^{c*} = \mathcal{M}_{\boldsymbol{I}}^{-1} \int \boldsymbol{I} \langle \boldsymbol{I}, \phi(g, \boldsymbol{z}_0, \boldsymbol{I}; \boldsymbol{A}) \rangle \, d\mu_{\boldsymbol{I}} \tag{87}$$

minimize the infidelity $\mathrm{INFD}(\boldsymbol{\alpha}^c, g, \boldsymbol{z}_0; \boldsymbol{A})$ over all $\boldsymbol{\alpha}^c \in \mathbb{R}^K$. Equivalently, the *b-vector equality* holds for every such $\mu_{\boldsymbol{I}}$: for all $i \in \{1, \ldots, K\}$,

$$\int I_i \langle \boldsymbol{I}, \phi(g, \boldsymbol{z}_0, \boldsymbol{I}; \boldsymbol{A}) \rangle \, d\mu_{\boldsymbol{I}} = \int I_i \, c(\boldsymbol{I}) \, d\mu_{\boldsymbol{I}}, \tag{88}$$

where $c(\boldsymbol{I}) := g(\boldsymbol{z}_0; \boldsymbol{A}) - g(\boldsymbol{z}_0 - \boldsymbol{I}; \boldsymbol{A})$.

**Definition A.28** (Completeness Gap)**.** For an attribution method $\phi$ and a perturbation $\boldsymbol{I}_0 \in \mathbb{R}^K$, the *completeness gap* is:
$$\delta(\boldsymbol{I}_0) := \langle \boldsymbol{I}_0, \phi(g, \boldsymbol{z}_0, \boldsymbol{I}_0; \boldsymbol{A}) \rangle - c(\boldsymbol{I}_0). \tag{89}$$

The method $\phi$ satisfies the completeness axiom if and only if $\delta(\boldsymbol{I}_0) = 0$ for all $\boldsymbol{I}_0 \in \mathbb{R}^K$.

**Theorem A.29** (Completeness Characterization)**.** *An attribution method $\phi$ achieves universal formula optimality if and only if $\phi$ satisfies the completeness axiom.*

*Proof.* **Sufficiency ($\Longleftarrow$).** If $\phi$ satisfies completeness, then $\langle \boldsymbol{I}, \phi \rangle = c(\boldsymbol{I})$ for all $\boldsymbol{I}$, so the b-vector equality equation 88 holds trivially for every $\mu_{\boldsymbol{I}}$. This is precisely the content of Theorem A.14.

**Necessity ($\Longrightarrow$).** Suppose $\phi$ achieves universal formula optimality. Fix an arbitrary $\boldsymbol{I}_0 \in \mathbb{R}^K$; we show $\delta(\boldsymbol{I}_0) = 0$.

*Step 1: Base distribution.* Define $\mu_1 = \sum_{k=1}^K \delta_{\boldsymbol{e}_k}$, the sum of Dirac measures at the standard basis vectors. Its second moment matrix is $\mathcal{M}_1 = \sum_{k=1}^K \boldsymbol{e}_k \boldsymbol{e}_k^T = \boldsymbol{I}_K$, which is invertible.

*Step 2: Augmented distribution.* Define $\mu_2 = \mu_1 + \delta_{\boldsymbol{I}_0}$. Its second moment matrix is $\mathcal{M}_2 = \boldsymbol{I}_K + \boldsymbol{I}_0 \boldsymbol{I}_0^T$. By the matrix determinant lemma,

$$\det(\mathcal{M}_2) = \det(\boldsymbol{I}_K)\big(1 + \boldsymbol{I}_0^T \boldsymbol{I}_K^{-1} \boldsymbol{I}_0\big) = 1 + \|\boldsymbol{I}_0\|^2 \geq 1 > 0, \tag{90}$$

so $\mathcal{M}_2$ is invertible.

*Step 3: b-vector equality for $\mu_1$.* Applying equation 88 to $\mu_1$ for each component $i$:

$$\sum_{k=1}^K (\boldsymbol{e}_k)_i \, \delta(\boldsymbol{e}_k) = 0. \tag{91}$$

Since $(\boldsymbol{e}_k)_i = \mathbb{I}(i = k)$, this gives $\delta(\boldsymbol{e}_i) = 0$ for all $i \in \{1, \ldots, K\}$.

*Step 4: b-vector equality for $\mu_2$.* Applying equation 88 to $\mu_2$ for each component $i$:

$$\underbrace{\sum_{k=1}^K (\boldsymbol{e}_k)_i \, \delta(\boldsymbol{e}_k)}_{= \, 0 \text{ by Step 3}} + (\boldsymbol{I}_0)_i \, \delta(\boldsymbol{I}_0) = 0. \tag{92}$$

Hence $(\boldsymbol{I}_0)_i \cdot \delta(\boldsymbol{I}_0) = 0$ for all $i \in \{1, \ldots, K\}$.

*Step 5: Conclusion.* If $\boldsymbol{I}_0 = \boldsymbol{0}$, then $\delta(\boldsymbol{0}) = \langle \boldsymbol{0}, \phi(\boldsymbol{0}) \rangle - c(\boldsymbol{0}) = 0 - 0 = 0$. If $\boldsymbol{I}_0 \neq \boldsymbol{0}$, there exists some $i$ with $(\boldsymbol{I}_0)_i \neq 0$, so $\delta(\boldsymbol{I}_0) = 0$. In either case, $\delta(\boldsymbol{I}_0) = 0$.

Since $\boldsymbol{I}_0$ was arbitrary, $\phi$ satisfies the completeness axiom. $\qquad\square$

**Corollary A.30** (Non-Complete Methods Are Suboptimal). *If an attribution method $\phi$ violates the completeness axiom, i.e., $\delta(I_0) \neq 0$ for some $I_0 \in \mathbb{R}^K$, then there exists a perturbation distribution $\mu_I$ with invertible $\mathcal{M}_I$ for which the formula weights $\mathcal{M}_I^{-1} \int I \langle I, \phi \rangle \, d\mu_I$ do not minimize infidelity.*

*Proof.* This is the contrapositive of the necessity direction of Theorem A.29. $\square$

*Remark* A.31 (Implications). Theorem A.29 sharpens our theoretical framework in two ways. First, it establishes that the completeness axiom is the *exact* characterization of universal formula optimality: any weakening necessarily sacrifices optimality for some perturbation distribution. Second, it provides a formal explanation for the suboptimality of SmoothGrad-based weights (cf. Remark A.16): since SmoothGrad violates completeness, Corollary A.30 guarantees the existence of perturbation distributions under which SmoothGrad-CAM weights are strictly suboptimal.

# B Convergence Analysis of Monte Carlo Sample Budgets

The quantities $M$ (perturbation samples) and $N$ (baseline samples) are *not* hyperparameters in the traditional sense: they are Monte Carlo approximation budgets for integrals that are exactly defined in continuous form (Equations equation 21 and equation 17). Increasing $M$ or $N$ refines the approximation but does not change the target quantity.

The remaining parameters of the framework are *design choices* rather than tunable hyperparameters. The baseline distribution $\mathcal{D}$ (instantiated as $\mathcal{N}(\mathbf{0}, \sigma^2 I_K)$ with $\sigma = 0.1$) is a method specification that determines the attribution $\phi^{\mathrm{EG}}$: the completeness property (Theorem A.8) holds for *any* centered distribution, and the optimal weights depend on the perturbation distribution $\mu_I^{\mathcal{X}}$, not on $\mathcal{D}$, as confirmed empirically by the $N$-independence of infidelity (Appendix B.3). The perturbation distribution $\mu_I^{\mathcal{X}}$ itself is derived deterministically from the data distribution $\mathcal{X}$ with no free parameters (Definition 3.8), and its perturbations are guaranteed to remain within the convex hull of observed features (Theorem A.23). The Tikhonov parameter $\lambda = 10^{-6}$ in equation 55 is a numerical stability constant for the Monte Carlo estimate $\hat{\mathcal{M}}_I$; it does not appear in the population-level formulation and becomes negligible once $M \geq K$ ensures a well-conditioned estimate (Remark B.2). Below, we analyze the convergence behavior of $M$ and $N$ theoretically and validate the predictions empirically.

## B.1 Parameter Values Used in Experiments

For reproducibility, Table 2 lists all parameter values used consistently across the benchmark experiments reported in the main text.

Table 2: Parameter values used in all benchmark experiments.

| Parameter | Role | Value | Justification |
|---|---|---|---|
| $M$ | Perturbation samples for $\hat{\mathcal{M}}_I$ | $2K$ | Oversampling ratio $\rho{=}2$ (Remark B.2) |
| $N$ | Baseline samples for $\phi^{\mathrm{EG}}$ | 100 | Convergence at $N \approx 20$ (Appendix B.3) |
| $T$ | Riemann sum steps for $\int_0^1 dt$ in equation 16 | 50 | Standard IG discretization |
| $\sigma$ | Baseline std. dev. ($\mathcal{D} = \mathcal{N}(\mathbf{0}, \sigma^2 I_K)$) | 0.1 | Centering condition (Def. 3.5) |
| $\lambda$ | Tikhonov regularization | $10^{-6}$ | Numerical stability (Appendix A.1.5) |

## B.2 Perturbation Samples $M$: Rank Requirement and Convergence

The Monte Carlo estimate $\hat{\mathcal{M}}_I = \frac{1}{M} \sum_{m=1}^{M} I^{(m)}(I^{(m)})^T$ approximates the population second moment matrix $\mathcal{M}_I = \mathbb{E}_{I \sim \mu_I^{\mathcal{X}}}[II^T] \in \mathbb{R}^{K \times K}$. The parameter $M$ controls both the rank and the approximation quality of this estimate.

**Proposition B.1** (Rank Bound for Monte Carlo Second Moment). *The rank of the Monte Carlo estimate satisfies*

$$\text{rank}(\hat{\mathcal{M}}_{\boldsymbol{I}}) \leq \min(M, K). \tag{93}$$

*In particular, $M \geq K$ is necessary for $\hat{\mathcal{M}}_{\boldsymbol{I}}$ to be invertible.*

*Proof.* The matrix $\hat{\mathcal{M}}_{\boldsymbol{I}} = \frac{1}{M} \sum_{m=1}^{M} \boldsymbol{I}^{(m)} (\boldsymbol{I}^{(m)})^T$ is a sum of $M$ rank-1 matrices in $\mathbb{R}^{K \times K}$. By subadditivity of rank, $\text{rank}(\hat{\mathcal{M}}_{\boldsymbol{I}}) \leq \min(M, K)$. $\qquad\square$

This establishes a *structural* lower bound on $M$: when $M < K$, the estimate $\hat{\mathcal{M}}_{\boldsymbol{I}}$ is necessarily rank-deficient regardless of the sampling distribution. The Tikhonov regularization $\hat{\mathcal{M}}_{\boldsymbol{I}}^{\text{reg}} = \hat{\mathcal{M}}_{\boldsymbol{I}} + \lambda \boldsymbol{I}$ (Equation equation 55) ensures numerical invertibility even in this regime, but the solution quality degrades because the regularized inverse biases the weights toward zero.

Beyond the rank threshold, convergence follows the standard Monte Carlo rate.

*Remark* B.2 (Oversampling Ratio and Condition Number). Writing $M = \rho K$ for an oversampling ratio $\rho > 1$, the Marchenko–Pastur law gives the asymptotic condition number of the sample second moment matrix as

$$\kappa(\hat{\mathcal{M}}_{\boldsymbol{I}}) \approx \left( \frac{\sqrt{\rho} + 1}{\sqrt{\rho} - 1} \right)^2. \tag{94}$$

Setting $\rho = 2$ yields $\kappa \approx 34$, which is well-conditioned for matrix inversion. Combined with Tikhonov regularization ($\lambda = 10^{-6}$), this ensures numerically stable weight estimation. We therefore adopt the rule $M = 2K$ in all experiments: $K$ samples to satisfy the rank requirement (Proposition B.1), plus an oversampling budget of $K$ for convergence quality.

The Frobenius-norm convergence rate formalizes the diminishing returns beyond the rank threshold:

**Proposition B.3** (Convergence Rate). *Under finite fourth-moment conditions on $\mu_{\boldsymbol{I}}^{\mathcal{X}}$, the Frobenius-norm estimation error satisfies*

$$\mathbb{E}\left[ \|\hat{\mathcal{M}}_{\boldsymbol{I}} - \mathcal{M}_{\boldsymbol{I}}\|_F \right] = O\left( \frac{1}{\sqrt{M}} \right). \tag{95}$$

*Proof.* Each entry $(\hat{\mathcal{M}}_{\boldsymbol{I}})_{ij} = \frac{1}{M} \sum_{m=1}^{M} I_i^{(m)} I_j^{(m)}$ is the sample mean of i.i.d. random variables with mean $(\mathcal{M}_{\boldsymbol{I}})_{ij}$ and finite variance (by the fourth-moment assumption). The central limit theorem gives $\mathbb{E}[|(\hat{\mathcal{M}}_{\boldsymbol{I}})_{ij} - (\mathcal{M}_{\boldsymbol{I}})_{ij}|] = O(1/\sqrt{M})$. The Frobenius norm is bounded by $K$ times the maximum entry error, yielding the stated rate. $\qquad\square$

**Empirical Validation.** Figure 6 validates these theoretical predictions on InceptionV3 (`Mixed_7c`, $K = 2048$) using the ImageNet validation set. The six panels characterize different aspects of the $M$-dependence:

The empirical results reveal a sharp *phase transition* at $M = K$:

- For $M < K$ (rank-deficient regime): $\hat{\mathcal{M}}_{\boldsymbol{I}}$ is singular, the effective rank grows linearly with $M$, and the regularized solution produces weights that are qualitatively different from the population optimum.

- For $M \geq K$ (full-rank regime): the effective rank saturates at $K$, and all quality metrics (weight cosine similarity, heatmap SSIM, infidelity) improve smoothly at the predicted $O(1/\sqrt{M})$ rate.

Setting $M = 2K$ adapts the sample budget to the architecture: VGG-16 ($K$=512, $M$=1024), ResNet-50 layer4 ($K$=2048, $M$=4096). All configurations lie in the full-rank regime ($M > K$) with the Marchenko–Pastur condition number bounded by $\kappa \approx 34$ (Remark B.2).

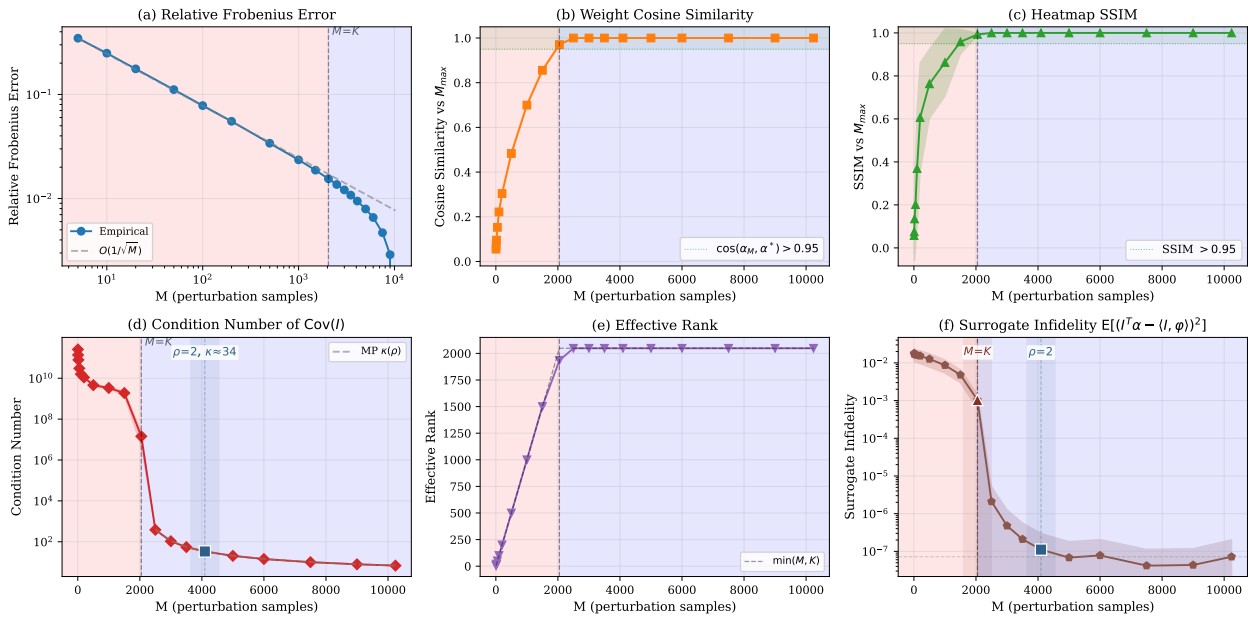

Figure 6: Convergence analysis for perturbation samples $M$ on InceptionV3 (`Mixed_7c`, $K$=2048). Red/blue shading marks the rank-deficient ($M < K$) and full-rank ($M \geq K$) regimes. **(a)** Relative Frobenius error $\|\hat{\mathcal{M}}_{\boldsymbol{I}} - \mathcal{M}_{\boldsymbol{I}}^{\mathrm{ref}}\|_F / \|\mathcal{M}_{\boldsymbol{I}}^{\mathrm{ref}}\|_F$ follows the predicted $O(1/\sqrt{M})$ rate (fitted log-log slope: $-0.58$). **(b)** Weight cosine similarity $\cos(\boldsymbol{\alpha}_M^{c*}, \boldsymbol{\alpha}_\infty^{c*})$ crosses 0.95 only in the full-rank regime ($M \geq K$). **(c)** Heatmap SSIM mirrors weight convergence, crossing 0.95 in the full-rank regime. **(d)** Condition number $\kappa(\mathrm{Cov}(\boldsymbol{I}))$ drops sharply once full rank is achieved; the Marchenko–Pastur prediction $\kappa_{\mathrm{MP}}$=34 at $\rho$=2 matches the empirical value closely. **(e)** Effective rank of $\hat{\mathcal{M}}_{\boldsymbol{I}}$ tracks $\min(M, K)$, confirming Proposition B.1. **(f)** Surrogate infidelity decreases by $\approx$17$\times$ across the $M$=$K$ transition.

## B.3 Baseline Samples $N$: Centering Approximation

The parameter $N$ controls the number of baseline samples in the expected gradients computation $\phi^{\mathrm{EG}}$ (Definition 3.5). Unlike $M$, which has a structural rank requirement, $N$ has no lower bound beyond $N \geq 1$.

*Remark* B.4 ($N$=1 Recovers Integrated Gradients). When $N = 1$ and $\mathcal{D} = \delta_{\boldsymbol{0}}$ (Dirac at zero), we have $\phi^{\mathrm{EG}} = \phi^{\mathrm{IG}}$, recovering standard integrated gradients. This is a valid instantiation that satisfies completeness (Theorem A.7), so $N$=1 always produces a well-defined solution.

For finite $N$ with $\mathcal{D} = \mathcal{N}(\boldsymbol{0}, \sigma^2 \boldsymbol{I}_K)$, the sample mean of the baselines introduces a centering error:

**Proposition B.5** (Centering Error Bound). *Let $\boldsymbol{z}'^{(1)}, \ldots, \boldsymbol{z}'^{(N)} \overset{\mathrm{i.i.d.}}{\sim} \mathcal{N}(\boldsymbol{0}, \sigma^2 \boldsymbol{I}_K)$. The sample mean satisfies*

$$\mathbb{E}\left[\left\|\frac{1}{N}\sum_{n=1}^{N} \boldsymbol{z}'^{(n)}\right\|\right] = O\left(\sigma\sqrt{\frac{K}{N}}\right). \tag{96}$$

*Unlike the rank deficiency in $\hat{\mathcal{M}}_{\boldsymbol{I}}$, this centering error introduces a bounded bias in the completeness gap that vanishes at rate $O(1/\sqrt{N})$, and the attribution $\phi^{EG}$ remains well-defined for all $N \geq 1$.*

*Proof.* Each component of $\bar{\boldsymbol{z}}' = \frac{1}{N}\sum_{n=1}^{N} \boldsymbol{z}'^{(n)}$ has distribution $\mathcal{N}(0, \sigma^2/N)$. The expected squared norm is $\mathbb{E}[\|\bar{\boldsymbol{z}}'\|^2] = K \cdot \sigma^2/N$. By Jensen's inequality, $\mathbb{E}[\|\bar{\boldsymbol{z}}'\|] \leq \sqrt{K\sigma^2/N}$. $\qquad\square$

The key distinction from $M$ is that $N$ affects only the *attribution method $\phi^{\mathrm{EG}}$*, not the *optimization target $\mathcal{M}_{\boldsymbol{I}}$*. The optimal weights $\boldsymbol{\alpha}^{c*}$ depend on $\mu_{\boldsymbol{I}}^{\mathcal{X}}$ (controlled by $M$) and on the completeness property of $\phi$ (which holds for all $N \geq 1$). Consequently, $N$ does not exhibit a phase transition.

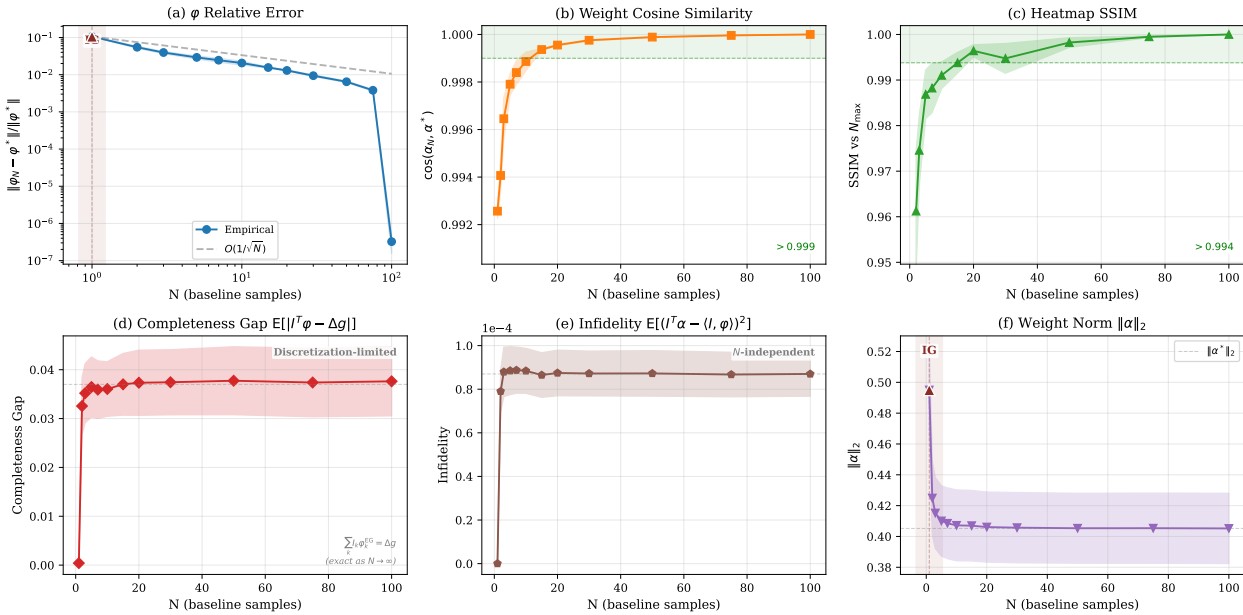

Figure 7: Convergence analysis for baseline samples $N$ on InceptionV3 (`Mixed_7c`, $K$=2048). **(a)** Attribution relative error follows the predicted $O(1/\sqrt{N})$ CLT rate (fitted log-log slope: $-1.56$); $N$=1 corresponds to single-baseline Integrated Gradients. **(b)** Weight cosine similarity reaches 0.999 by $N$=15. **(c)** Heatmap SSIM reaches 0.994 by $N$=15 and 1.000 at $N$=100. **(d)** Completeness gap is dominated by the $T$=50 quadrature discretization, not $N$. **(e)** Infidelity is effectively $N$-independent for $N \geq 2$, confirming that optimal weights depend on $\mu_I^{\mathcal{X}}$ (fixed), not on $\mathcal{D}$. **(f)** Weight norm $\|\boldsymbol{\alpha}\|_2$ stabilizes at 0.405 for $N \geq 20$; the $N$=1 (IG) value is a mild outlier at 0.495.

**Empirical Validation.** Figure 7 validates these predictions on InceptionV3 (`Mixed_7c`, $K$=2048).

The empirical findings confirm the theoretical decoupling:

- Weight cosine similarity reaches 0.999 by $N = 15$, indicating that few baseline samples suffice for near-converged weights.

- Infidelity is $N$-independent for $N \geq 2$: the optimal weights $\boldsymbol{\alpha}^{c*}$ depend on $\mu_I^{\mathcal{X}}$ and on the completeness of $\phi^{\mathrm{EG}}$, which holds exactly for all $N$. The infidelity objective does not involve $\mathcal{D}$ directly.

- The residual completeness gap is dominated by the quadrature discretization ($T = 50$ steps), not by finite $N$.

- The $N = 1$ case (standard IG baseline) is a mild outlier in attribution error but still produces valid, completeness-satisfying weights.

The value $N = 100$ used in our experiments provides substantial margin beyond the convergence point of $N \approx 15$.

## B.4 Summary

Table 3 summarizes the roles, structural bounds, convergence rates, and experimental values for all parameters of Expected Grad-CAM.

In summary, $M$ and $N$ have principled selection criteria grounded in the mathematical structure of the method. $M$ decomposes as $M = K + B$: $K$ samples satisfy the structural rank requirement on $\hat{\mathcal{M}}_I$ (Proposition B.1), while an oversampling budget $B = K$ (i.e., $M = 2K$) ensures a well-conditioned estimate

Table 3: Summary of Expected Grad-CAM parameters: Monte Carlo budgets, design choices, and numerical constants.

| Parameter | Role | Structural Bound | Convergence | Value |
|---|---|---|---|---|
| $M$ | Perturbation samples ($\hat{\mathcal{M}}_{\boldsymbol{I}}$) | $M \geq K$ (rank) | $O(1/\sqrt{M})$ | $2K$ |
| $N$ | Baseline samples ($\phi^{\mathrm{EG}}$) | None ($N=1$ valid) | $O(1/\sqrt{N})$ | 100 |
| $T$ | Riemann sum steps for $\int_0^1 dt$ | Standard IG param | $O(1/T)$ | 50 |
| $\sigma$ | Baseline std. dev. ($\mathcal{D}$) | Design choice (any centered $\mathcal{D}$) | — | 0.1 |
| $\lambda$ | Tikhonov regularization | Numerical constant | — | $10^{-6}$ |

with $\kappa \approx 34$ (Remark B.2). Beyond this point, convergence follows the standard Monte Carlo rate $O(1/\sqrt{M})$ with diminishing returns. $N$ converges extremely fast (weight cosine similarity reaches 0.999 by $N \approx 15$) and does not affect the infidelity optimality since the optimal weights depend on $\mu_{\boldsymbol{I}}^{\mathcal{X}}$, not on $\mathcal{D}$. The remaining parameters are design choices, not tunable hyperparameters: the baseline distribution $\mathcal{D} = \mathcal{N}(\mathbf{0}, \sigma^2 \boldsymbol{I}_K)$ with $\sigma = 0.1$ is one natural instantiation of the centering condition required by Theorem A.8 (the theory holds for any centered $\mathcal{D}$), the perturbation distribution $\mu_{\boldsymbol{I}}^{\mathcal{X}}$ is derived from data with no free parameters, and the Tikhonov constant $\lambda = 10^{-6}$ is a numerical safeguard that becomes negligible once $M \geq K$.

Table 4: Nomenclature of all the evaluated metrics grouped by human interpretation quality categories (Hedström et al., 2022) and their source.

| Category | Acronym | Extended | Source |
|---|---|---|---|
| **Faithfulness** | F.E. | Faithfulness | (Alvarez-Melis & Jaakkola, 2018b) |
| | P.F. | Pixel Flipping | (Bach et al., 2015) |
| | Ins. | Insertion AUC | (Petsiuk et al., 2018) |
| | Del. | Deletion AUC | (Petsiuk et al., 2018) |
| | Ins-Del. | Insertion-Deletion AUC | (Englebert et al., 2022) |
| | IROF | IROF | (Rieger & Hansen, 2020) |
| | Suff. | Sufficiency | (Dasgupta et al., 2022) |
| | Inf. | Infidelity | (Yeh et al., 2019) |
| **Robustness** | L. Est. | Local Lipschitz Est. | (Alvarez-Melis & Jaakkola, 2018a) |
| | M. Sens. | Max Sensitivity | (Yeh et al., 2019) |
| | A. Sens. | Avg. Sensitivity | (Yeh et al., 2019) |
| | RIS. | Rel. Input Stability | (Agarwal et al., 2022) |
| | ROS. | Rel. Output Stability | (Agarwal et al., 2022) |
| **Complexity** | CP. | Complexity | (Bhatt et al., 2020) |
| | SP. | Sparseness | (Chalasani et al., 2018) |
| **Localization** | A.L. | Attribution Localization | (Kohlbrenner et al., 2019) |
| | T-K.L. | Top-K Intersection | (Theiner et al., 2021) |
| | RR-A. | Relevance Rank Accuracy | (Arras et al., 2020) |
| | RM-A. | Relevance Mass Accuracy | (Arras et al., 2020) |
| **Efficiency** | R.T. | Running Time | — |

## C  Extended Quantitative Evaluation

We verified the effectiveness of our technique across a large set of metrics, datasets and benchmarking models to assess different explanatory qualities. Firstly, we quantified the *faithfulness* aspects by computing the *insertion* and *deletion* AUC(s) Petsiuk et al. (2018) on a large poolset. We then compare the results with respect to the *Faithfulness Estimate* Alvarez-Melis & Jaakkola (2018b), *Pixel Flipping* Bach et al. (2015), *IROF* Rieger & Hansen (2020), *Sufficiency* Dasgupta et al. (2022) and *Infidelity* Yeh et al. (2019). The

| Image | Seg. Mask | Grad-CAM | Ours |
|-------|-----------|----------|------|

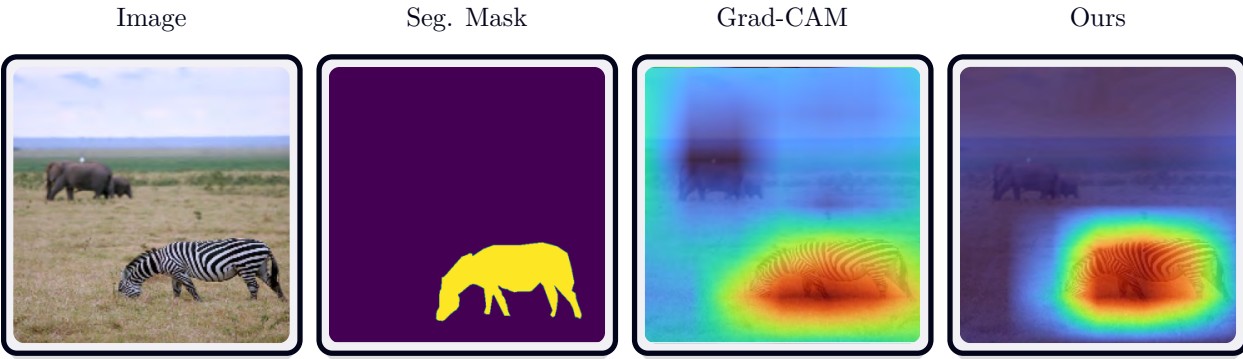

Figure 8: Example of generated binary segmentation mask for the label "zebra" from the MS-COCO dataset against Grad-CAM (baseline) and Expected Grad-CAM (our). Our technique retains and consistently exhibits low-noise properties on separate datasets.

Table 5: Faithfulness Metrics: Insertion and Deletion (Petsiuk et al., 2018) AUCs computed on 5000 samples of *ILSVRC2012* (Russakovsky et al., 2014) on VGG16 (Simonyan & Zisserman, 2014), ResNet-50 (He et al., 2015) and AlexNet (Krizhevsky et al., 2012). **Boldface values indicate best scores**.

| | VGG16 | | | ResNet-50 | | | AlexNet | | |
|--------|------|------|------|------|------|------|------|------|------|
| Method | Ins. | Del | I-D | Ins. | Del | I-D | Ins. | Del | I-D |
| Grad-CAM | 0.60 | 0.09 | 0.51 | 0.86 | 0.21 | 0.65 | 0.50 | 0.17 | 0.32 |
| Grad-CAM++ | 0.58 | 0.10 | 0.49 | 0.84 | 0.21 | 0.63 | 0.48 | 0.18 | 0.30 |
| S. Grad-CAM++ | 0.44 | 0.17 | 0.27 | 0.74 | 0.30 | 0.45 | 0.36 | 0.28 | 0.09 |
| Int. Grad-CAM | 0.61 | 0.09 | 0.52 | 0.86 | 0.21 | 0.65 | 0.51 | 0.17 | 0.34 |
| HiRes-CAM | 0.57 | 0.10 | 0.47 | 0.86 | 0.21 | 0.65 | 0.49 | 0.18 | 0.32 |
| XGrad-CAM | 0.62 | 0.09 | 0.53 | 0.86 | 0.2097 | 0.65 | 0.51 | 0.16 | 0.35 |
| LayerCAM | 0.57 | 0.10 | 0.47 | 0.83 | 0.22 | 0.61 | 0.47 | 0.19 | 0.28 |
| Score-CAM | 0.56 | 0.11 | 0.46 | 0.83 | 0.23 | 0.60 | 0.51 | 0.1522 | 0.3554 |
| Ablation-CAM | 0.57 | 0.10 | 0.48 | 0.85 | 0.21 | 0.64 | 0.50 | 0.17 | 0.33 |
| Expected Grad-CAM | **0.65** | **0.09** | **0.56** | **0.87** | **0.2093** | **0.66** | **0.52** | 0.1569 | **0.3556** |

*robustness* has been evaluated according to the *Local Lipschitz Estimate* Alvarez-Melis & Jaakkola (2018a), *Max-Sensitivity*, *Avg-Sensitivity* Yeh et al. (2019), *Relative Input Stability (RIS)*, Relative Output Stability (ROS) Agarwal et al. (2022). The complexity characteristic has been measured according to the *Sparseness* Chalasani et al. (2018) and *Complexity* criteria Bhatt et al. (2020). *Insertion* and *deletion* metrics have been computed using the IROF library Rieger & Hansen (2020), while the other metrics using the *Quantus* framework v0.4.4. Hedström et al. (2022). **Notably.** *F.E* has been adopted for a more fair comparison as it is known to exhibit rank-order conflicts Rong et al. (2022); Hedström et al. (2023) with similar metrics (*e.g.*, *P.F*). Due to space constraints we have attached the extended results below. The attribution baseline methods Grad-CAM, Grad-CAM++, Smooth Grad-CAM++, XGrad-CAM, Layer-CAM, Score-CAM, for Integrated Grad-CAM the code from the official repository has been adopted.

In Table 5 are shown the extended *faithfulness* results across the three benchmarking models, while in Table 6 are presented the findings of the localization metrics. In Figure 8 is shown an example of a generated binary segmentation masks. As we employed a binary mask, the results of RM-A Arras et al. (2020) are comparable to A.L Kohlbrenner et al. (2019) which we propose in table 7. The relative robustness (RIS/ROS) results are tabulated in table 8. Ultimately, the infidelity aspect has also been additionally verified on the CIFAR-10 and its results showed in table 9.

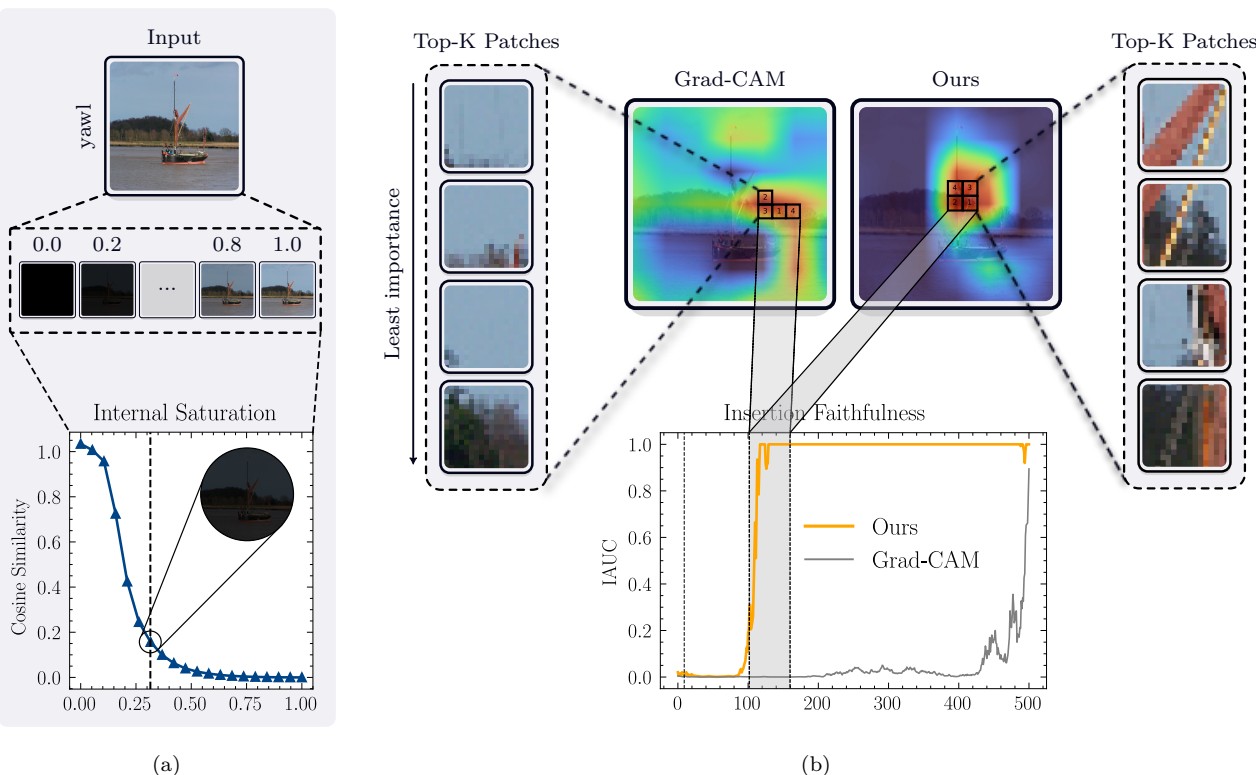

(c) Comparison of the attribution maps under internal saturation conditions. In Figure 2a is shown the cosine similarity of the target layer's embeddings with respect to the interpolator parameter ($\alpha$). Figure 2b shows the attribution maps of the different methods under the saturation condition. The internal saturation condition causes the baseline method to under-represent feature importances across saturating ranges. By extracting the top-4 most important features (fig. 2b) we can observe that the baseline method fails to capture the relevant discriminative regions, which produce low insertion AUCs (fig. 2b) as deemed not important by the model.

Table 6: Localization Metrics: scores computed on 500 samples on the MS-COCO (Lin et al., 2014) dataset on VGG16 (Simonyan & Zisserman, 2014), ResNet-50 (He et al., 2015) and AlexNet (Krizhevsky et al., 2012). Computed on labels "zebra" and "stop sign". **Boldface values indicate best scores**.

| Method | VGG16 | | | ResNet-50 | | | AlexNet | | |
| --- | --- | --- | --- | --- | --- | --- | --- | --- | --- |
| | A.L. | T-K.I. | RR-A | A.L. | T-K.I. | RR-A | A.L. | T-K.I. | RR-A |
| Grad-CAM | 0.11 | 0.24 | 0.24 | 0.09 | 0.11 | 0.12 | 0.09 | 0.07 | 0.1 |
| Grad-CAM++ | 0.13 | 0.30 | 0.29 | **0.106** | 0.11 | 0.128 | 0.08 | 0.03 | 0.07 |
| Smooth Grad-CAM++ | 0.10 | 0.18 | 0.19 | 0.07 | 0.11 | 0.12 | 0.08 | 0.03 | 0.06 |
| Integrated Grad-CAM | 0.12 | 0.34 | 0.31 | 0.097 | 0.119 | 0.13 | 0.08 | 0.07 | 0.1 |
| HiRes-CAM | 0.11 | 0.22 | 0.23 | 0.097 | 0.11 | 0.12 | 0.08 | 0.04 | 0.08 |
| XGrad-CAM | 0.11 | 0.24 | 0.24 | 0.09 | 0.11 | 0.12 | 0.08 | 0.05 | 0.08 |
| LayerCAM | 0.11 | 0.25 | 0.24 | 0.08 | 0.1 | 0.11 | 0.07 | 0.02 | 0.06 |
| Score-CAM | 0.12 | 0.25 | 0.23 | 0.09 | 0.118 | 0.132 | 0.109 | 0.17 | 0.15 |
| Ablation-CAM | 0.15 | 0.36 | 0.33 | 0.09 | 0.11 | 0.12 | 0.106 | 0.15 | 0.14 |
| Expected Grad-CAM | **0.18** | **0.42** | **0.36** | 0.104 | **0.18** | **0.17** | **0.13** | **0.23** | **0.18** |

## C.1 Internal Saturation

Following Sundararajan et al. (2016) we evaluated the saturation at various points on modernly pretrained VGG-16 Simonyan & Zisserman (2014), ResNet-50 He et al. (2015) and AlexNet Krizhevsky et al. (2012). In

Table 7: Localization Metrics: Rank Mass Accuracy (Arras et al., 2020) computed on 500 samples on the MS-COCO (Lin et al., 2014) dataset on VGG16 (Simonyan & Zisserman, 2014), ResNet-50 (He et al., 2015) and AlexNet (Krizhevsky et al., 2012). Computed on labels "zebra" and "stop sign". **Boldface values indicate best scores**.

| Method | VGG16 ↑ RM-A | ResNet-50 ↑ RM-A | AlexNet ↑ RM-A |
|---|---|---|---|
| G-CAM | 0.11 | 0.09 | 0.09 |
| G-CAM++ | 0.13 | **0.11** | 0.08 |
| Sm. G-CAM++ | 0.10 | 0.07 | 0.08 |
| Int. G-CAM | 0.12 | 0.10 | 0.08 |
| HiRes-CAM | 0.11 | 0.10 | 0.08 |
| XG-CAM | 0.11 | 0.09 | 0.08 |
| LayerCAM | 0.11 | 0.08 | 0.07 |
| Score-CAM | 0.12 | 0.09 | 0.11 |
| Ablation-CAM | 0.15 | 0.09 | 0.11 |
| Exp. G-CAM | **0.18** | 0.11 | **0.13** |

Table 8: Robustness Metrics: RIS/ROS (Agarwal et al., 2022) computed on 500 samples on the *ILSVRC2012* (Russakovsky et al., 2014) dataset on VGG-16 (Simonyan & Zisserman, 2014) and ResNet-50 (He et al., 2015). Methods marked with a "−" have been excluded due to zero-attribution values under infinitesimal perturbations. **Boldface values indicate best scores**.

| Method | VGG-16 ↓ RIS | ↓ ROS | ResNet-50 ↓ RIS | ↓ ROS |
|---|---|---|---|---|
| G-CAM | 169.197 | 5527.376 | 103.162 | 1.55e+04 |
| G-CAM++ | 0.045 | 1.3 | 357.893 | 3130.042 |
| Sm. G-CAM++ | 25.003 | 2.704 | 59.733 | 1180.478 |
| Int G-CAM | - | - | - | - |
| Hi-Res CAM | - | - | - | - |
| XG-CAM | 33.872 | 2812.874 | 111.022 | 1.65e+04 |
| LayerCAM | 0.023 | 33.782 | 11.712 | 555.22 |
| Score-CAM | 0.09 | 14.97 | 19.046 | 2053.248 |
| Ablation-CAM | - | - | - | - |
| Exp. G-CAM | **0.004** | **0.12** | **0.573** | **73.934** |

Figure 10a are shown the 25 random samples utilized, alongside a selected excerpt of the samples generated using the following feature scaling procedure (fig. 10b) for $N = 25$:

$$\{\alpha_i \mid \alpha_i \sim U(0,1),\ i = 1, 2, \ldots, N\}$$

Figures 11c and 12c shows the saturating behavior *w.r.t.*, the output and intermediary layers targeted by CAM methods. Both the *pre-softmax* and *post-softmax* outputs quickly flatten and plateaus for very small value of the feature scaling factor $\alpha$, with the softmax outputs showing the swiftest rate of change and abruptly converge to saturation (fig. 12c). When selecting an arbitrary intermediary layer (*i.e.*, the one targeted by the analyzed CAM methods) the saturation phenomena is still present but offset due to the reduced path (depth) (fig. 9c). As $\alpha$ increases, the cosine similarity of the target layer's embeddings quickly flattens (Figure 9a), leading to an underestimation of feature attributions. This results in sparse, uninformative, and ill-formed explanations (Figure 9b). This is evident when inspecting the top-k most important patches according to the generated attribution maps, which focus on background areas rather than the target class (*yawl*). Consequently, when these patches are inserted, they produce low model confidence (Insertion IAUC) (Figure 9b). Conversely, our method focuses on salient discriminative areas of the image that characterize the target label (*i.e.*, *yawl*) and highly activate the neural network, demonstrating high fidelity to the model's inner workings, robustness to internal saturation, and high localization by focusing only on the most important regions.

Table 9: Infidelity (Yeh et al., 2019) on 500 CIFAR10 (Ho-Phuoc, 2018) samples using VGG-16 (Simonyan & Zisserman, 2014), ResNet-50 (He et al., 2015), and AlexNet (Krizhevsky et al., 2012). Samples upsampled to $96 \times 96$; patch size 32. Due to the low sample resolution, absolute values are high; for readability, values are scaled by $10^7$, $10^8$, $10^9$ respectively.

| | VGG16 | ResNet-50 | AlexNet |
| Method | ↓ Inf. | ↓ Inf. | ↓ Inf. |
|---|---|---|---|
| Grad-CAM | 1592.0 | 94.2 | 594.6 |
| Grad-CAM++ | 1506.5 | 88.9 | 542.0 |
| Smooth Grad-CAM++ | 1673.5 | 82.9 | 479.0 |
| Integrated Grad-CAM | 1640.0 | 37.9 | 483.0 |
| Hi-Res CAM | 1585.6 | 77.6 | 594.1 |
| XGrad-CAM | 1549.2 | 93.5 | 575.0 |
| LayerCAM | 1457.0 | 92.5 | 555.1 |
| Score-CAM | 1751.7 | 157.3 | 656.6 |
| Ablation-CAM | 1670.2 | 76.5 | 619.7 |
| Expected Grad-CAM | **4.7** | **3.8** | **9.6** |

Table 10: Running time (seconds) on 100 sequential runs using VGG-16 (Simonyan & Zisserman, 2014) on CIFAR10 (Ho-Phuoc, 2018). Averaged values displayed. Methods evaluated under identical hardware conditions.

| Method | ↓ R.T. |
|---|---|
| Grad-CAM | **0.006** |
| Grad-CAM++ | **0.006** |
| Smooth Grad-CAM++ | 0.121 |
| Integrated Grad-CAM | 0.156 |
| Hi-Res CAM | **0.006** |
| XGrad-CAM | **0.006** |
| LayerCAM | **0.006** |
| Score-CAM | 0.261 |
| Ablation-CAM | 0.302 |
| Expected Grad-CAM | 0.115 |

Samples

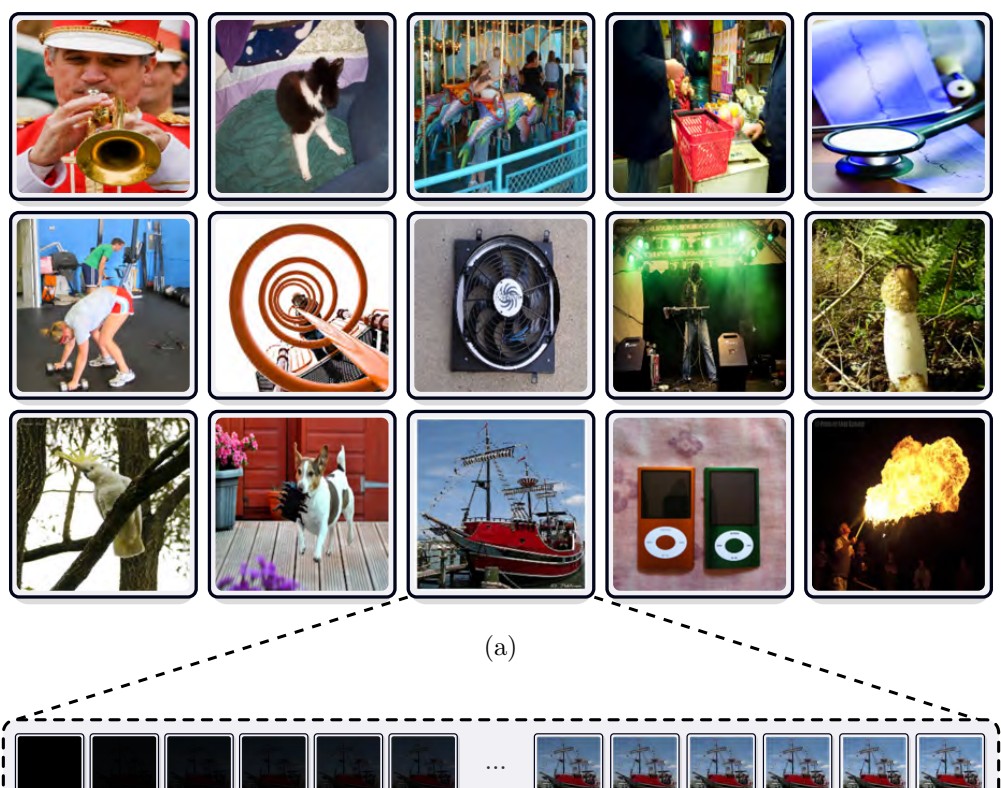

(a)

(b)

(c) Excerpt of 25 random samples from *ILSVRC2012* Russakovsky et al. (2014) (10a) used to evaluate internal saturation at various points. Figure 10b presents a subset of samples generated through feature scaling over 25 steps.

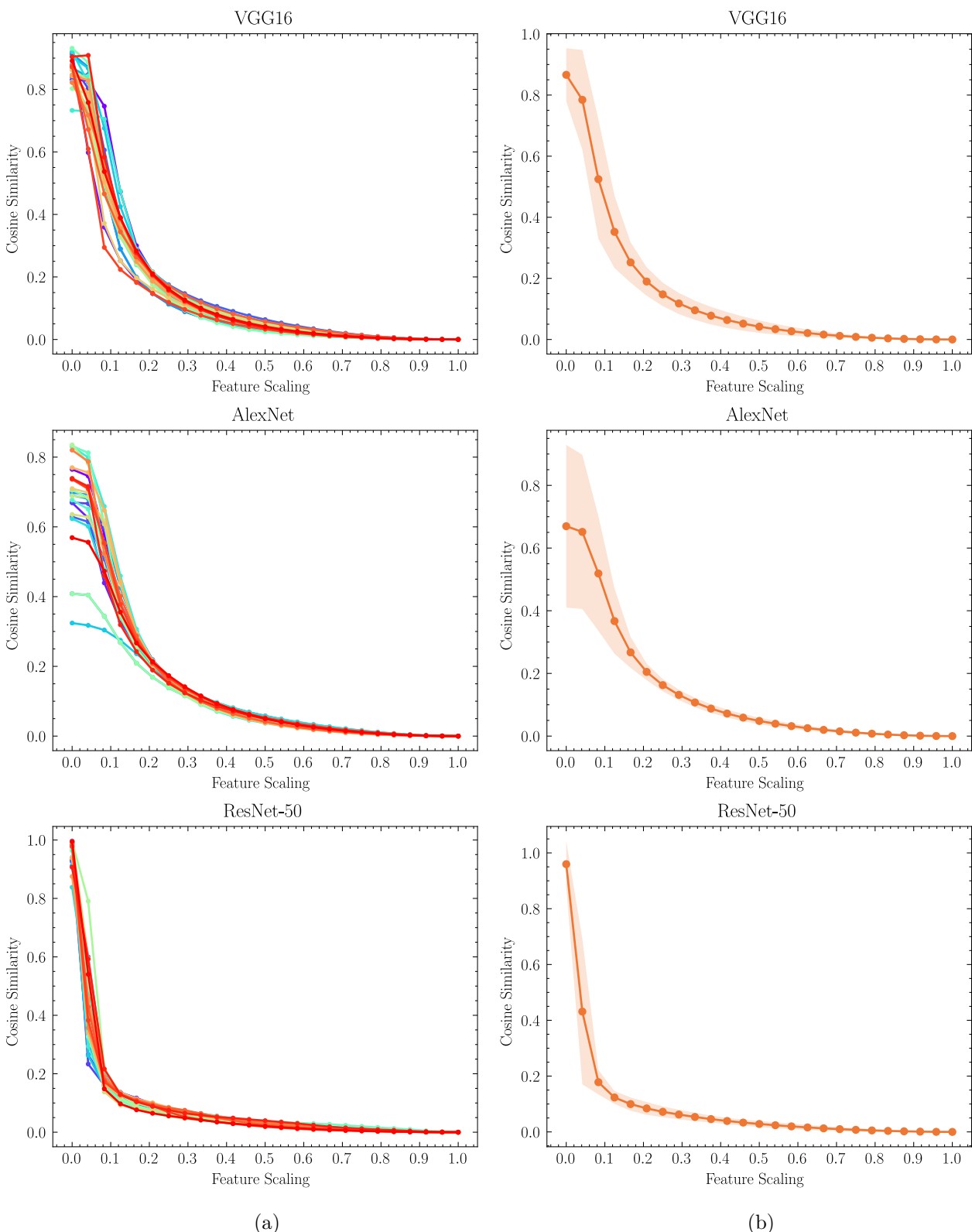

(a)                                                             (b)

(c) Internal saturation analysis of intermediary target layers in VGG16 Simonyan & Zisserman (2014), AlexNet Krizhevsky et al. (2012), and ResNet-50 He et al. (2015). Figure 11a presents the cosine similarity between activation vectors of CAM target filters. Figure 11b depicts the mean values with error bars indicating 2 standard deviations. For VGG16 and AlexNet, the final feature layer is used, while for ResNet-50, the *layer*4 is selected.

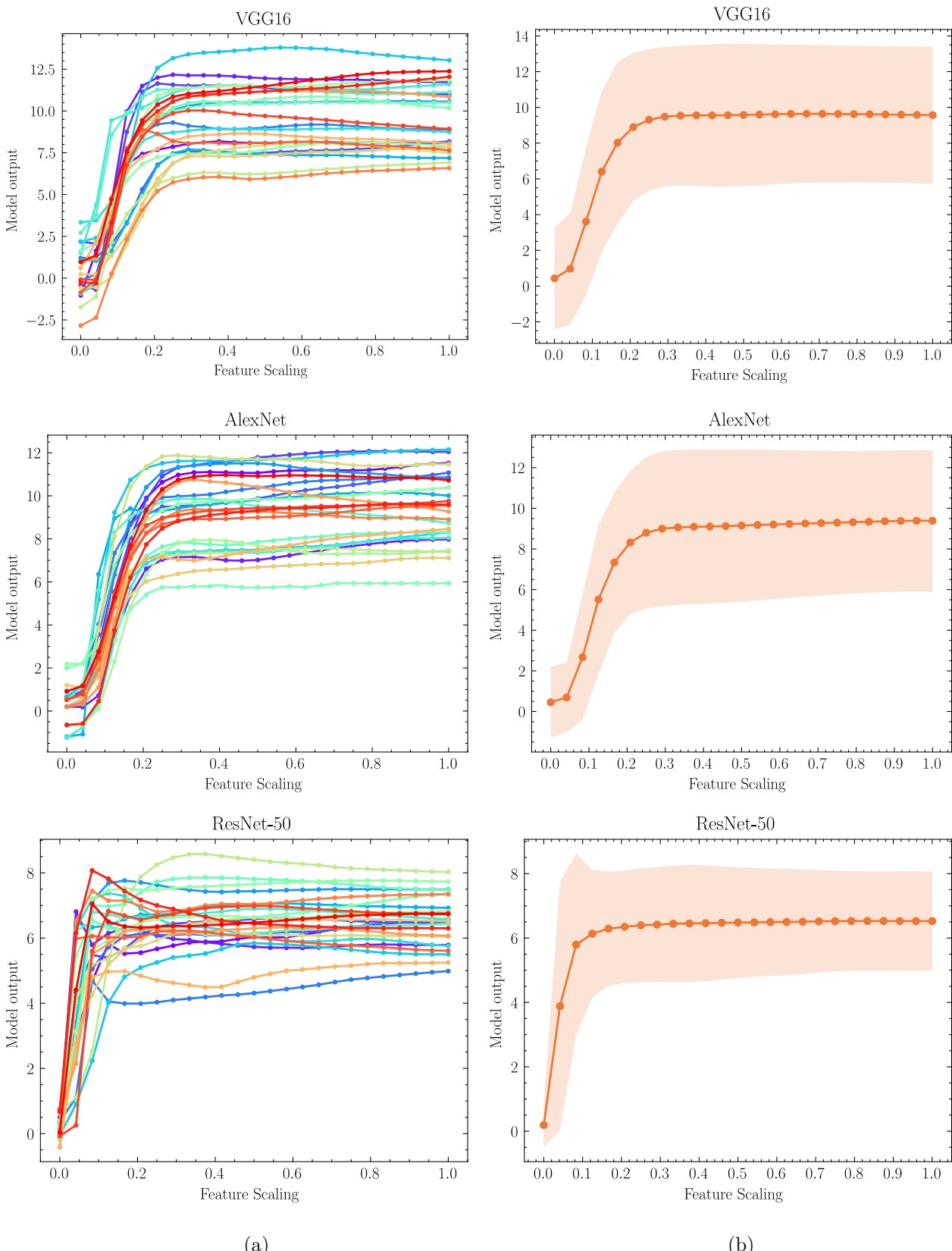

(a)

(b)

(c) Output saturation analysis in VGG16 Simonyan & Zisserman (2014), AlexNet Krizhevsky et al. (2012), and ResNet-50 He et al. (2015) (fig. 12a). Figure 12a displays the softmax scores for the top label, while Figure 12b depicts the mean values with error bars indicating 2 standard deviations.

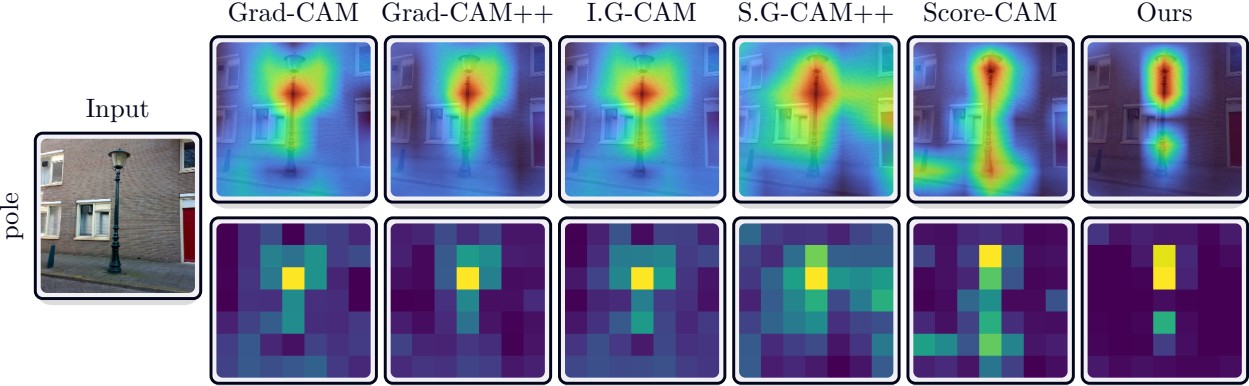

Figure 13: Gradients are noisy. A comparison of gradient-based CAM methods under optimal conditions shows that even recent methods exhibit high sensitivity.

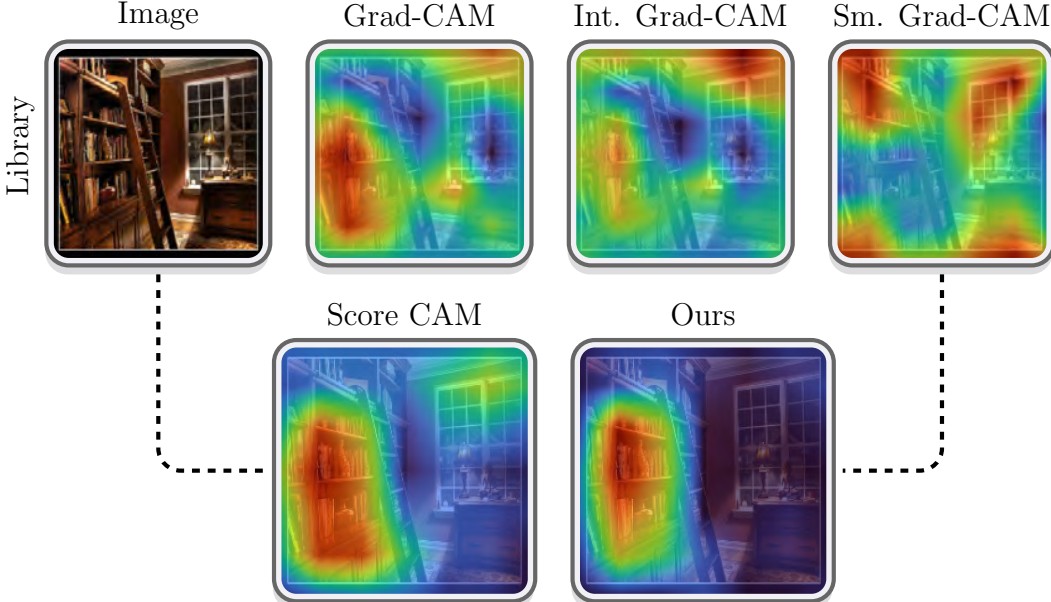

Figure 14: Comparison of gradient-based and non-gradient-based CAM methods. The top row illustrates the noisiness and tendency to produce ill-formed explanations in gradient-based methods, including recent approaches Sattarzadeh et al. (2021). *Score-CAM* Wang et al. (2019) addresses this issue by eliminating the use of gradients. Our method demonstrates the ability to generate sharper and more stable explanations consistently, even with the use of gradients.

## D    Qualitative Evaluation

Next we provide the extended version of all the figures *i.e.*, including all the comparative baseline methods and some additional examples.

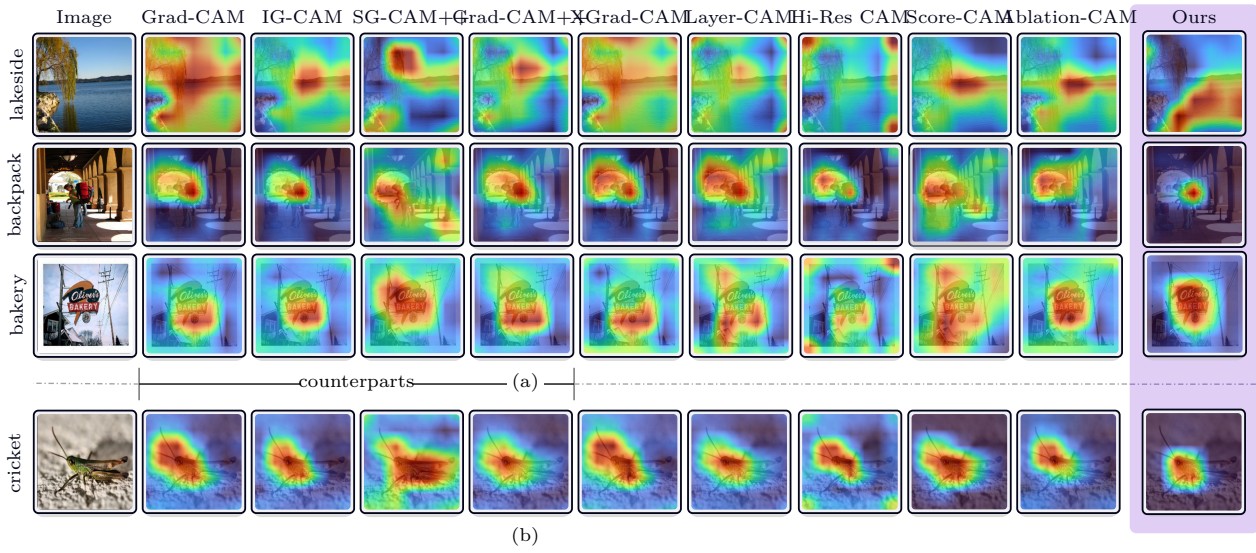

(c) Comparison of the attribution maps for various methods under normal (15b) and internal saturation condition (15a). Extended version containing all baseline attribution methods.

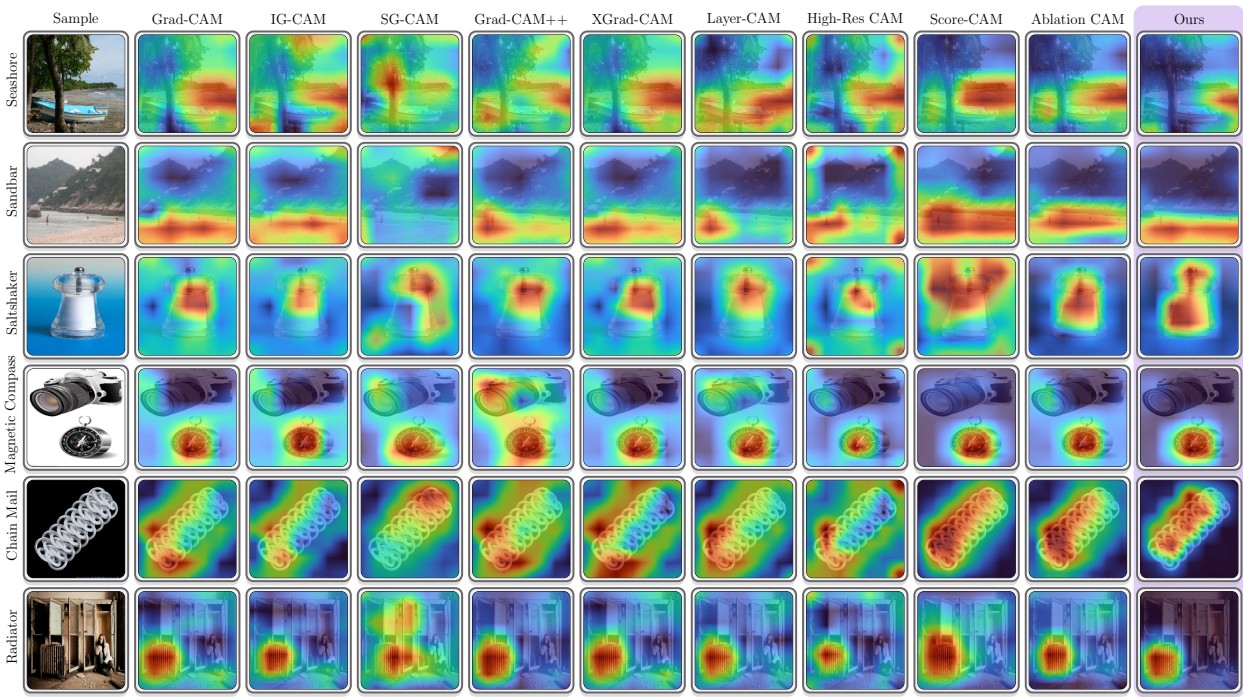

Figure 16: Comparison of saliency maps between our method and all the baseline methods on the *ILSVRC2012* Russakovsky et al. (2014).

# E  Fine-Grained Recognition and Architecture Generalization

To assess the generality of our approach beyond the ImageNet/VGG-16 setting of the main paper, we conduct an extensive evaluation spanning **5 fine-grained recognition datasets**, **8 architectures**, and **10 CAM methods**. This yields **40 dataset–architecture combinations** with $N$=500 images each, totaling **20,000**

**evaluation instances**. The fine-grained setting is particularly demanding: discriminative regions are small and local (*e.g.*, beak shape, wing pattern), making localization quality a critical differentiator.

### E.1  Experimental Setup

**Datasets.** We evaluate on five standard fine-grained recognition benchmarks: CUB-200-2011 Wah et al. (2025) (200 bird species with part annotations), Stanford Dogs Khosla et al. (2012) (120 dog breeds), Oxford-IIIT Pet Parkhi et al. (2012) (37 cat/dog breeds with segmentation masks), Flowers-102 Nilsback & Zisserman (2008) (102 flower categories), and FGVC Aircraft Maji et al. (2013) (100 aircraft variants). These datasets span diverse visual domains and object scales. All five provide spatial ground-truth annotations for localization evaluation: CUB-200, Oxford Pet, and Flowers-102 include pixel-level segmentation masks, while Stanford Dogs and FGVC Aircraft provide bounding boxes.

**Architectures.** Table 11 summarizes the eight architectures evaluated, spanning six standard convolutional designs and two architectures with non-standard feature extraction (ConvNeXt-Tiny and ViT-B/16). The feature dimension $K$ of the CAM target layer ranges from 512 to 2,048, directly determining the perturbation budget $M=2K$.

Table 11: Architectures evaluated in the fine-grained experiments, grouped by design family. $K$ denotes the feature dimension of the CAM target layer. The perturbation budget is $M=2K$ for all architectures (Table 2). All models use ImageNet-pretrained weights.

| | Perturbation Budget | |
|---|---|---|
| **Architecture** | $K$ | $M=2K$ |
| *Standard convolutional* | | |
| VGG-16 Simonyan & Zisserman (2014) | 512 | 1,024 |
| ResNet-18 He et al. (2015) | 512 | 1,024 |
| ResNet-50 He et al. (2015) | 2,048 | 4,096 |
| DenseNet-121 Huang et al. (2016) | 1,024 | 2,048 |
| EfficientNet-B0 Tan & Le (2019) | 1,280 | 2,560 |
| Inception V3 Szegedy et al. (2015) | 2,048 | 4,096 |
| *Non-standard feature extraction* | | |
| ConvNeXt-Tiny Liu et al. (2022) | 768 | 1,536 |
| ViT-B/16 Dosovitskiy et al. (2020) | 768 | 1,536 |

**Methods and evaluation.** We evaluate all 10 methods and hyperparameter configurations from the main paper (Table 2) across seven Quantus Hedström et al. (2022) metrics as defined in Table 4.

### E.2  Aggregate Results Across Architectures and Datasets

Table 12(a) presents the metrics averaged across all 40 combinations. Expected Grad-CAM achieves the best Attribution Localization (0.700 vs. 0.656 for the best baseline S. Grad-CAM++, a +6.7% improvement), the best RMA (0.717), the best Sparseness (0.693, a 37% improvement over the second-best Score-CAM, 0.506), and dramatically lower Effective Complexity (30.4k vs. ∼50k for most baselines, a ∼41% reduction). These gains indicate that the infidelity-minimization framework concentrates attributions on genuinely discriminative regions while producing sharper, more interpretable saliency maps.

S. Grad-CAM++ leads on RRA (0.706) and AUC (0.778). This is expected: our method optimizes for explanation faithfulness (infidelity) rather than rank-ordering of pixel importance. Higher sparseness naturally redistributes ranks among secondary features. Importantly, unlike S. Grad-CAM++, our method satisfies the completeness axiom, providing a theoretical guarantee of faithfulness.

Table 12(b) confirms these findings through average rankings. Expected Grad-CAM achieves the best average rank of 3.2 out of 10 methods, winning 5 of 7 individual metric rankings outright (Attr.Loc., RMA, Sparse., Complex., Eff.Comp.). No other method achieves an average rank below 5.4.

Table 12: Aggregate evaluation across 5 fine-grained recognition datasets and 8 architectures (40 combinations, $N$=500 images each). *Panel (a)*: metric values averaged across all combinations; *Panel (b)*: average rank across combinations (lower is better). Best in **bold**, second-best underlined.

| Method | Localization | | | | Complexity | | | Avg. Rank |
|---|---|---|---|---|---|---|---|---|
| | Attr.Loc.↑ | RRA↑ | RMA↑ | AUC↑ | Sparse.↑ | Complex.↓ | Eff.Comp.↓ | |
| *(a) Metric values (averaged across 40 combinations)* | | | | | | | | |
| Grad-CAM | 0.646 | 0.689 | 0.646 | 0.755 | 0.458 | 10.47 | 51.7k | |
| Grad-CAM++ | 0.651 | 0.703 | 0.654 | 0.773 | 0.457 | 10.44 | 50.4k | |
| S. Grad-CAM++ | 0.656 | **0.706** | 0.656 | **0.778** | 0.459 | 10.44 | 51.2k | |
| Int. Grad-CAM | 0.651 | 0.696 | 0.651 | 0.766 | 0.453 | 10.49 | 52.5k | |
| HiResCAM | 0.642 | 0.679 | 0.642 | 0.741 | 0.471 | 10.45 | 51.7k | |
| XGrad-CAM | 0.638 | 0.672 | 0.638 | 0.730 | 0.470 | 10.45 | 51.1k | |
| LayerCAM | 0.651 | 0.698 | 0.651 | 0.768 | 0.458 | 10.47 | 51.5k | |
| Score-CAM | 0.643 | 0.669 | 0.643 | 0.728 | 0.506 | 10.36 | 49.7k | |
| Ablation-CAM | 0.617 | 0.681 | 0.632 | 0.748 | 0.462 | 10.35 | 49.5k | |
| **Expected Grad-CAM** | **0.700** | 0.686 | **0.717** | 0.718 | **0.693** | **9.74** | **30.4k** | |
| *(b) Average rank across 40 combinations (lower is better)* | | | | | | | | |
| Grad-CAM | 5.2 | 5.4 | 5.2 | 6.5 | 5.8 | 5.4 | 4.4 | 5.4 |
| Grad-CAM++ | 6.5 | 4.3 | 6.4 | 4.7 | 7.2 | 7.4 | 7.4 | 6.3 |
| S. Grad-CAM++ | 5.7 | **3.7** | 5.7 | **3.6** | 7.2 | 7.7 | 8.3 | 6.0 |
| Int. Grad-CAM | 5.5 | 5.5 | 5.5 | 5.5 | 6.6 | 6.4 | 6.0 | 5.9 |
| HiResCAM | 6.4 | 7.2 | 6.4 | 7.0 | 5.3 | 5.6 | 5.7 | 6.2 |
| XGrad-CAM | 7.6 | 8.3 | 7.6 | 6.9 | 5.4 | 6.3 | 6.2 | 6.9 |
| LayerCAM | 6.5 | 4.8 | 6.6 | 5.1 | 7.7 | 8.0 | 8.3 | 6.7 |
| Score-CAM | 7.4 | 7.9 | 7.5 | 8.0 | 5.8 | 6.1 | 6.5 | 7.0 |
| Ablation-CAM | 6.0 | 5.3 | 6.1 | 4.7 | 7.0 | 6.3 | 6.2 | 5.9 |
| **Expected Grad-CAM** | **2.2** | 6.6 | **2.1** | 7.9 | **1.5** | **1.2** | **1.2** | **3.2** |

## E.3 Per-Architecture Analysis

We now examine how performance varies across architectures, grouping results by architectural family.

**VGG-16** (Table 13). VGG-16 is the architecture used in the main paper. On fine-grained datasets, Expected Grad-CAM dominates with Attr.Loc. 0.836, RMA 0.836, Sparseness 0.809, and Eff.Comp. 21.3k, all substantially ahead of all baselines. The large receptive fields of VGG-16 provide spatially rich feature maps that our method exploits effectively.

Table 13: Quantus metrics for VGG-16 Simonyan & Zisserman (2014) ($K$=512, $M$=2$K$=1024), averaged across 5 fine-grained datasets ($N$=500 per combination).

| Method | Attr.Loc.↑ | RRA↑ | RMA↑ | AUC↑ | Sparse.↑ | Complex.↓ | Eff.Comp.↓ |
|---|---|---|---|---|---|---|---|
| Grad-CAM | 0.620 | 0.652 | 0.620 | 0.691 | 0.448 | 10.46 | 49.0k |
| Grad-CAM++ | 0.646 | 0.722 | 0.646 | 0.783 | 0.411 | 10.53 | 50.2k |
| S. Grad-CAM++ | 0.645 | 0.720 | 0.645 | 0.780 | 0.412 | 10.53 | 50.2k |
| Int. Grad-CAM | 0.666 | 0.699 | 0.666 | 0.757 | 0.477 | 10.40 | 49.2k |
| HiResCAM | 0.581 | 0.582 | 0.581 | 0.583 | 0.542 | 10.29 | 49.0k |
| XGrad-CAM | 0.589 | 0.619 | 0.589 | 0.633 | 0.438 | 10.48 | 49.3k |
| LayerCAM | 0.630 | 0.672 | 0.630 | 0.715 | 0.464 | 10.44 | 50.2k |
| Score-CAM | 0.766 | 0.742 | 0.766 | **0.835** | 0.631 | 10.02 | 48.2k |
| Ablation-CAM | 0.683 | 0.701 | 0.683 | 0.770 | 0.505 | 10.35 | 49.3k |
| **Expected Grad-CAM** | **0.836** | **0.754** | **0.836** | 0.768 | **0.809** | **9.31** | **21.3k** |

**ResNet-18 and ResNet-50** (Tables 14 and 15). Expected Grad-CAM leads on Attr.Loc. (0.728 and 0.687), RMA, Sparseness, Complexity, and Eff.Comp. for both ResNet variants. The performance gap between

ResNet-18 and ResNet-50 is modest, with ResNet-18 slightly stronger; we attribute this to its smaller feature dimension ($K=512$ vs. $K=2048$), which makes the second moment matrix $\mathcal{M}_I$ better-conditioned with $M=2K$ samples.

Table 14: Quantus metrics for ResNet-18 He et al. (2015) ($K=512$, $M=2K=1024$), averaged across 5 fine-grained datasets ($N=500$ per combination).

| Method | Attr.Loc.↑ | RRA↑ | RMA↑ | AUC↑ | Sparse.↑ | Complex.↓ | Eff.Comp.↓ |
|---|---|---|---|---|---|---|---|
| Grad-CAM | 0.643 | 0.744 | 0.643 | 0.844 | 0.372 | 10.58 | 49.8k |
| Grad-CAM++ | 0.637 | 0.761 | 0.637 | 0.862 | 0.348 | 10.61 | 50.0k |
| S. Grad-CAM++ | 0.639 | **0.763** | 0.639 | **0.864** | 0.351 | 10.60 | 50.0k |
| Int. Grad-CAM | 0.643 | 0.744 | 0.643 | 0.844 | 0.372 | 10.58 | 49.8k |
| HiResCAM | 0.643 | 0.744 | 0.643 | 0.844 | 0.372 | 10.58 | 49.8k |
| XGrad-CAM | 0.643 | 0.744 | 0.643 | 0.844 | 0.372 | 10.58 | 49.8k |
| LayerCAM | 0.637 | 0.761 | 0.637 | 0.862 | 0.348 | 10.61 | 50.0k |
| Score-CAM | 0.629 | 0.734 | 0.629 | 0.835 | 0.347 | 10.61 | 50.0k |
| Ablation-CAM | 0.636 | 0.748 | 0.636 | 0.850 | 0.351 | 10.60 | 50.0k |
| **Expected Grad-CAM** | **0.728** | 0.711 | **0.728** | 0.778 | **0.616** | **10.05** | **36.0k** |

Table 15: Quantus metrics for ResNet-50 He et al. (2015) ($K=2048$, $M=2K=4096$), averaged across 5 fine-grained datasets ($N=500$ per combination).

| Method | Attr.Loc.↑ | RRA↑ | RMA↑ | AUC↑ | Sparse.↑ | Complex.↓ | Eff.Comp.↓ |
|---|---|---|---|---|---|---|---|
| Grad-CAM | 0.637 | 0.716 | 0.637 | 0.814 | 0.380 | 10.56 | 49.7k |
| Grad-CAM++ | 0.627 | 0.721 | 0.627 | 0.818 | 0.357 | 10.59 | 50.1k |
| S. Grad-CAM++ | 0.631 | **0.726** | 0.631 | **0.827** | 0.358 | 10.59 | 50.1k |
| Int. Grad-CAM | 0.637 | 0.716 | 0.637 | 0.814 | 0.380 | 10.56 | 49.7k |
| HiResCAM | 0.637 | 0.716 | 0.637 | 0.814 | 0.380 | 10.56 | 49.7k |
| XGrad-CAM | 0.637 | 0.716 | 0.637 | 0.814 | 0.380 | 10.56 | 49.7k |
| LayerCAM | 0.627 | 0.720 | 0.627 | 0.817 | 0.356 | 10.59 | 50.1k |
| Score-CAM | 0.612 | 0.684 | 0.612 | 0.778 | 0.351 | 10.60 | 50.1k |
| Ablation-CAM | 0.625 | 0.712 | 0.625 | 0.810 | 0.351 | 10.60 | 50.0k |
| **Expected Grad-CAM** | **0.687** | 0.676 | **0.687** | 0.735 | **0.611** | **10.06** | **35.8k** |

**DenseNet-121** (Table 16). DenseNet's dense connectivity produces feature maps with high channel correlation. Expected Grad-CAM achieves the best Attr.Loc. (0.735), Sparseness (0.660), and Eff.Comp. (33.3k). The XGrad-CAM baseline is notably strong on Eff.Comp. (44.6k) due to DenseNet's skip connections affecting gradient flow.

**EfficientNet-B0** (Table 17). On EfficientNet-B0, Expected Grad-CAM leads Attr.Loc. (0.814), Sparseness (0.743), and Eff.Comp. (25.2k). EfficientNet's compound scaling produces highly efficient feature representations, and the particularly low Effective Complexity suggests that our method successfully identifies compact discriminative regions in this architecture.

**Inception V3** (Table 18). Inception V3 has the highest feature dimension ($K=2048$) and employs multi-scale convolutions. Expected Grad-CAM achieves Attr.Loc. 0.824, alongside the best RRA (0.768), RMA (0.824), Sparseness (0.747), and Eff.Comp. (45.2k). The multi-scale nature of Inception features appears to complement our data-aware perturbation strategy well.

**ConvNeXt-Tiny** (Table 19). ConvNeXt uses depthwise separable convolutions with a design inspired by Vision Transformers. All CAM methods show degraded performance relative to purely convolutional architectures, particularly the gradient-based variants. S. Grad-CAM++ achieves the best Attr.Loc. (0.756) and Sparseness (0.748), while Expected Grad-CAM maintains the best Complexity (9.27) and Eff.Comp. (19.6k). The degradation reflects a known limitation: CAM methods assume spatial feature maps from

Table 16: Quantus metrics for DenseNet-121 Huang et al. (2016) ($K$=1024, $M$=2$K$=2048), averaged across 5 fine-grained datasets ($N$=500 per combination).

| Method | Attr.Loc.↑ | RRA↑ | RMA↑ | AUC↑ | Sparse.↑ | Complex.↓ | Eff.Comp.↓ |
|---|---|---|---|---|---|---|---|
| Grad-CAM | 0.646 | 0.741 | 0.646 | 0.844 | 0.387 | 10.56 | 49.8k |
| Grad-CAM++ | 0.643 | 0.751 | 0.643 | 0.856 | 0.370 | 10.58 | 49.9k |
| S. Grad-CAM++ | 0.645 | **0.754** | 0.645 | **0.860** | 0.370 | 10.58 | 49.9k |
| Int. Grad-CAM | 0.646 | 0.740 | 0.646 | 0.844 | 0.386 | 10.56 | 49.8k |
| HiResCAM | 0.651 | 0.737 | 0.651 | 0.840 | 0.404 | 10.53 | 49.8k |
| XGrad-CAM | 0.612 | 0.638 | 0.612 | 0.703 | 0.497 | 10.35 | 44.6k |
| LayerCAM | 0.648 | 0.750 | 0.648 | 0.854 | 0.384 | 10.56 | 50.0k |
| Score-CAM | 0.590 | 0.624 | 0.590 | 0.672 | 0.481 | 10.39 | 45.0k |
| Ablation-CAM | 0.646 | 0.740 | 0.646 | 0.844 | 0.386 | 10.56 | 49.8k |
| **Expected Grad-CAM** | **0.735** | 0.721 | **0.735** | 0.772 | **0.660** | **9.93** | **33.3k** |

Table 17: Quantus metrics for EfficientNet-B0 Tan & Le (2019) ($K$=1280, $M$=2$K$=2560), averaged across 5 fine-grained datasets ($N$=500 per combination).

| Method | Attr.Loc.↑ | RRA↑ | RMA↑ | AUC↑ | Sparse.↑ | Complex.↓ | Eff.Comp.↓ |
|---|---|---|---|---|---|---|---|
| Grad-CAM | 0.714 | 0.768 | 0.714 | 0.846 | 0.503 | 10.36 | 49.9k |
| Grad-CAM++ | 0.759 | 0.773 | 0.759 | 0.850 | 0.598 | 10.13 | 39.3k |
| S. Grad-CAM++ | 0.762 | 0.776 | 0.762 | 0.855 | 0.602 | 10.12 | 40.7k |
| Int. Grad-CAM | 0.714 | 0.768 | 0.714 | 0.846 | 0.503 | 10.36 | 49.9k |
| HiResCAM | 0.714 | 0.768 | 0.714 | 0.846 | 0.503 | 10.36 | 49.9k |
| XGrad-CAM | 0.714 | 0.768 | 0.714 | 0.846 | 0.503 | 10.36 | 49.9k |
| LayerCAM | 0.758 | 0.774 | 0.758 | 0.851 | 0.596 | 10.13 | 39.5k |
| Score-CAM | 0.776 | 0.731 | 0.776 | 0.801 | 0.725 | 9.73 | 29.0k |
| Ablation-CAM | 0.754 | **0.777** | 0.754 | **0.864** | 0.566 | 10.20 | 42.8k |
| **Expected Grad-CAM** | **0.814** | 0.756 | **0.814** | 0.792 | **0.743** | **9.62** | **25.2k** |

Table 18: Quantus metrics for Inception V3 Szegedy et al. (2015) ($K$=2048, $M$=2$K$=4096), averaged across 5 fine-grained datasets ($N$=500 per combination).

| Method | Attr.Loc.↑ | RRA↑ | RMA↑ | AUC↑ | Sparse.↑ | Complex.↓ | Eff.Comp.↓ |
|---|---|---|---|---|---|---|---|
| Grad-CAM | 0.726 | 0.762 | 0.726 | 0.871 | 0.563 | 10.83 | 86.2k |
| Grad-CAM++ | 0.715 | 0.763 | 0.715 | 0.869 | 0.538 | 10.89 | 89.3k |
| S. Grad-CAM++ | 0.716 | 0.764 | 0.716 | 0.870 | 0.538 | 10.89 | 89.3k |
| Int. Grad-CAM | 0.726 | 0.762 | 0.726 | 0.871 | 0.563 | 10.83 | 86.2k |
| HiResCAM | 0.726 | 0.762 | 0.726 | 0.871 | 0.563 | 10.83 | 86.2k |
| XGrad-CAM | 0.726 | 0.762 | 0.726 | 0.871 | 0.563 | 10.83 | 86.2k |
| LayerCAM | 0.715 | 0.763 | 0.715 | 0.869 | 0.538 | 10.89 | 89.3k |
| Score-CAM | 0.706 | 0.758 | 0.706 | 0.859 | 0.526 | 10.91 | 89.3k |
| Ablation-CAM | 0.720 | 0.764 | 0.720 | **0.872** | 0.549 | 10.86 | 89.2k |
| **Expected Grad-CAM** | **0.824** | 0.768 | **0.824** | 0.831 | **0.747** | **10.20** | **45.2k** |

standard convolutions, which is partially violated by depthwise convolutions. Notably, this limitation affects *all* CAM methods, not our approach specifically.

Table 19: Quantus metrics for ConvNeXt-Tiny Liu et al. (2022) ($K$=768, $M$=$2K$=1536), averaged across 5 fine-grained datasets ($N$=500 per combination).

| Method | Attr.Loc.↑ | RRA↑ | RMA↑ | AUC↑ | Sparse.↑ | Complex.↓ | Eff.Comp.↓ |
|---|---|---|---|---|---|---|---|
| Grad-CAM | 0.742 | 0.713 | 0.742 | 0.772 | 0.733 | 9.75 | 30.1k |
| Grad-CAM++ | 0.729 | 0.705 | 0.758 | 0.754 | 0.731 | 9.57 | 25.6k |
| S. Grad-CAM++ | **0.756** | 0.707 | **0.759** | 0.766 | **0.748** | 9.57 | 29.8k |
| Int. Grad-CAM | 0.737 | **0.723** | 0.737 | **0.797** | 0.669 | 9.95 | 35.5k |
| HiResCAM | 0.742 | 0.713 | 0.742 | 0.772 | 0.733 | 9.75 | 30.1k |
| XGrad-CAM | 0.742 | 0.713 | 0.742 | 0.772 | 0.733 | 9.75 | 30.1k |
| LayerCAM | 0.743 | 0.723 | 0.743 | 0.793 | 0.692 | 9.88 | 34.0k |
| Score-CAM | 0.568 | 0.586 | 0.568 | 0.573 | 0.593 | 10.13 | 39.5k |
| Ablation-CAM | 0.349 | 0.482 | 0.470 | 0.440 | 0.484 | 9.43 | 23.5k |
| **Expected Grad-CAM** | 0.472 | 0.577 | 0.609 | 0.566 | 0.641 | **9.27** | **19.6k** |

**ViT-B/16** (Table 20). Vision Transformers fundamentally lack the spatial convolution structure that CAM methods rely on. All methods perform substantially worse than on CNNs, with Ablation-CAM showing the strongest localization (Attr.Loc. 0.526, RRA 0.524). Expected Grad-CAM achieves competitive Attr.Loc. (0.505) and excels on Sparseness (0.714) and Eff.Comp. (26.5k). The uniformly poor performance across all methods confirms that this is an inherent limitation of the CAM paradigm on non-convolutional architectures, not a weakness specific to our framework.

Table 20: Quantus metrics for ViT-B/16 Dosovitskiy et al. (2020) ($K$=768, $M$=$2K$=1536), averaged across 5 fine-grained datasets ($N$=500 per combination).

| Method | Attr.Loc.↑ | RRA↑ | RMA↑ | AUC↑ | Sparse.↑ | Complex.↓ | Eff.Comp.↓ |
|---|---|---|---|---|---|---|---|
| Grad-CAM | 0.443 | 0.413 | 0.443 | 0.357 | 0.275 | 10.68 | 49.4k |
| Grad-CAM++ | 0.450 | 0.428 | 0.450 | 0.390 | 0.300 | 10.65 | 49.2k |
| S. Grad-CAM++ | 0.455 | 0.438 | 0.455 | 0.401 | 0.290 | 10.66 | 49.8k |
| Int. Grad-CAM | 0.443 | 0.413 | 0.443 | 0.357 | 0.275 | 10.68 | 49.4k |
| HiResCAM | 0.443 | 0.413 | 0.443 | 0.357 | 0.275 | 10.68 | 49.4k |
| XGrad-CAM | 0.443 | 0.413 | 0.443 | 0.357 | 0.275 | 10.68 | 49.4k |
| LayerCAM | 0.450 | 0.424 | 0.450 | 0.386 | 0.285 | 10.67 | 49.4k |
| Score-CAM | 0.493 | 0.490 | 0.493 | 0.470 | 0.389 | 10.52 | 46.6k |
| Ablation-CAM | **0.526** | **0.524** | **0.526** | **0.538** | 0.502 | 10.23 | 41.1k |
| **Expected Grad-CAM** | 0.505 | 0.523 | 0.505 | 0.501 | **0.714** | **9.48** | **26.5k** |

### E.4 Per-Dataset Analysis

We now analyze performance across datasets, averaged over all 8 architectures.

**CUB-200-2011** (Table 21). CUB-200 is the most challenging dataset, with fine-grained bird species requiring precise localization of small discriminative parts (beak, wing markings, tail shape). Expected Grad-CAM achieves Attr.Loc. 0.563, a +32.8% improvement over S. Grad-CAM++ (0.424), alongside the best Sparseness (0.740) and Eff.Comp. (27.4k). This substantial gain directly validates the infidelity-minimization framework: by minimizing the discrepancy between the explanation and the model's behavior under perturbation, our method more accurately identifies the small, discriminative regions that determine bird species classification.

**Stanford Dogs** (Table 22). Expected Grad-CAM achieves the best Attr.Loc. (0.864) and RMA (0.871), with Sparseness 0.719 and Eff.Comp. 29.1k. The strong localization performance reflects that dog breed

Table 21: Quantus metrics on CUB-200-2011 Wah et al. (2025), averaged across 8 architectures ($N$=500 per combination).

| Method | Attr.Loc.↑ | RRA↑ | RMA↑ | AUC↑ | Sparse.↑ | Complex.↓ | Eff.Comp.↓ |
|---|---|---|---|---|---|---|---|
| Grad-CAM | 0.405 | 0.568 | 0.405 | 0.784 | 0.494 | 10.40 | 51.2k |
| Grad-CAM++ | 0.423 | 0.598 | 0.423 | 0.818 | 0.501 | 10.37 | 49.7k |
| S. Grad-CAM++ | 0.424 | **0.600** | 0.424 | **0.819** | 0.500 | 10.37 | 50.5k |
| Int. Grad-CAM | 0.413 | 0.580 | 0.413 | 0.799 | 0.496 | 10.40 | 51.9k |
| HiResCAM | 0.402 | 0.552 | 0.402 | 0.775 | 0.509 | 10.37 | 51.2k |
| XGrad-CAM | 0.395 | 0.540 | 0.395 | 0.760 | 0.507 | 10.37 | 50.5k |
| LayerCAM | 0.418 | 0.588 | 0.418 | 0.809 | 0.501 | 10.38 | 50.4k |
| Score-CAM | 0.411 | 0.534 | 0.411 | 0.768 | 0.533 | 10.28 | 49.7k |
| Ablation-CAM | 0.377 | 0.535 | 0.382 | 0.772 | 0.518 | 10.23 | 47.9k |
| **Expected Grad-CAM** | **0.563** | 0.595 | **0.563** | 0.787 | **0.740** | **9.61** | **27.4k** |

discrimination relies on head shape and coat patterns, *i.e.*, spatially concentrated features that our method identifies well.

Table 22: Quantus metrics on Stanford Dogs Khosla et al. (2012), averaged across 8 architectures ($N$=500 per combination).

| Method | Attr.Loc.↑ | RRA↑ | RMA↑ | AUC↑ | Sparse.↑ | Complex.↓ | Eff.Comp.↓ |
|---|---|---|---|---|---|---|---|
| Grad-CAM | 0.841 | 0.832 | 0.841 | 0.757 | 0.473 | 10.45 | 51.9k |
| Grad-CAM++ | 0.824 | 0.832 | 0.840 | 0.754 | 0.465 | 10.39 | 49.4k |
| S. Grad-CAM++ | 0.838 | 0.834 | 0.841 | 0.760 | 0.479 | 10.38 | 50.1k |
| Int. Grad-CAM | 0.845 | **0.837** | 0.845 | **0.767** | 0.468 | 10.47 | 52.5k |
| HiResCAM | 0.838 | 0.828 | 0.838 | 0.740 | 0.485 | 10.42 | 51.9k |
| XGrad-CAM | 0.838 | 0.827 | 0.838 | 0.737 | 0.482 | 10.43 | 51.3k |
| LayerCAM | 0.840 | 0.835 | 0.840 | 0.759 | 0.473 | 10.45 | 51.4k |
| Score-CAM | 0.837 | 0.822 | 0.837 | 0.730 | 0.511 | 10.35 | 49.7k |
| Ablation-CAM | 0.823 | 0.824 | 0.826 | 0.744 | 0.493 | 10.38 | 50.7k |
| **Expected Grad-CAM** | **0.864** | 0.814 | **0.871** | 0.703 | **0.719** | **9.67** | **29.1k** |

**Oxford-IIIT Pet** (Table 23). Expected Grad-CAM leads on Attr.Loc. (0.848), RRA (0.788), and RMA (0.848), alongside dominant Sparseness (0.691) and Eff.Comp. (32.3k). Notably, this is the only dataset where Expected Grad-CAM achieves the best RRA, suggesting that breed-discriminative features in this dataset are well-aligned with both faithfulness and rank-ordering criteria.

Table 23: Quantus metrics on Oxford-IIIT Pet Parkhi et al. (2012), averaged across 8 architectures ($N$=500 per combination).

| Method | Attr.Loc.↑ | RRA↑ | RMA↑ | AUC↑ | Sparse.↑ | Complex.↓ | Eff.Comp.↓ |
|---|---|---|---|---|---|---|---|
| Grad-CAM | 0.783 | 0.779 | 0.783 | 0.774 | 0.472 | 10.45 | 51.7k |
| Grad-CAM++ | 0.784 | 0.785 | 0.786 | 0.781 | 0.462 | 10.43 | 50.2k |
| S. Grad-CAM++ | 0.788 | 0.784 | 0.788 | 0.782 | 0.465 | 10.42 | 50.8k |
| Int. Grad-CAM | 0.788 | 0.787 | 0.788 | **0.788** | 0.466 | 10.47 | 52.5k |
| HiResCAM | 0.780 | 0.772 | 0.780 | 0.758 | 0.483 | 10.43 | 51.8k |
| XGrad-CAM | 0.774 | 0.763 | 0.774 | 0.746 | 0.483 | 10.43 | 51.1k |
| LayerCAM | 0.785 | 0.786 | 0.785 | 0.781 | 0.461 | 10.47 | 51.7k |
| Score-CAM | 0.764 | 0.755 | 0.764 | 0.730 | 0.508 | 10.37 | 50.2k |
| Ablation-CAM | 0.761 | 0.775 | 0.761 | 0.760 | 0.473 | 10.44 | 51.7k |
| **Expected Grad-CAM** | **0.848** | **0.788** | **0.848** | 0.751 | **0.691** | **9.82** | **32.3k** |

**Flowers-102** (Table 24). Flower classification primarily relies on color and petal shape, producing more diffuse discriminative regions than the other datasets. Expected Grad-CAM still leads on Attr.Loc. (0.473), RMA (0.476), Sparseness (0.645), and Eff.Comp. (36.6k), but the margins are smaller. S. Grad-CAM++ leads on RRA (0.503) and AUC (0.682).

Table 24: Quantus metrics on Flowers-102 Nilsback & Zisserman (2008), averaged across 8 architectures ($N$=500 per combination).

| Method | Attr.Loc.↑ | RRA↑ | RMA↑ | AUC↑ | Sparse.↑ | Complex.↓ | Eff.Comp.↓ |
|---|---|---|---|---|---|---|---|
| Grad-CAM | 0.420 | 0.476 | 0.420 | 0.641 | 0.383 | 10.59 | 52.5k |
| Grad-CAM++ | 0.430 | 0.494 | 0.430 | 0.670 | 0.389 | 10.57 | 52.2k |
| S. Grad-CAM++ | 0.433 | **0.503** | 0.433 | **0.682** | 0.385 | 10.59 | 53.4k |
| Int. Grad-CAM | 0.423 | 0.480 | 0.423 | 0.648 | 0.370 | 10.63 | 53.5k |
| HiResCAM | 0.413 | 0.470 | 0.413 | 0.629 | 0.402 | 10.56 | 52.5k |
| XGrad-CAM | 0.411 | 0.459 | 0.411 | 0.615 | 0.402 | 10.56 | 52.0k |
| LayerCAM | 0.425 | 0.487 | 0.425 | 0.661 | 0.389 | 10.59 | 53.0k |
| Score-CAM | 0.406 | 0.451 | 0.406 | 0.606 | 0.437 | 10.50 | 50.5k |
| Ablation-CAM | 0.426 | 0.490 | 0.431 | 0.666 | 0.383 | 10.54 | 51.3k |
| **Expected Grad-CAM** | **0.473** | 0.473 | **0.476** | 0.640 | **0.645** | **9.96** | **36.6k** |

**FGVC Aircraft** (Table 25). Aircraft discrimination depends on structural features (tail shape, engine placement, wing configuration) that span larger spatial extents. S. Grad-CAM++ achieves the best Attr.Loc. (0.795) and AUC (0.846), while Expected Grad-CAM leads on RMA (0.828), Sparseness (0.667), and Eff.Comp. (26.5k). The broader spatial extent of discriminative features in aircraft images slightly favors methods that produce wider activations.

Table 25: Quantus metrics on FGVC Aircraft Maji et al. (2013), averaged across 8 architectures ($N$=500 per combination).

| Method | Attr.Loc.↑ | RRA↑ | RMA↑ | AUC↑ | Sparse.↑ | Complex.↓ | Eff.Comp.↓ |
|---|---|---|---|---|---|---|---|
| Grad-CAM | 0.783 | 0.788 | 0.783 | 0.819 | 0.465 | 10.47 | 51.4k |
| Grad-CAM++ | 0.793 | 0.805 | 0.793 | 0.840 | 0.466 | 10.47 | 50.7k |
| S. Grad-CAM++ | **0.795** | **0.808** | 0.795 | **0.846** | 0.464 | 10.47 | 51.3k |
| Int. Grad-CAM | 0.789 | 0.795 | 0.789 | 0.829 | 0.466 | 10.48 | 51.9k |
| HiResCAM | 0.777 | 0.776 | 0.777 | 0.802 | 0.477 | 10.45 | 51.4k |
| XGrad-CAM | 0.774 | 0.768 | 0.774 | 0.793 | 0.476 | 10.45 | 50.8k |
| LayerCAM | 0.788 | 0.796 | 0.788 | 0.831 | 0.464 | 10.47 | 51.3k |
| Score-CAM | 0.795 | 0.780 | 0.795 | 0.807 | 0.539 | 10.30 | 48.4k |
| Ablation-CAM | 0.700 | 0.781 | 0.762 | 0.800 | 0.443 | 10.17 | 45.7k |
| **Expected Grad-CAM** | 0.752 | 0.759 | **0.828** | 0.707 | **0.667** | **9.65** | **26.5k** |

## F  Applicability to Vision-Language Models

To demonstrate that our framework extends beyond supervised classification, we apply Expected Grad-CAM to CLIP Radford et al. (2021), a contrastive vision-language model that pairs a visual encoder with a text encoder to perform zero-shot recognition via natural language prompts. This setting is particularly compelling: by conditioning explanations on free-form text rather than a fixed label set, one can interrogate the same image under different semantic queries without retraining.

We use the CLIP ResNet-50 visual backbone and generate saliency maps for diverse text prompts on ImageNet validation images. Figure 17 presents a qualitative comparison between Grad-CAM and Expected Grad-CAM across four scenes. Each panel shows one input image with two semantically distinct prompts, producing two saliency maps per method.

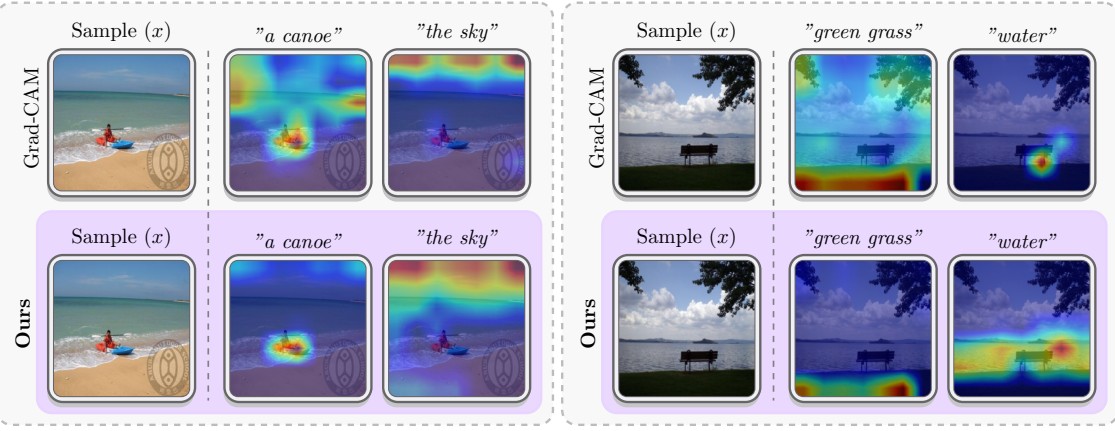

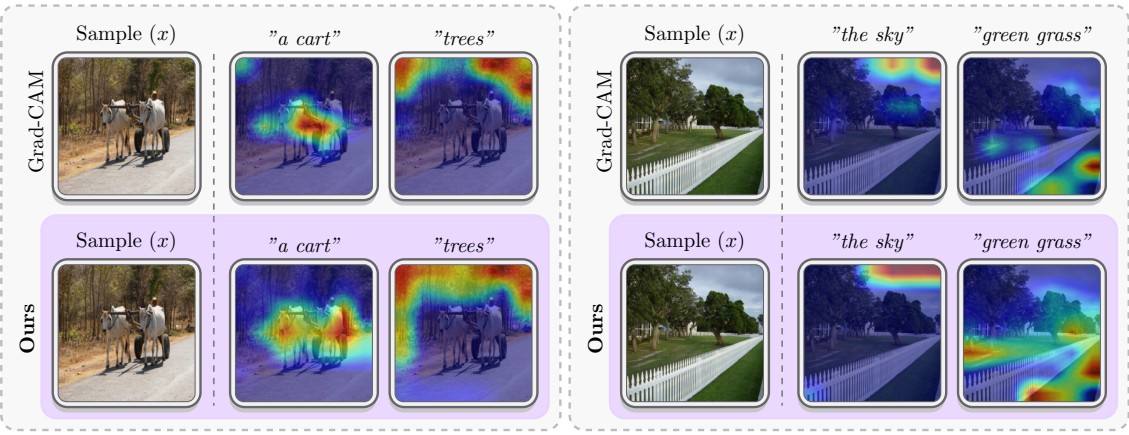

Figure 17: Prompt-conditioned explanations on CLIP ResNet-50. Each panel presents a single image with two text prompts. **Top rows**: Grad-CAM; **bottom rows**: Expected Grad-CAM (ours). Grad-CAM produces diffuse, largely prompt-invariant heatmaps; for instance, in the canoe scene both "a canoe" and "the sky" activate overlapping central regions. In contrast, Expected Grad-CAM yields spatially distinct saliency maps that shift with the prompt: the canoe is tightly localized in the foreground while the sky activation spreads across the upper region. Similarly, for the bench scene, our method correctly separates "green grass" (lower-left) from "water" (right horizon), whereas Grad-CAM conflates both. The same pattern holds across all panels: "a cart" highlights the ox-drawn cart while "trees" shifts activation toward the canopy, and "the sky" vs. "green grass" are correctly separated in the fence scene.

