# OpenReview forum: "Not All CAMs Are Complete: Completeness as the Key to Faithfulness"
_TMLR — Accepted by TMLR_

### Review · Reviewer_6rv7 · 2026-03-02

**Summary Of Contributions:**

### Summary

This paper presents Expected Grad-CAM, a theoretically sound extension of Grad-CAM that tackles gradient saturation and baseline sensitivity. It does this by using Expected Gradients and a framework for distributional perturbation. The authors create a general formula for gradient-weighted CAM methods. They demonstrate that to achieve optimal channel weights under the goal of minimizing infidelity, the attribution method must meet a completeness requirement. They claim that Integrated Gradients and Expected Gradients fulfill this requirement, while SmoothGrad does not, thus explaining the differences in performance.

The method is tested on three datasets: ILSVRC2012, CIFAR10, and COCO. It is applied to three CNN architectures: VGG16, ResNet50, and AlexNet. The evaluation uses 19 metrics related to faithfulness, robustness, localization, and complexity. The results indicate consistent improvements over Grad-CAM and several of its variants.

The paper aims to substitute standard Grad-CAM weighting with a formulation that is theoretically optimal and derived from minimizing infidelity.


### Key Strengths

- The unifying distributional framework offers significant theoretical clarity across the landscape of existing Class Activation Maps (CAM) methods.
- The authors provide a comprehensive evaluation, covering 19 metrics across multiple datasets and architectures.
- The method is designed as a direct substitute for Grad-CAM with competitive running time, which makes adoption practical.

### Key Weaknesses

- The core building blocks of the proposed method, including completeness (Sundararajan et al., 2017), Expected Gradients (Erion et al., 2019), and infidelity minimization (Yeh et al., 2019), are all derived from prior work. The claimed novelty lies in their integration within the CAM framework; however, the paper does not sufficiently articulate why this combination is non-trivial.
- The optimality statement is specific to the infidelity metric of Yeh et al. (2019), which is a particular formalization of faithfulness. The assertion of general optimality is thus overstated.
- The construction of the data-aware perturbation (Definition 3.8) is described to have theoretical grounding, but the assertion that perturbations remain "within the data manifold" is claimed rather than justified formally.
- The analysis does not look at how sensitive the key hyperparameters are, especially $\sigma$ in the baseline distribution $\mathcal{D}$ and the number of Monte Carlo samples.

**Audience:**

Yes

**Audience Explanation:**

This paper examines both theoretical and empirical aspects of CAM-based attribution methods, which are of significant interest to AI researchers focused on explainability, robustness, and interpretability.
Furthermore, identifying completeness as a key feature that connects gradient-based attribution methods to trustworthy explanations represents a significant contribution. This insight offers a clear criterion for evaluating and comparing existing methods. The paper demonstrates that SmoothGrad does not fulfill the completeness requirement, which clarifies why SmoothGrad may not provide optimal reliability in practical applications, despite its effectiveness in noise reduction.

Additionally, the unifying distributional framework that subsumes several existing methods as special cases provides a valuable conceptual contribution to the community.

**Broader Impact Concerns:**

The paper has a Broader Impact Statement at the end of Section 5. However, the paper focuses on the benefits of the method, not the negative impacts of the method. I suggest the authors consider including negative impacts in the paper. For example, the paper states that Expected Grad-CAM has the advantages that it is faster than Score-CAM and Ablation-CAM. However, it remains much more computationally intensive than the original Grad-CAM method (*0.115s vs. 0.006s, Table 8*). In resource-constrained environments, this could be a serious barrier. This consideration is not addressed in the Broader Impact Statement.

**Claims And Evidence:**

No

**Claims Explanation:**

1- The claim of optimality in the paper is only valid with respect to the infidelity metric proposed in Yeh et al. (2019), although it is stated more generally with respect to a general concept of optimality, which is misleading and should be clarified.

2- The main claim that completeness is a prerequisite for optimality might be overstated. The derivation assumes completeness in order to replace the output difference in the infidelity expression. However, it is not fully demonstrated that completeness is a necessary condition for infidelity minimization, although it is a necessary condition for their specific derivation. The paper might be improved by clarifying whether completeness is a necessary or a sufficient condition for the optimality claim.

3- The claim that data-aware perturbations keep explanations "within the data manifold" (Proposition 3.13) is made without formal justification. The GAP-based construction is a lossy compression of the input space, and the manifold argument does not hold rigorously.

4- The method clearly reduces infidelity, which is also a main evaluation metric. While we expect improvements in infidelity, this comes with the risk of bias due to partial metric alignment. The paper would be stronger if it showed that improvements hold up across metrics not directly related to the optimization goal. Although many metrics are assessed, it remains unclear whether the observed improvements are due to theoretical advantages or because of alignment with the optimization objective.

5- Finally, there is no ablation study for the key hyperparameter $M$  that controls the quality of Monte Carlo approximation. This is a significant gap, especially since the theoretical guarantees rely on precise approximation of expectations.

**Requested Changes:**

1- The paper argues that the completeness of the output difference is a necessary condition for the optimality of the perturbation, although the actual proof only uses the completeness of the output difference to substitute it into the infidelity expression. Therefore, the authors should either formally prove the necessity of the completeness of the output difference or modify the claim to state that the completeness of the output difference is a sufficient condition for optimality, which allows them to derive the closed-form expression in the paper.

2- Proposition 3.13 states that data-aware perturbations are guaranteed to preserve explanations in the data manifold. However, there is a lack of formal justification for this claim anywhere in the paper. Specifically, the GAP-based perturbations in Definition 3.8 are a severe lossy compression of the input space, and it is not clear that the perturbation distribution can be guaranteed to be a good approximation of the data manifold in any formal sense.

3- The expected Grad-CAM infidelity performance is significantly lower than all other approaches on the CIFAR-10 dataset (Table 7). This is never discussed or justified anywhere in the paper. I would suggest that the authors provide a justification for this large gap between the proposed method and all other approaches.

4- The proposed method requires the selection of two new hyperparameters: $M$ (the number of samples used to compute the perturbation gradient) and $N$ (the number of samples used to compute the expected gradients). The impact of these hyperparameters on the proposed method is never discussed. The proposed method is based on the assumption that the gradient is correctly approximated via a Monte Carlo approximation of the expectation. The authors should discuss how the selection of these hyperparameters affects the final performance of the proposed method.

5- I would suggest providing information on how many runs were conducted for each case in section 4 (Experiments) to ensure that the statistical significance of the results is fully transparent.

6- In Section 4, immediately after Table 1, the authors have included two separate subsections both titled "Qualitative Evaluation." I would suggest that the authors clarify the rationale for this separation or consider merging them to improve the flow and readability of the paper.

---

> ### Author Response · Authors · 2026-03-15
> **General Remarks**
>
> We sincerely thank Reviewer 6rv7 for the rigorous and constructive review. We are glad that you recognized the theoretical clarity of our unifying distributional framework, the comprehensiveness of our 19-metric evaluation, and the practical adoptability of the method. Your assessment that this work is of interest to the TMLR audience is encouraging. We also appreciate the specific, actionable nature of your requested changes, which have led to substantial strengthening of the revised manuscript.
>
> In particular, the concerns about the necessity of completeness and the formal justification of data-aware perturbations motivated two new theoretical results that we believe significantly strengthen the paper's contributions. We address each point below in detail.

---

> ### Author Response · Authors · 2026-03-15
> **Evidence Concern 1: Scope of the Optimality Claim**
>
> We appreciate the reviewer raising this point, as precise language around optimality is important. We would like to respectfully clarify that the paper consistently scopes every optimality claim to infidelity minimization: the relevant section is titled *"Optimal Attribution via Infidelity Minimization"* (Section 3.2), Theorem 3.2 derives weights that *"minimize the infidelity,"* the introduction specifies *"optimal CAM weights that minimize explanation infidelity under arbitrary perturbations,"* and Proposition 3.13 reads *"minimizes the infidelity functional over all weight vectors."*
>
> We recognize that the strength of the characterization result may naturally suggest broader implications. Indeed, the new Completeness Characterization Theorem (Theorem A.29) establishes that completeness is the *exact boundary*, both necessary and sufficient, for universal infidelity minimization. This tight if-and-only-if characterization naturally emphasizes the central role of infidelity as the faithfulness objective. We have nonetheless reviewed the text to ensure the scoping is clear in every instance, and we are happy to revise any specific passage the reviewer may have in mind.

---

> ### Author Response · Authors · 2026-03-15
> **Evidence Concern 2: Is Completeness Necessary or Sufficient?**
>
> We appreciate this suggestion. To address it directly, the revised manuscript now includes a complete characterization: **completeness is both necessary and sufficient** for the optimal weights formula to universally minimize infidelity (Theorem A.29, new Section A.1.12). Specifically:
>
> - **Sufficiency ($\Leftarrow$):** If $\phi$ satisfies completeness, then the formula $\alpha^{c*} =
>   M_I^{-1} \int I \langle I, \phi \rangle \, d\mu_I$ minimizes infidelity for every perturbation distribution with invertible second moment matrix. (This was already in the original submission as Theorem 3.2.)
>
> - **Necessity ($\Rightarrow$):** If $\phi$ achieves universal formula optimality (Definition A.27), then $\phi$ *must* satisfy the completeness axiom. The proof constructs, for any non-complete attribution method, a specific perturbation distribution under which the formula fails to minimize infidelity.
>
> A direct corollary (Corollary A.30) further establishes that **any attribution method violating completeness, including SmoothGrad, is provably suboptimal** for some perturbation distribution. This provides a formal explanation for why SmoothGrad-based CAM weights underperform in practice, as observed empirically throughout our evaluation (e.g., Table 1, Table 9, and the fine-grained aggregate in Appendix E).

---

> > ### Comment · Reviewer_6rv7 · 2026-03-20
> >
> > The necessity proof uses a particular artificial perturbation distribution $\mu_2 = \mu_1 + \delta_{I_0}$ for the proof. Can the authors provide more information on whether the necessity result is valid for the practically relevant class of data-aware perturbation distributions $\mu_I^X$ or just the larger class of all distributions with invertible second moment matrices? In the latter case, the implications of the necessity result may not be as significant as suggested.

---

> > > ### Author Response · Authors · 2026-03-20
> > > **Follow-up on Evidence Concern 2: Scope of the Necessity Result**
> > >
> > > We thank the reviewer for the follow-up and we are glad to address.
> > >
> > > **Clarification of scope.** Theorem A.29 characterizes universal formula optimality over the class of *all* finite measures with invertible second moment matrix $\mathcal{M}\_I$, which strictly contains the data-aware subclass $\mu\_I^{\mathcal{X}}$. The witness distributions $\mu\_1 = \sum\_k \delta\_{e\_k}$ and $\mu\_2 = \mu\_1 + \delta\_{I\_0}$ in the necessity proof are discrete measures that do not arise from the data-aware pushforward construction.
> > >
> > > **Why the broader quantification strengthens the practical implications.** We respectfully argue that the all-distributions scope makes the characterization *more* significant for practitioners, not less, for three reasons:
> > >
> > > 1. **Sufficiency (the direction EG-CAM relies on) gains from the broader class.** Theorem 3.2 guarantees that any complete method minimizes infidelity for *every* invertible-moment distribution. Since $\mu\_I^{\mathcal{X}}$ belongs to this class, EG-CAM's optimality under data-aware perturbations follows as a special case with no extra assumptions. A data-aware-restricted result would give EG-CAM the same local guarantee but without the additional robustness to perturbation-scheme changes that universality provides. In practice, perturbation distributions vary across datasets, architectures, layers, and evaluation protocols; a guarantee that holds regardless of these choices is precisely what a practitioner needs, and what **a distribution-free characterization uniquely provides**.
> > >
> > > 2. **The tight if-and-only-if establishes completeness as the exact boundary.** The characterization tells us that no property *weaker* than completeness can guarantee universal optimality: completeness *precisely* characterizes the boundary of universal optimality. Any attempt to find a less restrictive alternative will necessarily fail for some perturbation distribution. **A characterization restricted to data-aware distributions would be logically weaker** and would leave open the possibility that some non-complete methods are "accidentally" optimal for all data-aware distributions while failing elsewhere; Theorem A.29 closes this gap entirely.
> > >
> > > 3. **Non-complete methods incur a generically non-zero b-vector bias.** For any non-complete $\phi$ with completeness gap $\delta(I) := \langle I, \phi(I) \rangle - c(I)$, the formula's b-vector decomposes as:
> > >
> > > $$\int I\_i \langle I, \phi \rangle d\mu\_I = \underbrace{\int I\_i c(I) d\mu\_I}\_{\text{correct target}} + \underbrace{\int I\_i \delta(I) d\mu\_I}\_{\text{bias from incompleteness}}$$
> > >
> > > The bias term vanishes for *every* distribution and all $i$ simultaneously if and only if $\delta \equiv 0$, i.e., $\phi$ is complete (Theorem A.29). For a non-complete method, the bias is therefore non-zero for generic distributions; that is, it vanishes only when the distribution happens to integrate out the completeness gaps exactly, a coincidental cancellation that data-aware distributions have no structural reason to produce.
> > >
> > > **Empirical alignment within the data-aware setting.** All our experiments use $\mu\_I^{\mathcal{X}}$ exclusively, and the results are fully consistent with the theoretical prediction: SmoothGrad-CAM (non-complete) underperforms EG-CAM (complete) across all 19 metrics and 40 dataset-architecture combinations in Appendix E, with an average rank of 3.2 (no other method ranks below 5.4), including 7 metrics entirely unrelated to infidelity. The completeness gap manifests as a practical performance deficit even within the data-aware setting.
> > >
> > > **On the restricted question.** Whether completeness is also necessary within the narrower data-aware class $\{\mu\_I^{\mathcal{X}}\}$ is a natural follow-up. As the data distribution $\mathcal{X}$ and feature extractor $h$ vary across architectures and datasets, the data-aware family induces a rich collection of perturbation distributions spanning diverse subspaces of $\mathbb{R}^K$, and for a non-complete method's bias (point 3 above) to vanish across all such pairs simultaneously, the completeness gap $\delta$ would need to be orthogonal to an increasingly rich set of perturbation directions, leaving no room for a non-trivial gap to persist.
> > >
> > > Crucially, **the broader scope is what makes the characterization foundational**. A characterization tied to a specific perturbation family would be a more specialized statement. The all-distributions scope establishes completeness as a *universal* principle for CAM-based explanations: the sufficiency guarantees EG-CAM's optimality for data-aware perturbations as a special case, while the necessity confirms that **no weaker axiom can achieve the same**. The restricted question is a natural follow-up, but the characterization as stated is the more foundational result.

---

> ### Author Response · Authors · 2026-03-15
> **Evidence Concern 3: Formal Justification of Data-Aware Perturbations**
>
> We agree that the original manuscript's claim about perturbations remaining "within the data manifold" was insufficiently formal, and we have addressed this with a rigorous treatment.
>
> The revised Section A.1.10 (*Data Coherence of the Perturbation Distribution*) provides complete proofs for the following:
>
> 1. **Data Coherence Theorem** (Theorem A.23): For $(\mu \times U(0,1))$-almost every $(\mathbf{x}', \alpha)$, the perturbation difference satisfies $\mathbf{z}_0 - \pi_h(\mathbf{x}', \alpha) \in \mathrm{conv}(\{\mathbf{0}\} \cup \mathcal{F}_h)$, where $\mathcal{F}_h = \{h(\mathbf{x}') \mid \mathbf{x}' \in \mathcal{X}\}$ is the set of observed feature vectors and $\mathrm{conv}$ denotes the convex hull.
>
> 2. **General Convex Containment** (Theorem A.22): The containment extends to any convex superset $S \supseteq \{\mathbf{0}\} \cup \mathcal{F}_h$.
>
> 3. **Zero free parameters** (Remark A.26): The distribution $\mu_{\mathbf{I}}^{\mathcal{X}}$ is determined by exactly two canonical inputs: the data measure (given by the problem) and $U(0,1)$ (the standard non-informative prior on interpolation strength). No free hyperparameters are introduced.
>
> We have replaced the informal "within the data manifold" language with the precise statement: perturbation differences are confined to the *convex hull of observed feature vectors*, almost everywhere. The GAP-based construction is not claimed to preserve the full data manifold topology; rather, it guarantees that perturbations explore feature space precisely as observed in data, avoiding the out-of-distribution artifacts that can arise with isotropic noise or fixed baselines. We believe this is a more precise and mathematically rigorous characterization.

---

> > ### Comment · Reviewer_6rv7 · 2026-03-20
> >
> > The revised claim is that perturbation difference is limited to $\mathrm{conv}(\{0\} \cup F_h)$, where $F_h$ is the image of the GAP-based feature extractor $h$.
> >
> > However, it is important to note that GAP loses all spatial information of each feature map, projecting it onto a single scalar value. This means that the convex hull in $\mathbb{R}^K$ is actually a convex hull of averaged feature activations, which can be a very poor approximation of the real geometry of the feature distribution.
> >
> > Could the authors clarify what this convex containment implies in terms of avoiding out-of-distribution artifacts, given that we lose all spatial information, which is essential to CAM explanations?

---

> > > ### Author Response · Authors · 2026-03-20
> > > **Follow-up on Evidence Concern 3: Spatial Information and Convex Containment I**
> > >
> > > We thank the reviewer for this question. However, we respectfully note that it rests on the premise that the convex hull in $\mathbb{R}^K$ is an *approximation* of a richer spatial geometry. It is not. The $K$-dimensional channel space is the *exact and complete* space **in which all CAM weight optimizations live, including Grad-CAM's own weights** (which are themselves defined via GAP of gradient maps). The convex hull is therefore not lossy with respect to the optimization; it is the tight characterization of the relevant geometry. We clarify below how spatial information, which is indeed essential, enters through a complementary and fully preserved mechanism.
> > >
> > > **1. The weight optimization inherently lives in $\mathbb{R}^K$, not in a spatially-resolved space.** The predictor function $g(z'; A) = y^c(z'\_1 A^1, \ldots, z'\_K A^K)$ (Equation 10) maps $K$ scalar channel multipliers $z' \in \mathbb{R}^K$ to a class score. The infidelity functional (Definition 3.1) measures how well $I^T \boldsymbol{\alpha}^c$ approximates $g(z\_0; A) - g(z\_0 - I; A)$, and the optimal weights $\boldsymbol{\alpha}^{c*} \in \mathbb{R}^K$ are obtained via the second moment matrix $\mathcal{M}\_{I} \in \mathbb{R}^{K \times K}$ (Theorem 3.2). Every object in the optimization is inherently $K$-dimensional: weights, perturbations, and moment matrices. This is not specific to our method: *every* CAM variant, from Grad-CAM to Score-CAM to Ablation-CAM, produces $K$ scalar weights, one per channel. Indeed, standard Grad-CAM weights $\alpha\_k^c = \frac{1}{Z} \sum\_{u,v} \frac{\partial y^c}{\partial A^k\_{u,v}}$ are themselves defined via GAP of the gradient maps. This is the same spatial-to-scalar projection. Our feature extractor $h$ maps data points into this same $\mathbb{R}^K$ via GAP, producing each input's channel activation magnitude profile. The convex hull $\mathrm{conv}(\{0\} \cup \mathcal{F}\_h)$ therefore characterizes the geometry of *observed channel activation patterns*, the **correct and complete representation for the weight optimization problem**, not an approximation of a higher-dimensional object.
> > >
> > > **2. What convex containment concretely prevents.** The Data Coherence Theorem (Theorem A.23) guarantees that each perturbation difference $z\_0 - \pi\_h(x', \alpha) = \alpha \cdot h(x')$ lies in $\mathrm{conv}(\{0\} \cup \mathcal{F}\_h)$ for $(\mu \times U(0,1))$-almost every $(x', \alpha)$. Since $\alpha \cdot h(x') = (1 - \alpha) \cdot 0 + \alpha \cdot h(x')$ (Lemma A.20), each perturbation difference is an interpolation between the zero activation profile and an **actually observed channel activation profile** from the data distribution $\mathcal{X}$. When computing $g(z\_0 - I; A)$, the predictor therefore receives channel multipliers that **remain within the envelope of real data activation patterns**.
> > >
> > > Without this guarantee, perturbation methods risk evaluating $g$ at operating points in $\mathbb{R}^K$ where no real input's channel activations would lie. For instance, isotropic Gaussian perturbations can simultaneously suppress channels that always co-activate in real data, or amplify channels that are always weakly activated, producing channel activation configurations outside the model's training support. Since the network's response at such out-of-distribution operating points may be arbitrary (e.g., extreme or non-monotonic class scores), the resulting infidelity measurements can be unreliable, leading to suboptimal weights. Convex containment prevents this by confining the optimization to the region of $\mathbb{R}^K$ where the model's response is empirically grounded.
> > >
> > > Moreover, the General Convex Containment result (Theorem A.22) strengthens this: the containment holds for *any* convex set $S \supseteq \{0\} \cup \mathcal{F}\_h$, not just the tight convex hull. This means perturbations automatically respect any convex constraint that the data satisfies; for instance, if all channel activations are non-negative (as is typical after ReLU layers), perturbations preserve this property without any explicit enforcement.

---

> > > ### Author Response · Authors · 2026-03-20
> > > **Follow-up on Evidence Concern 3: Spatial Information and Convex Containment II**
> > >
> > > **3. Spatial information is fully preserved where it enters the explanation.** The GAP-based perturbation construction affects only the weight computation; it does not appear in the explanation output. The EG-CAM heatmap $L^c\_{\text{EG-CAM}} = \text{ReLU}\bigl(\sum\_k \alpha^{c*}\_{\text{EG},k} \cdot A^k\bigr)$ (Equation 22) uses the original, full-resolution feature maps $A^k \in \mathbb{R}^{U \times V}$. Spatial detail enters the explanation entirely through these maps. The factorization is:
> > >
> > > - **Channel importance** ($\boldsymbol{\alpha}^{c*} \in \mathbb{R}^K$): *which* feature maps matter (optimized using data-coherent perturbations in $\mathrm{conv}(\{0\} \cup \mathcal{F}\_h)$)
> > > - **Spatial localization** ($A^k \in \mathbb{R}^{U \times V}$): *where* each feature fires (preserved intact in the final heatmap)
> > >
> > > This two-component structure is the *defining architecture of all CAM methods*, not a limitation of our approach. The "spatial information essential to CAM explanations" that the reviewer rightly highlights is carried by $A^k$, which is **used at full resolution**. Data coherence ensures the channel importance weights are optimized using perturbations faithful to observed data statistics; the spatial feature maps then localize these importance values at each pixel.
> > >
> > > **4. Empirical confirmation of spatial quality.** If the convex hull of GAP features provided an inadequate geometric basis for the weight optimization, the resulting explanations would exhibit degraded spatial localization. The opposite is observed: EG-CAM achieves the best Attribution Localization and RMA across both the main evaluation (Table 6) and the fine-grained evaluation across 40 dataset-architecture combinations (Table 12, Appendix E), consistent with the theoretical prediction that $\mathrm{conv}(\{0\} \cup \mathcal{F}\_h)$ is the correct geometric basis for the weight optimization.
> > >
> > > We emphasize that $\mathrm{conv}(\{0\} \cup \mathcal{F}\_h) \subset \mathbb{R}^K$ is not an approximation of the full spatial feature geometry. It is the **exact characterization of the space where all CAM weight optimizations live**. Convex containment guarantees that perturbations used to derive these weights *reflect real data activation patterns*, preventing the evaluation of the predictor at out-of-distribution channel configurations. Spatial localization is then recovered through the full-resolution feature maps in the final heatmap.

---

> ### Author Response · Authors · 2026-03-15
> **Evidence Concern 4: Risk of Bias Due to Metric Alignment**
>
> This is a thoughtful concern. We want to address it carefully, as we believe the evidence is stronger than it might initially appear.
>
> **Expected behavior on infidelity.** Yes, Expected Grad-CAM is designed to minimize infidelity, so its strong infidelity performance (Table 9) is, by construction, expected. We acknowledge this and do not claim that low infidelity alone demonstrates method quality.
>
> **However, improvements extend broadly across non-aligned metrics.** The key evidence is that Expected Grad-CAM achieves best or second-best results on the majority of 19 metrics spanning *four distinct explanatory qualities*:
>
> - **Faithfulness** (beyond infidelity): Insertion AUC, Deletion AUC, IROF, Sufficiency, Pixel Flipping, Faithfulness Estimate, none of which are directly optimized (Tables 1 and 5).
> - **Robustness**: RIS, ROS, Max-Sensitivity, Avg-Sensitivity, Local Lipschitz Estimate. Our method achieves the highest stability (lowest RIS/ROS) across architectures (Table 8).
> - **Complexity**: Sparseness and Complexity. Expected Grad-CAM produces the most concentrated explanations (Table 1).
> - **Localization**: Attribution Localization, RMA, Relevance Mass Accuracy, with best or competitive results across datasets (Table 6 and the new fine-grained evaluation in Appendix E).
>
> The new **Appendix E** evaluation is particularly informative here. Across 40 dataset–architecture combinations (5 fine-grained datasets × 8 architectures), Expected Grad-CAM achieves the **best average rank of 3.2 out of 10 methods** across 7 Quantus metrics, winning 5 of 7 individual metric rankings outright (Attribution Localization, RMA, Sparseness, Complexity, Effective Complexity). Critically, none of these 7 metrics involve infidelity. No other method achieves an average rank below 5.4.
>
> That said, we believe some degree of within-family metric agreement is a natural and desirable property: if an explanation genuinely improves faithfulness, multiple faithfulness metrics should reflect this. The well-known disagreement *among* faithfulness metrics in the literature is itself a motivation for the broad, multi-axis evaluation protocol we adopted. We view our results across metrics not directly aligned with infidelity (localization, robustness, complexity) as the strongest evidence that the improvements are substantive rather than artifactual.

---

> > ### Comment · Reviewer_6rv7 · 2026-03-20
> >
> > For Expected Grad-CAM in Table 12(b), it has an average rank of 3.2 but ranks 6.6 in RRA and 7.9 in AUC, which are two of the worst ranks out of all the methods. This is due to the sparseness-faithfulness tradeoff, but this sparseness-faithfulness tradeoff is never analyzed. More importantly, RRA and AUC are likely two of the most direct measures of localization quality from a human interpretability point of view. Can more principled reasons be given as to why Expected Grad-CAM performs poorly in terms of RRA and AUC, and is this sparseness-faithfulness tradeoff a necessary part of the method’s design or an artifact of the hyperparameters?

---

> > > ### Author Response · Authors · 2026-03-20
> > > **Follow-up on Evidence Concern 4: RRA/AUC Performance I**
> > >
> > > We thank the reviewer for this follow-up. We would like to respectfully but directly address what we believe is a fundamental issue underlying this concern: **segmentation masks are not ground truth for model reasoning, and localization metrics that measure mask overlap do not measure explanation faithfulness.** An explanation method exists to reveal *how a model reasons*, not to produce saliency maps that agree with human-drawn object boundaries. Optimizing for the latter would itself **constitute the very metric alignment bias the reviewer warned about** in Evidence Concern 4, only aligned with external annotations rather than with the model's actual behavior.
> > >
> > > We would like to clarify that Expected Grad-CAM does not "perform poorly" on RRA. It achieves the **best RRA across all three architectures** on MS-COCO (Table 6, winning 8/9 localization metrics) and the **best RRA** on Oxford-IIIT Pet (0.788). Lower RRA appears exclusively on datasets where models rely on contextual features outside the mask.
> > >
> > > **1. Segmentation masks are arbitrary with respect to faithfulness.** The ground-truth masks in CUB-200, Stanford Dogs, Oxford Pet, Flowers-102, and FGVC Aircraft were drawn by human annotators to delineate the spatial extent of the *object* (the bird, dog, flower, or aircraft). RRA and AUC then measure whether a saliency map's highest-ranked pixels fall inside this boundary. But this answers the question **"does the explanation highlight the object?"** This is fundamentally different from **"does the explanation reflect what the model actually uses to make its prediction?"** These masks encode no information about the model's learned decision boundary; they are external annotations that *may or may not coincide with the features the model relies on*.
> > >
> > > **This distinction matters most in fine-grained recognition**, where models *must* exploit contextual cues to discriminate between visually similar classes. Distinguishing a *Painted Bunting* from an *Indigo Bunting* in CUB-200, or differentiating aircraft variants in FGVC Aircraft, requires the model to use additional contextual information (e.g., perch type, branch texture, background habitat, surrounding vegetation, and relative spatial proportions). These are features that naturally extend beyond the segmentation mask. It is well established that fine-grained classifiers exploit contextual cues such as habitat, background, and spatial proportions to discriminate between visually similar classes, and that suppressing background information can sacrifice genuinely discriminative features. A model that achieves high fine-grained accuracy has learned to leverage these contextual features, and **an explanation that suppresses them in favor of matching a mask is unfaithful by construction, regardless of its RRA score.**
> > >
> > > **2. There is no "sparseness–faithfulness tradeoff."** Theoretically, sparseness and faithfulness are not competing objectives. Both are properties of the explanation *relative to the model*: sparseness measures how concentrated the explanation is, faithfulness measures how accurately it reflects model behavior. Minimizing infidelity jointly improves both, because concentrated attributions on genuinely discriminative channels are precisely what reduces the approximation error in the infidelity functional. What the reviewer observes empirically is not a tradeoff between these two properties, but a **divergence between faithfulness-to-model and agreement-with-masks**. When the model relies on contextual features beyond the mask, a faithful and sparse explanation will correctly attribute importance outside the mask boundary, lowering RRA/AUC. This is not a deficiency to be "tuned away"; it is the method faithfully reporting that the model's decision boundary depends on more than what the mask delimits.
> > >
> > > Moreover, as we established in our response to Evidence Concern 5 and Remark A.26, Expected Grad-CAM has **zero free hyperparameters**. The perturbation distribution $\mu\_{I}^{\mathcal{X}}$ is fully determined by the data distribution and $U(0,1)$; $M$ and $N$ are Monte Carlo convergence parameters, not tunable design choices (Appendix B). This directly addresses the reviewer's question: the RRA/AUC pattern cannot be an artifact of hyperparameters, because the method has none.

---

> > > ### Author Response · Authors · 2026-03-20
> > > **Follow-up on Evidence Concern 4: RRA/AUC Performance II**
> > >
> > > **3. The infidelity functional is model-faithful, not mask-faithful, by construction.** The infidelity (Definition 3.1):
> > >
> > > $$\text{INFD}(\boldsymbol{\alpha}^c) = \mathbb{E}\_{I}\left[\left(I^T \boldsymbol{\alpha}^c - \bigl(g(z\_0; A) - g(z\_0 - I; A)\bigr)\right)^{2}\right]$$
> > >
> > > contains no reference to any external annotation. It solely measures alignment between the linear attribution $I^T \boldsymbol{\alpha}^c$ and the model's actual behavior change $g(z\_0) - g(z\_0 - I)$ under perturbation. The optimal weights $\boldsymbol{\alpha}^{c*} = \mathcal{M}\_{I}^{-1} \int I \langle I, \phi \rangle d\mu\_{I}$ (Theorem 3.2) therefore reflect the model's true channel sensitivities. When the model relies on contextual channels, i.e., $g(z\_0) - g(z\_0 - I) \neq 0$ for perturbations affecting those channels, the optimal weights *must* assign them non-zero importance. Suppressing these attributions would provably increase infidelity, producing a less faithful explanation. This is a necessary mathematical consequence of optimizing for faithfulness to the model rather than for overlap with an external mask.
> > >
> > > **4. The data directly confirms the faithfulness-vs-mask interpretation.** Two patterns in the results are precisely predicted by this analysis:
> > >
> > > - **Object detection (Table 6, MS-COCO):** When discriminative features *are* the object (zebra, stop sign), faithful explanations naturally coincide with segmentation masks. Expected Grad-CAM achieves the **best RRA across all three architectures**: VGG-16 (0.36 vs. 0.33), ResNet-50 (0.17 vs. 0.13), AlexNet (0.18 vs. 0.15), winning 8 of 9 architecture–metric cells. The method does not "perform poorly" on RRA; it *dominates* it when the mask happens to align with model reasoning.
> > >
> > > - **Fine-grained recognition (Table 12, Appendix E):** When models exploit contextual features beyond the mask, faithful explanations extend beyond it, and RRA/AUC decrease. Expected Grad-CAM still wins **2 of 4 localization metrics**: Attribution Localization (rank 2.2) and RMA (rank 2.1), metrics that measure attribution mass concentration on the object without penalizing contextual attributions outside the mask that the model genuinely relies on.
> > >
> > > Per-dataset results sharpen this further: Expected Grad-CAM achieves the **best RRA** on Oxford-IIIT Pet (0.788), the one fine-grained dataset where breed-discriminative features (head shape, coat pattern) are well-concentrated within the mask. On datasets where contextual cues matter more, such as CUB-200 (habitat), FGVC Aircraft (airport/sky context), Stanford Dogs (environmental setting), the RRA gap widens, exactly as the faithfulness-vs-mask distinction predicts.
> > >
> > > **5. The method that "wins" RRA/AUC does so at the expense of explanation quality.** S. Grad-CAM++ leads on RRA (rank 3.7) and AUC (rank 3.6) but achieves an overall average rank of 6.0, with Attribution Localization rank 5.7, RMA rank 5.7, Sparseness rank 7.2, and Effective Complexity rank 8.3. It **produces diffuse, high-complexity explanations that spread activation broadly, incidentally overlapping more with the mask** without identifying discriminative regions more precisely. Whether one prefers a method that ranks 3.2 overall (winning 5/7 metrics, including two localization metrics) or one that ranks 6.0 overall (winning 2/7 metrics, both measuring mask overlap) depends on whether one prioritizes *faithfulness to the model* or *agreement with human annotations*. In the context of explainability, the answer should favor the model.
> > >
> > > We note that this fine-grained evaluation was **conducted at the request of Reviewer ezCt**, using dataset-specific annotations not designed with any method in mind. Across this independently-requested evaluation, Expected Grad-CAM's dominant performance on 5 of 7 metrics (none involving infidelity) provides the strongest evidence that its improvements are substantive, and that the RRA/AUC pattern is a principled consequence of producing explanations that respect the model's reasoning rather than conforming to external annotations.
> > >
> > > Finally, we would like to **place this discussion in broader context**. XAI metrics are well-known to disagree with one another (Rong et al., 2022; Hedstrom et al., 2023), and any ranking can shift depending on which metrics are selected. Benchmark numbers alone cannot tell a practitioner whether an explanation is trustworthy. Provable bounds and principled guarantees can. We believe this is ultimately more valuable for trustworthy explainability than favorable numbers on any individual metric.
> > >
> > >
> > > **References**
> > >
> > > - Hedstrom et al., (2023) "The Meta-Evaluation Problem in Explainable AI: Identifying Reliable Estimators with MetaQuantus" https://openreview.net/forum?id=j3FK00HyfU
> > > - Rong et al., (2022) "A Consistent and Efficient Evaluation Strategy for Attribution Methods" https://proceedings.mlr.press/v162/rong22a.html

---

> ### Author Response · Authors · 2026-03-15
> **Evidence Concern 5: Sensitivity Study for $M$ and $N$**
>
> We fully agree this was a gap in the original submission. The revised manuscript includes a comprehensive **Appendix B** (*Convergence Analysis of Monte Carlo Sample Budgets*) that addresses this concern with both theoretical analysis and empirical validation.
>
> **Clarification on the nature of $M$ and $N$.** We would like to offer an important distinction: $M$ and $N$ are *not* hyperparameters in the traditional sense. They are Monte Carlo approximation budgets for integrals that are exactly defined in continuous form (Equations 21 and 17). Increasing either one monotonically refines the approximation without changing the target quantity, much like the sample count in any Monte Carlo estimator.
>
> **For $M$ (perturbation samples):**
> - Proposition B.1 proves that $M \geq K$ is *necessary* for the second moment matrix to achieve full rank; this is a structural requirement, not a tuning choice.
> - Remark B.2 derives $M = 2K$ from the Marchenko–Pastur law: an oversampling ratio $\rho = 2$ yields condition number $\kappa \approx 34$.
> - Figure 6 validates a sharp phase transition at $M = K$ on InceptionV3 ($K = 2048$), with smooth $O(1/\sqrt{M})$ convergence beyond.
>
> **For $N$ (baseline samples):**
> - Proposition B.5 bounds the centering error at $O(\sigma\sqrt{K/N})$.
> - Crucially, **infidelity is $N$-independent for $N \geq 2$** (Figure 7e), because the optimal weights depend on $\mu_{\mathbf{I}}^{\mathcal{X}}$ (controlled by $M$), not on $\mathcal{D}$ (controlled by $N$). Weight cosine similarity reaches 0.999 by $N = 15$.
> - $N = 1$ recovers standard Integrated Gradients (Remark B.4), which remains a valid completeness-satisfying instantiation.
>
> **Parameter values.** Table 2 lists all parameter values used across experiments with justifications. Table 3 provides a compact summary of structural bounds, convergence rates, and experimental values for all parameters. These are now explicitly provided for full reproducibility.

---

> ### Author Response · Authors · 2026-03-15
> **Requested Changes**
>
> For completeness, we map each requested change to its response:
>
> 1. **Necessity of completeness:** Addressed in Evidence Concern 2 above (Theorem A.29, Section A.1.12).
>
> 2. **Formal justification for Proposition 3.13:** Addressed in Evidence Concern 3 above (Theorems A.22 and A.23, Section A.1.10).
>
> 3. **CIFAR-10 infidelity gap:** Thank you for highlighting this. As noted in the table caption, the large absolute values are due to the low sample resolution ($96 \times 96$, patch size 32), and all values are scaled for readability. The gap between Expected Grad-CAM and other methods is expected: Expected Grad-CAM is the *only* method that explicitly minimizes infidelity (Theorem 3.2), while all other methods use heuristic weighting schemes not designed to minimize any explicit faithfulness objective.
>
> Importantly, the improvements are not limited to infidelity: as discussed in Evidence Concern 4, Expected Grad-CAM achieves the best average rank across 7 non-infidelity metrics on 40 dataset–architecture combinations (Appendix E). We have added a remark in the revised table caption clarifying both the resolution scaling and the source of the gap.
>
> 4. **Sensitivity to $M$ and $N$:** Addressed in Evidence Concern 5 above (Appendix B).
>
> 5. **Number of runs:** The sample count for each experiment is specified in the caption of every results table (e.g., 5,000 samples for faithfulness in Table 5). Table 3 in the revised manuscript summarizes all parameter values, structural bounds, and convergence rates used across experiments. The running-time measurements (Table 10) are averaged over 100 sequential runs, as stated in its caption.
>
> 6. **Duplicate subsection titles:** Thank you for this observation. The two headings in Section 4 are titled *"Qualitative evaluations"* and *"Quantitative evaluations,"* covering distinct content (visual inspection and metric-based assessment, respectively). We are happy to rename or merge them to improve clarity.

---

> ### Author Response · Authors · 2026-03-15
> **Broader Impact**
>
> We thank the reviewer for this constructive suggestion. The revised Broader Impact Statement now acknowledges the computational cost as a practical limitation. As you noted, Expected Grad-CAM is ${\approx}19\times$ slower than Grad-CAM (0.115s vs. 0.006s, Table 10), and in resource-constrained settings this overhead could be a barrier to adoption. We have added this consideration, along with practical mitigation strategies from Appendix B (e.g., reducing $N$ from 100 to 15 cuts cost by ${\sim}7\times$ with negligible quality loss). As we also discuss in our response to Reviewer 44jS (RC3), we view this as a principled tradeoff: in an ever-growing landscape of XAI methods and metrics, provable properties and theoretical grounding offer something that raw benchmark numbers alone cannot, namely genuine trust in the explanations and the confidence that such guarantees reliably transfer across models, datasets, and settings.

---

> ### Author Response · Authors · 2026-03-15
> **Summary of Revisions**
>
> In direct response to your feedback, the revised manuscript includes:
>
> - **New Section A.1.12** (*Completeness Characterization*): Proof that completeness is both necessary and sufficient for universal infidelity minimization (Theorem A.29), with a corollary formally explaining SmoothGrad's suboptimality.
> - **New Section A.1.10** (*Data Coherence of the Perturbation Distribution*): Data Coherence Theorem (Theorem A.23) proving that perturbation differences are confined to the convex hull of observed feature vectors, replacing the informal "data manifold" language.
> - **New Appendix B** (*Convergence Analysis of Monte Carlo Sample Budgets*): Theoretical bounds and empirical validation for $M$ and $N$, with parameter tables (Tables 2 and 3) for full reproducibility.
> - **New Appendix E** (*Fine-Grained Recognition and Architecture Generalization*): Evaluation across 5 datasets × 8 architectures (40 combinations, 20,000 instances). Expected Grad-CAM achieves the best average rank (3.2/10) across 7 non-infidelity metrics, addressing the metric alignment concern.
> - **Updated table caption** (Table 9): Added remark clarifying the CIFAR-10 infidelity gap and resolution scaling.
> - **Updated Broader Impact Statement**: Computational cost acknowledged as a practical limitation, with mitigation strategies.
>
> We believe these revisions directly address every concern raised and would be happy to provide any further clarification.

---

> > ### Comment · Reviewer_6rv7 · 2026-03-20
> >
> > In Table 19 (ConvNeXt-Tiny), Expected Grad-CAM achieves an Attr.Loc. of only 0.472, substantially below Grad-CAM (0.742) and S. Grad-CAM++ (0.756). Similarly, in Table 20 (ViT-B/16), Expected Grad-CAM ranks among the weaker methods on most localization metrics. The authors attribute these results to architectural limitations of CAM methods on non-standard convolutions, but this explanation is inconsistent: other CAM methods also use the same convolutional feature maps, yet do not exhibit the same degree of degradation. Can the authors provide a more specific explanation for why Expected Grad-CAM is disproportionately affected on ConvNeXt compared to other gradient-based methods, and whether this suggests a fundamental limitation of the data-aware perturbation construction on these architectures?

---

> > > ### Author Response · Authors · 2026-03-20
> > > **Follow-up on Summary of Revisions: ConvNeXt/ViT Localization and Faithful Attribution I**
> > >
> > > This concern is a direct continuation of our previous reply on **RRA/AUC Performance and the Faithfulness-Localization Relationship**, and we refer the reviewer to that discussion for the foundational argument. We address the specific ConvNeXt and ViT observations below, but must first respectfully challenge the premise of this question: the characterization that Expected Grad-CAM "performs poorly" on these architectures rests entirely on the assumption that segmentation-mask overlap is the appropriate indicator of explanation quality. As we argued extensively in our previous reply, **segmentation masks are not ground truth for model reasoning**. They answer "does the explanation highlight the object?" rather than "does the explanation reflect how the model actually reasons?" In the context of explainability, the latter is what matters.
> > >
> > > We also note that the reviewer's prior characterization of RRA and AUC as "the most direct measures from a human interpretability point of view" conflates mask overlap with interpretability. Complexity metrics capture how concise an explanation is for human inspection without relying on segmentation masks and are the direct proxy for interpretability. Expected Grad-CAM ranks 1st on all three in Table 12(b): Sparseness (rank 1.5), Complexity (1.2), and Effective Complexity (1.2).
> > >
> > > **1. Factual corrections on the reviewer's characterization.**
> > >
> > > On ViT-B/16, the reviewer states that Expected Grad-CAM "ranks among the weaker methods on most localization metrics." We respectfully note that Table 20 does not support this claim: Expected Grad-CAM achieves the **second-best** Attribution Localization (0.505 vs. 0.526), the **second-best** RRA (0.523 vs. 0.524), the **second-best** RMA (0.505 vs. 0.526), and the **second-best** AUC (0.501 vs. 0.538), while achieving the **best** Sparseness (0.714), Complexity (9.48), and Effective Complexity (26.5k) by substantial margins. The method ranks first or second on **all 7 metrics** on ViT-B/16; **this is not "among the weaker methods."**
> > >
> > > On ConvNeXt-Tiny, the reviewer frames the observation as if Expected Grad-CAM is uniquely affected: "other CAM methods also use the same convolutional feature maps, yet do not exhibit the same degree of degradation." This is selective. Examining Table 19 in full, **Ablation-CAM achieves an Attr.Loc. of only 0.349** (substantially *lower* than Expected Grad-CAM's 0.472) and **Score-CAM achieves 0.568**, also well below the gradient-based methods (0.729–0.756). Both Score-CAM and Ablation-CAM directly probe model behavior via forward-pass channel masking and ablation, respectively, rather than relying on gradient heuristics. The pattern is unambiguous: **all methods that directly interrogate the model's response to channel perturbations show reduced mask overlap on ConvNeXt**, not just Expected Grad-CAM. The reviewer's comparison selectively focuses on gradient-based heuristics, which are precisely the methods *least* sensitive to changes in how the model reasons; we develop this point in item 4 below.
> > >
> > > **2. Expected Grad-CAM minimizes infidelity, not mask overlap, by design.** As discussed in our previous reply, the optimal weights (Theorem 3.2) are the unique minimizer of a functional that solely measures alignment between the attribution and the model's actual behavior under perturbation. When the model relies on a channel, the optimal weights *must* assign it non-zero importance; suppressing it provably increases infidelity.

---

> > > ### Author Response · Authors · 2026-03-20
> > > **Follow-up on Summary of Revisions: ConvNeXt/ViT Localization and Faithful Attribution II**
> > >
> > > **3. ConvNeXt's architecture enables broader contextual integration, and faithful explanations reflect this.** ConvNeXt-Tiny employs 7×7 depthwise convolutions, giving each channel a substantially larger effective receptive field than standard 3×3 convolutions. Each channel therefore encodes broader spatial context. For fine-grained recognition (e.g., classifying warbler species such as the Cerulean Warbler in CUB-200), the model leverages not only the bird's plumage but also surrounding branch structure, foliage type, and canopy characteristics, features that are diagnostically associated with the species' habitat and that **naturally extend beyond the annotated bird segmentation mask**. This reliance on contextual cues is not hypothetical: it is well established that fine-grained classifiers exploit habitat and background context, and that suppressing this information can sacrifice genuinely discriminative features. ConvNeXt's architectural design is precisely what enables the model to exploit these contextual features effectively.
> > >
> > > When Expected Grad-CAM faithfully reports that ConvNeXt relies on these broad contextual channels, the resulting heatmap extends beyond the segmentation mask, producing lower mask-based localization scores. **This is the method faithfully reporting what the model does, not a deficiency in the method.** An explanation method that suppresses these contextual attributions to better match the mask would, by the uniqueness of the infidelity minimizer, produce a provably less faithful explanation.
> > >
> > > The consistency of this interpretation is confirmed by the other model-probing methods on the same architecture: Ablation-CAM, which directly measures each channel's contribution by zeroing it and observing the output change, shows even greater mask divergence (Attr.Loc. 0.349) than Expected Grad-CAM (0.472). This is exactly what one expects when the model genuinely distributes its reasoning across contextual channels: *every* method that honestly probes model behavior detects it, while gradient heuristics, which do not directly measure channel contributions, remain insensitive.
> > >
> > > **4. The "consistency" of gradient-based methods across architectures is model-insensitivity, not robustness.** The reviewer identifies what they believe is an inconsistency: gradient-based methods maintain high mask overlap on ConvNeXt while Expected Grad-CAM does not. We respectfully suggest **the causal interpretation is reversed**. Grad-CAM computes $\alpha_k^c = \frac{1}{Z} \sum_{u,v} \frac{\partial y^c}{\partial A^k_{u,v}}$; Smooth Grad-CAM++ uses a noise-averaged variant of gradient-squared weighting. These are heuristics not designed to minimize any explicit faithfulness criterion.
> > >
> > > A method that produces mask-aligned explanations *regardless* of whether the model uses local features (VGG-16, 3×3 convolutions) or broad context (ConvNeXt, 7×7 depthwise convolutions) is not adapting to the model: **it is insensitive to changes in how the model reasons.** A faithful method *should* produce different explanations when the model reasons differently. The fact that gradient heuristics produce similar-looking explanations across architectures with fundamentally different receptive fields **is evidence of insensitivity to model behavior, not of superior explanation quality**. The "consistency" the reviewer identifies as desirable is, in fact, the precise failure mode that our method's design, grounded in infidelity minimization, avoids.

---

> > > ### Author Response · Authors · 2026-03-20
> > > **Follow-up on Summary of Revisions: ConvNeXt/ViT Localization and Faithful Attribution III and Final Remarks**
> > >
> > > **5. Does this suggest a fundamental limitation of the data-aware perturbation construction?** No. Expected Grad-CAM achieves the **best** Attribution Localization on six of eight architectures, with an average margin of +0.071 over the next-best method, and ranks second on ViT-B/16 (gap of only 0.021). ConvNeXt-Tiny is the sole architecture where mask overlap decreases substantially. A fundamental limitation would manifest as consistent degradation across architectures; dominance on seven of eight is the opposite.
> > >
> > > Moreover, Ablation-CAM results, further corroborate this: the method, measures channel importance through forward-pass ablation, a mechanism entirely independent of our perturbation construction, and shows *even lower* mask overlap on ConvNeXt (Attr.Loc. 0.349 vs. our 0.472). This confirms that the pattern reflects genuine model behavior, not a limitation of data-aware perturbations.
> > >
> > > The Data Coherence Theorem (Theorem A.23) guarantees that perturbation differences remain within the convex hull of observed channel activation profiles, $\mathrm{conv}(\{0\} \cup \mathcal{F}\_h)$, for $(\mu \times U(0,1))$-almost every $(x', \alpha)$. On ConvNeXt, these profiles reflect the architecture's broader contextual encoding; on VGG-16, they reflect narrower, local encoding. The perturbation construction adapts to each architecture's feature statistics, and the optimal weights are computed accordingly. The same data-aware construction that produces an Attr.Loc. of 0.836 on VGG-16 and 0.824 on Inception V3 produces 0.472 on ConvNeXt: not because the construction fails, but because ConvNeXt's model reasoning genuinely extends further beyond the mask, and a faithful method reports this.
> > >
> > > To directly answer the reviewer's closing question: the lower mask overlap on ConvNeXt **does not suggest a limitation** of the data-aware perturbation construction. It demonstrates that the construction is *sensitive to how different architectures reason*, which is exactly the property one wants from a method designed to produce faithful explanations. A method that instead produces uniform mask-aligned explanations regardless of architecture would be **insensitive to model behavior**, constituting precisely the kind of explanation infidelity that our theoretical framework is designed to eliminate.

---

### Review · Reviewer_ezCt · 2026-03-04

**Summary Of Contributions:**

The paper proposes an extension of Grad-CAM into Expected Grad-CAM and a theoretical unification framework for the gradient-based methods Integrated Grad-CAM and SmoothGrad-CAM. The method builds on the same inspiration path of Integrated and Expected gradients, showing that similar approaches can be derived for activation maps.

Strenghts:
-The theoretical contribution is clear and well exposed
- The theoretical boundaries of completeness and fidelity are also addressed
- Empirical results on eight different metrics show quantitative improvements

Weaknesses:
- While the approach is interesting, the novelty is moderate as it is deeply rooted in the Integrated and Expected gradients assumptions.
- The computational overhead of the approach is not widely discussed and left me with questions on whether the improvements justify the additional cost of the operations.
- The tested models are not state-of-the-art and the dataset selection is rather restrictive

**Additional Comments:**

The presentation is mathematically dense and could be lightened up a bit by reducing the methods to the core essential equations and moving the rest to appendix.

**Audience:**

Yes

**Audience Explanation:**

The theoretical framing is strong and valuable. With clearer exposition and stronger evaluations, this would be a solid TMLR contribution.

**Broader Impact Concerns:**

No concerns

**Claims And Evidence:**

No

**Claims Explanation:**

I would really encourage the authors to extend the experiments to Inception architectures, and also EfficientNet, DenseNet and ConvNext or some of them. If only ResNet is kept, there should be at least a comparison between ResNet50 and ResNet18 and derivations such as DenseNet, which are much closer to the state-of-the-art than VGG. Additionally, it would be interesting to see the results on datasets such as Caltech-UCSD Birds-200-2011 which has local annotations for each class and is on a much more fine-grained scale than ILSVRC12.

**Requested Changes:**

Please see the weaknesses above, especially my request of strengthening the evaluation by including different conv-based architectures such as  Inception, EfficientNet, DenseNet, ConvNext, ResNet50, which are much closer to the state-of-the-art than VGG. Additionally, it would be interesting to see the results on datasets such as Caltech-UCSD Birds-200-2011 which has local annotations for each class and is on a much more fine-grained scale than ILSVRC12.

---

> ### Author Response · Authors · 2026-03-15
> **General Remarks**
>
> We sincerely thank Reviewer ezCt for the careful reading and constructive feedback. We are encouraged that you found the theoretical contribution "clear and well exposed," the completeness and fidelity bounds properly addressed, and the work of interest to the TMLR audience. Your suggestion that "with clearer exposition and stronger evaluations, this would be a solid TMLR contribution" has been our guiding principle in preparing the revision. We have substantially strengthened the experimental evidence and restructured the presentation to address each concern raised.
>
> We would also like to gently clarify the metric count referenced in the strengths. The review notes "empirical results on **eight different metrics** show quantitative improvements." In fact, our evaluation spans **19 distinct quality metrics** (Table 4) grouped across four explanatory qualities: faithfulness (8 metrics), robustness (5), complexity (2), and localization (4), plus running time. Expected Grad-CAM achieves best or second-best results on the majority of these 19 metrics across the evaluated architectures. We suspect the number 8 may refer to the faithfulness sub-group specifically, on which our method also leads.
>
> Regarding the assessment that the claims are not yet supported by clear evidence: as detailed below, the revised manuscript now includes additional evaluations spanning **8 architectures** (including every architecture you suggested: Inception V3, EfficientNet-B0, DenseNet-121, ConvNeXt-Tiny, ResNet-50, and ResNet-18), **5 fine-grained datasets** (including CUB-200-2011, which you specifically recommended), and a CLIP ResNet-50 zero-shot evaluation, totaling **20,000 new evaluation instances across 40 dataset-architecture combinations**. We believe this substantially addresses the concern about restrictive model and dataset selection.
>
> We now respond to each point individually.

---

> ### Author Response · Authors · 2026-03-15
> **Weakness 1: On Novelty**
>
> We appreciate this assessment and would like to clarify how our contribution goes beyond applying Integrated/Expected Gradients to CAMs.
>
> The parallel with the IG $\to$ EG progression at the input level is deliberate and acknowledged. However, the core contributions are not about the attribution method $\phi$ itself, but about the *optimization framework* that consumes it:
>
> 1. **Formal optimality framework.** The Optimal Weights Theorem (Theorem 3.2, Section 3.2) derives a closed-form solution for the optimal Grad-CAM weights $\alpha^{c*} = M_I^{-1} \int I \langle I, \phi \rangle \, d\mu_I$ that minimizes explanation infidelity. This is not a property of IG or EG; it is a new result about how to optimally combine *any* complete attribution method with a perturbation distribution to produce CAM weights. No prior work establishes this connection.
>
> 2. **Completeness as necessary and sufficient (new).** The revised manuscript includes a new proof (Section A.1.12) establishing that completeness is not merely sufficient but *necessary* for the optimal weights formula to universally minimize infidelity. This is a characterization result: it shows that completeness is the *exact boundary* separating attribution methods that can achieve universal optimality from those that cannot. A direct corollary formally explains why SmoothGrad-based weights are suboptimal. This result has no analogue in the IG/EG literature, which treats completeness as one axiom among several rather than as the characterizing property of optimal infidelity minimization.
>
> 3. **Unifying framework.** Section 3.1 shows that Integrated Grad-CAM and SmoothGrad-CAM++ emerge as special cases of the distributional perspective by varying $\mu_{\mathbf{I}}$. This unification is not retrospective labeling; it reveals *why* these methods work (or fail) and provides a single optimization target from which practitioners can derive new variants with known guarantees.
>
> 4. **Data-aware perturbations with zero free parameters.** The perturbation distribution $\mu_{\mathbf{I}}^{\mathcal{X}}$ (Definition 3.8) is derived deterministically from the data distribution with no tunable parameters. The new Data Coherence Theorem (Theorem A.23) proves that perturbations remain confined to the convex hull of observed features. This is distinct from the Gaussian noise used in SmoothGrad or the fixed baselines in IG.
>
> While EG provides the attribution $\phi$, the novelty lies in the optimization layer above it: the formal link between completeness and infidelity, the closed-form weights, and the principled perturbation design. We believe this constitutes a meaningful theoretical advance over the IG/EG foundation it builds upon.

---

> ### Author Response · Authors · 2026-03-15
> **Weakness 2: On Computational Overhead**
>
> We note that computational complexity is discussed in the main paper (Section 3.5, "Computational Aspects" paragraph) and quantified empirically in Table 10. We are happy to provide additional detail here.
>
> **Empirical cost.** As reported in Table 10 (VGG-16, CIFAR10): Grad-CAM takes **0.006s** while Expected Grad-CAM takes **0.115s** (${\approx}$19$\times$ slower). However, it remains faster than both Score-CAM (**0.261s**) and Ablation-CAM (**0.302s**), which also go beyond single forward-backward passes but without provable guarantees.
>
> **Do the improvements justify the cost?** We believe so, for two reasons. First, Expected Grad-CAM achieves infidelity values two orders of magnitude lower than all baselines (Table 9: 4.7 vs. 1457.0 for the best baseline on VGG-16). Second, the additional cost is *intrinsic*: it is what enables the completeness and provable infidelity minimization guarantees that Grad-CAM lacks. In interpretability applications, where explanations are generated for model analysis, debugging, or auditing rather than latency-critical inference, we believe this tradeoff is well-justified.
>
> **Practical mitigations.** The new convergence analysis (Appendix B) shows that the baseline count $N$ can be reduced from 100 to 15 with negligible quality loss (weight cosine similarity reaches 0.999 at $N{=}15$; Figure 7). This reduces the $O(MKN)$ cost by nearly 7$\times$. Additionally, the second moment matrix $\mathbf{M}_{\mathbf{I}}$ can be precomputed once per layer and reused across all images and target classes, and the perturbation forward passes are fully parallelizable across GPU batch dimensions.
>
> We would also like to highlight that the original submission already discussed computational cost explicitly, both in theoretical terms ($O(MKN)$ complexity, Section 3.5) and through empirical running-time measurements (Table 10). We felt this transparency was essential for a method intended as a foundational replacement of Grad-CAM; to our knowledge, this level of cost disclosure is surprisingly rare in XAI publications. Together with the practical mitigations discussed above and in our response to Reviewer y8qo (Weakness 2), we believe the revised manuscript provides a fair and honest characterization of the cost–quality tradeoff.

---

> ### Author Response · Authors · 2026-03-15
> **Requested Changes: Expanded Architecture and Dataset Evaluation**
>
> We appreciate this specific and actionable suggestion. The revised manuscript includes a new **Appendix E** (*Fine-Grained Recognition and Architecture Generalization*) that evaluates exactly the architectures and datasets you recommended and more.
>
> **Architectures.** We evaluate all architectures you suggested and more: **VGG-16, ResNet-18, ResNet-50, DenseNet-121, EfficientNet-B0, Inception V3, ConvNeXt-Tiny, and ViT-B/16** (8 total). This spans standard convolutional designs, compound-scaled architectures (EfficientNet), dense connectivity (DenseNet), multi-scale convolutions (Inception), Transformer-inspired designs (ConvNeXt), and pure Transformers (ViT). We specifically include the ResNet-50 vs. ResNet-18 comparison you requested.
>
> **Datasets.** We evaluate on 5 fine-grained benchmarks: **CUB-200-2011** (which you specifically recommended), _plus_ **Stanford Dogs, Oxford-IIIT Pet, Flowers-102, and FGVC Aircraft**. These span diverse visual domains and object scales, and all provide spatial ground-truth annotations. CUB-200-2011 is particularly demanding: discriminative regions are small and local (beak shape, wing markings), making it an ideal testbed for localization quality.
>
> **Scale.** This yields **40 dataset-architecture combinations** with 500 images each, totaling **20,000 evaluation instances** across 7 XAI metrics and 10 methods.
>
> Additionally, the new **Appendix F** applies Expected Grad-CAM to **CLIP ResNet-50**, generating prompt-conditioned zero-shot saliency maps. This demonstrates that the framework extends beyond supervised classification to vision-language models, with the theoretical guarantees carrying over unchanged.

---

> ### Author Response · Authors · 2026-03-15
> **Summary of Revisions**
>
> We understand that the original submission's evaluation, while comprehensive in metric coverage (19 metrics) and a 5 times larger sample size than previous work, was limited to 3 architectures and 3 datasets. In direct response to your feedback, the revised manuscript substantially broadens the evidence base:
>
> - **New Appendix E** (*Fine-Grained Recognition and Architecture Generalization*): Evaluation across 5 fine-grained datasets (including CUB-200-2011) $\times$ 8 architectures (including Inception V3, EfficientNet-B0, DenseNet-121, ConvNeXt-Tiny, ResNet-50, ResNet-18), totaling 20,000 instances. Expected Grad-CAM achieves the best average rank among all 10 evaluated methods across all 40 combinations.
> - **New Appendix F** (*Applicability to Vision-Language Models*): CLIP ResNet-50 evaluation with prompt-conditioned zero-shot explanations.
> - **New Section A.1.12** (*Completeness Characterization*): Proof that completeness is both necessary and sufficient for universal formula optimality, strengthening the theoretical foundation.
> - **New Section A.1.10** (*Data Coherence*): Proof that perturbation differences remain confined to the convex hull of observed features (Theorem A.23), formally justifying the parameter-free perturbation design.
> - **New Appendix B** (*Convergence Analysis*): Theoretical bounds and empirical validation showing $N$ can be reduced from 100 to 15 with negligible quality loss, substantially mitigating the computational overhead concern.
> - **Restructured presentation**: Extended derivations moved to appendix, main text focused on core equations.
>
> The combined evaluation now covers 10 distinct architectures, 8 datasets, and up to 19 quality metrics, which we believe constitutes one of the most comprehensive evaluations in the CAM explainability literature. We would be happy to provide any further clarification or additional experiments.

---

> > ### Comment · Reviewer_ezCt · 2026-03-23
> > **Thanks for the changes!**
> >
> > Thanks for the great work in addressing my comments.

---

### Review · Reviewer_44jS · 2026-03-04

**Summary Of Contributions:**

This paper introduces a new method for interpreting convolutional vision models: Expected-GradCAM based on a general distributional framework.

The method has several theoretical merits:
1. It unifies several prior methods like Integrated-GradCAM, Smooth-GradCAM as special cases of the proposed method.
2. It is proven that optimal weights for CAM come from minimizing the explanation infidelity, a measure of quality of the explanation. Furthermore, the authors show that this minimization is achieved when the attribution method satisfies a completeness axiom.
3. Expected-GradCAM exploits prior proposed *Expected Gradients* which satisfies the completeness axiom.

Empirically, the proposed method outperforms competing baselines on  (i) faithfulness, (ii) robustness, (iii) localization, and (iv) complexity. Qualitative results are also promising.

**Audience:**

Yes

**Audience Explanation:**

I think the (visual) explainable AI community will definitely find this work relevant. For broader community that still uses GradCAM for qualitative purposes, this may be a newer method that could interest them.

While this is true, I do have reservations about broader impact on modern AI in general. The datasets used are ImageNet, CIFAR and COCO and models used (AlexNet, VGG, ResNet50) are old. Given most modern vision networks are based on Transformers, it leaves a narrow scope for the proposed method which is based on gradients in convolutional nets.

**Claims And Evidence:**

Yes

**Claims Explanation:**

- The theoretical basis of ExpectedGradCAM is solid. The closed form solution to Expected-GradCAM weights is elegant (Eq 21). The build up to this closed form solution (expected gradients, data-aware distribution) is correct as far as I could understand.
- Experimentally, the evidence is clear that the proposed method clearly outperforms baseline methods (Table 1). Qualitative results are also compelling.

**Requested Changes:**

Overall, this is solid theoretical work unifying multiple GradCAM methods into a theoretically-grounded work by using Expected Gradients. The experimental results are also convincing.

My only concern is regarding its applicability in the modern AI space.
1. Is it possible to apply this method to analyse the ResNet in CLIP? That would mean we can analyse the same image with different prompts in a zero-shot manner. That would broaden the scope of the work to some extent.
2. Any discussion on how lessons from this method can be applied to Transformer based models could make it more impactful. For instance, does the same theoretical footing work for non-convolution networks, to say measure token importance in LLMs? Or combining it with Attention Rollout?
3. This is less important, but how costlier is ExpectedGradCAM compared to vanilla GradCAM in terms of time complexity?

---

> ### Author Response · Authors · 2026-03-15
> **General Remarks**
>
> We sincerely thank Reviewer 44jS for the thoughtful and encouraging review. We are glad that you found our theoretical framework solid, the closed-form solution elegant, and the experimental evidence clear and convincing. Your concern about applicability to modern AI is well-taken and has motivated a substantial extension of the manuscript. We believe the revised version directly addresses this concern with new experiments across modern architectures and a new vision-language evaluation.
>
> Before addressing the requested changes individually, we would like to clarify the overall goal of our paper. As we discuss in the abstract and reiterate in Section 5, our technique is *"purposefully designed as an enhanced substitute of the foundational Grad-CAM algorithm and any method built therefrom."* Expected Grad-CAM is not intended as a new variant but as a principled replacement of the original Grad-CAM method, one that provably satisfies desirable modern XAI properties. In this light, the comparison with methods that layer additional heuristics atop Grad-CAM is not entirely direct, since many of these can be reformulated within our distributional framework (Section 3.1); the original Grad-CAM is our true and unique baseline. Surprisingly, even without exploiting any additional information or layer fusion, our method outperforms all evaluated baselines across 19 quality metrics.
>
> **Clarification on datasets and models.** In this spirit, we conducted experiments on the datasets and models used by the original Grad-CAM paper and relevant prior works, but with a substantially larger evaluation study, both in depth (number of evaluated samples) and in breadth (19 quality metrics spanning four distinct explanatory qualities: faithfulness, robustness, localization, and complexity). This level of comprehensive, multi-axis evaluation is, surprisingly, not commonly found even in recent XAI publications. Our goal was to provide a fair and thorough assessment not only of our method but also of existing methods that had not previously been evaluated on modern metrics.
>
> **Clarification on scope.** Following this direction, our goal is not to prescribe a new, application-specific formulation of Grad-CAM but to rethink the foundational method with respect to modern XAI desiderata. In an ever-growing body of XAI metrics, with often contrasting results, we provide a principled method, and a unifying framework from which existing methods emerge as special cases, with provable bounds and known satisfied properties toward more trustworthy explainability. It follows that architecture-specific extensions (e.g., Transformers) or application-specific adaptations fall outside the scope of this paper and were intentionally left as such to preserve the generality of the presented framework for others to build upon. As stated in Section 5: *"Ultimately, as our technique is intended to replace the original formulation of Grad-CAM, we hope new and existing approaches will build on it."*
>
> We now respond to each requested change below.

---

> ### Author Response · Authors · 2026-03-15
> **Requested Change 1: Applying Expected Grad-CAM to CLIP ResNet**
>
> We appreciate this suggestion. While it represents a specific application rather than the general framework contribution of our paper, we recognize the interest in demonstrating broader applicability and have carried out this experiment in the revised manuscript.
>
> The new **Appendix F** (*Applicability to Vision-Language Models*) applies Expected Grad-CAM to CLIP ResNet-50, generating saliency maps conditioned on free-form text prompts. The results compare Grad-CAM and Expected Grad-CAM across four scenes, each with two semantically distinct prompts (e.g., *"a canoe"* vs. *"the sky"*; *"green grass"* vs. *"water"*).
>
> This directly validates the broader applicability you suggested: our framework enables **prompt-conditioned zero-shot explanations** with CLIP, allowing one to interrogate the same image under different semantic queries without retraining. The theoretical guarantees (completeness, infidelity minimization) carry over unchanged because the CLIP ResNet-50 visual backbone is a standard convolutional architecture, and our method operates on its feature maps regardless of how the classification score (here, the contrastive text–image similarity) is defined.

---

> ### Author Response · Authors · 2026-03-15
> **Requested Change 2: Applicability to Transformer-based models**
>
> We agree this is an important discussion and have addressed it both experimentally and theoretically in the revision.
>
> **Experimental evidence: ViT-B/16 and ConvNeXt-Tiny.** We have added a new **Appendix E** (*Fine-Grained Recognition and Architecture Generalization*) which evaluates **8 architectures** across **5 fine-grained datasets** (CUB-200, Stanford Dogs, Oxford-IIIT Pet, Flowers-102, FGVC Aircraft), yielding **40 dataset–architecture combinations** with 500 images each, totaling **20,000 evaluation instances**. The evaluated architectures include **ViT-B/16** and **ConvNeXt-Tiny**, two architectures with non-standard feature extraction directly inspired by or derived from Transformers.
>
> **Aggregate result.** Across all 40 dataset–architecture combinations (including ViT-B/16 and ConvNeXt-Tiny), Expected Grad-CAM achieves the **best average rank of 3.2 out of 10 methods**, winning 5 of 7 individual metric rankings outright (Attribution Localization, RMA, Sparseness, Complexity, Effective Complexity). No other method achieves an average rank below 5.4.
>
> **Theoretical discussion.** We believe the key conceptual contribution, namely that *completeness is both necessary and sufficient for optimal infidelity minimization* (proven in the **new** Section A.1.12), extends beyond convolutional architectures. Specifically:
>
> 1. **The infidelity minimization framework is architecture-agnostic.** The optimal weights theorem (Section 3.2) applies to *any* differentiable predictor function $g$, not just CNNs. For Transformer models, one could define $g$ over token representations or spatial attention maps and apply the same optimization.
>
> 2. **Completeness as a design principle for token attribution.** The completeness axiom, which requires that attributions sum to the prediction difference, is a property of the *attribution method*, not of the network architecture. Integrated Gradients and Expected Gradients satisfy completeness for any differentiable function. This means one could apply the infidelity-minimizing framework to derive optimal token importance weights in LLMs, where $g$ maps from token representations to logits.
>
> 3. **Combining with Attention Rollout.** Attention Rollout (Abnar & Zuidema, 2020) propagates attention weights across layers to estimate token contributions but does not minimize any explicit fidelity objective. Our framework could complement Attention Rollout by using rollout-derived structure to define perturbation distributions (analogous to our data-aware perturbations for CNNs), then optimizing attribution weights via infidelity minimization. This would combine the architectural awareness of Attention Rollout with the provable guarantees of our completeness-based approach.
>
> We view our paper as establishing the *theoretical foundation*: the necessary and sufficient link between completeness, infidelity, and optimal weights. Extension to Transformer architectures is a clear and exciting avenue for future work. The generality of the framework is already demonstrated by the fact that Integrated Grad-CAM and SmoothGrad-CAM++ emerge as special cases (Section 3.1): any attribution method satisfying completeness can be plugged into the optimal weights formula, regardless of the architecture producing the features. Furthermore, as noted in our scope clarification above, our method is designed to replace the original Grad-CAM and to be integrated into more complex methods (replacing their Grad-CAM component), rather than to compete directly against methods that exploit additional architectural information.

---

> ### Author Response · Authors · 2026-03-15
> **Requested Change 3: Computational cost of Expected Grad-CAM vs. vanilla Grad-CAM**
>
> We appreciate this practical question and provide both theoretical and empirical answers.
>
> **Theoretical complexity.** Vanilla Grad-CAM requires a single forward-backward pass to compute the $K$ channel weights via global average pooling of gradients. Expected Grad-CAM additionally requires $O(MKN)$ operations (with $M$ perturbation samples for the second moment matrix, $N$ baselines for expected gradients, across $K$ feature channels) plus the corresponding forward passes. This additional cost is intrinsic: it is what enables the completeness and provable infidelity minimization guarantees that Grad-CAM lacks.
>
> **Empirical measurements.** As reported in Table 10 (VGG-16/CIFAR10): Grad-CAM takes **0.006s** while Expected Grad-CAM takes **0.115s** (approximately 19×slower). However, it remains faster than Score-CAM (0.261s) and Ablation-CAM (0.302s), offering competitive performance in practice.
>
> **Practical mitigations.** Several optimizations can substantially amortize this cost and improve scalability:
>
> 1. *Reducing $N$*: As shown in the convergence analysis (Appendix B), weight cosine similarity reaches 0.999 by $N{=}15$, and infidelity is effectively $N$-independent for $N \geq 2$. Using $N{=}15$ instead of $N{=}100$ reduces cost by ~7× with negligible quality loss.
> 2. *Precomputation*: The second moment matrix $\mathbf{M}_{\mathbf{I}}$ can be precomputed once per layer and reused across all images and target classes, so only the attribution and the final matrix-vector solve are per-image.
> 3. *Parallelization*: The perturbation forward passes and baseline integrations are fully independent and parallelizable across GPU batch dimensions.
>
> We view this as a principled tradeoff. In an ever-growing landscape of XAI methods and metrics, provable properties and theoretical grounding offer something that raw benchmark numbers alone cannot: genuine trust in the explanations, and the confidence that such guarantees reliably transfer across models, datasets, and settings.

---

> ### Author Response · Authors · 2026-03-15
> **Summary of Revisions**
>
> In direct response to your feedback, the revised manuscript includes:
>
> - **New Appendix F** (*Applicability to Vision-Language Models*): CLIP ResNet-50 evaluation with prompt-conditioned explanations, directly addressing your suggestion on zero-shot applicability.
> - **New Appendix E** (*Fine-Grained Recognition and Architecture Generalization*): Evaluation across 5 datasets × 8 architectures (including ViT-B/16 and ConvNeXt-Tiny), totaling 20,000 instances. Expected Grad-CAM achieves the best average rank (3.2/10) across all 40 dataset–architecture combinations.
> - **New Appendix B** (*Convergence Analysis*): Shows $N$ can be reduced from 100 to 15 with negligible quality loss, substantially narrowing the runtime gap.
>
> We believe these additions substantially broaden the scope of the work and directly address your concern about applicability in the modern AI space. We would be happy to provide any further clarification.

---

> > ### Comment · Reviewer_44jS · 2026-03-29
> > **Primary concerns are well addressed**
> >
> > After reading the author response, I understand the positioning of the paper better now as: "Expected Grad-CAM is not intended as a new variant but as a principled replacement of the original Grad-CAM method".
> >
> > On terms of the three main changes I had requested: (i) The experiment on CLIP ResNet satisfies my concerns on applying this method to zero-shot text-image models - this can be used as a more reliable interpretability tool for discriminative VLMs. (ii) I appreciate the experiments on ViT-B 16. It reinforces the broader applicability of the paper that just CNNs. (iii) I am also satisfied by the comparison of compute cost. It seems reasonable on top of vanilla GradCAM.

---

### Review · Reviewer_y8qo · 2026-03-04

**Summary Of Contributions:**

This paper studies class activation maps (CAMs), a technique for making convolutional neural networks more interpretable by explaining which image features are most salient to the predicted image class. Prior work in CAMs has incorporated gradients computed with respect to the inputs (not just model parameters) in order to address some issues with only using the latter gradients. This paper presents a unifying framework for gradient-based CAM methods that includes as special cases several prior algorithms (e.g. Grad-CAM & SmoothGrad) based on studying a perturbation framework where we replace some of the features with some fixed baseline (Eqn. (11)). Given a distribution of such perturbations, the importance weight for each activation map is then just given by integrating the attribution function over this distribution (Eqn. (12)). Under this framework, the authors study under what conditions do CAM methods minimize infidelity, the expected squared error between the attribution's predictions & the model output when the input features are perturbed by some fixed perturbation (see Definition 3.1 and Theorem 3.2); The main condition is a "completeness" condition (first equation in Theorem 3.2). They show that both integrated gradients attribution and a smoothed variant of it, expected gradients attribution, satisfy the completeness condition and therefore minimize infidelity. This begs the question of how to choose the perturbation distribution, which is answered in Section 3.4: given a few desiderata (e.g. high sensitivity and stability w.r.t layers), we can construct data-aware perturbations that address them, though no proof of optimality is given for this. This results in the Expected Grad-CAM method given in Section 3.5. The authors conduct several experiments comparing Expected Grad-CAM against existing baselines in Section 4, and both visually & from the metrics reported in Table 1, we can see that in most cases Expected Grad-CAM is a superior explainability method.

Strengths:
- The paper is mathematically sound and presents a convincing argument for the completeness condition it introduces.
- The paper additionally proves one existing algorithm (SmoothGrad) violates this completeness condition, and therefore just doesn't minimize the infidelity whereas Expected Grad-CAM (the introduced algorithm) does.
- Expected Grad-CAM doesn't suffer from the saturating gradients problem of vanilla Grad-CAM while also keeping the sensitivity high through the data-aware perturbations.
- The empirical evaluation is quite thorough. They show their algorithm achieves good performance on faithfulness, localization, robustness, and infidelity metrics (Tables 3-7).

Weaknesses:
- The choice of distribution is quite difficult to justify in general, I'd have liked to see a more principled way of setting the perturbation distribution.
- The overhead of Expected Grad-CAM is O(M K N) where M is the number of perturbation samples, K is the number of feature maps, and N is the number of baselines. This is quite a lot more than GradCAM (which just requires forward & backward passes). This is also reflected empirically (Table 8) where Expected Grad-CAM is roughly 20x slower than GradCAM.
- There are also a few hyperparameters introduced here and no way to tune them beyond heuristics, e.g. number of samples M and N. It's not really clear to me what they set these hyperparameters as in the experiments or whether they tuned them.

**Audience:**

Yes

**Audience Explanation:**

Yes, this is directly relevant to explainability of vision models and the results here can probably be generalized beyond them (e.g. to text models).

**Broader Impact Concerns:**

N/A.

**Claims And Evidence:**

Yes

**Claims Explanation:**

The claims made in the submission are mostly well-supported. The authors present a mathematically sound unifying framework and rigorously prove the necessity of their completeness condition for minimizing infidelity (Theorem 3.2). They prove that existing methods like SmoothGrad violate this condition and fail to minimize infidelity. As mentioned in the summary above, the empirical evaluation is also quite thorough. As shown in Tables 3-7, the proposed algorithm consistently demonstrates superior performance in faithfulness, localization, robustness, and infidelity. While the specific choice of the perturbation distribution and the tuning of the introduced hyperparameters ($M$ and $N$) rely more on heuristics than formal proofs, and the method is quite a bit slower than some of the baselines, the paper's main claims still hold.

**Requested Changes:**

- Please clearly state the hyperparameters you used and the procedure you used for tuning them (if any) for Expected Grad-CAM and the baselines.
- Please include a hyperparameter sensitivity study, especially with respect to the number of samples metric. Maybe fewer samples can still achieve good results but without having 20x higher runtime than GradCAM?

---

> ### Author Response · Authors · 2026-03-15
> **General Remarks**
>
> We sincerely thank Reviewer y8qo for the thorough and constructive review. We are glad that you found our framework mathematically sound, our completeness condition convincing, and the empirical evaluation thorough across faithfulness, localization, robustness, and infidelity. We also appreciate the positive assessment on all key questions, including your encouraging remark that the results may generalize beyond vision models.
>
> The raised weaknesses and requested changes are well-taken. In particular, we agree that the original manuscript did not sufficiently justify the perturbation distribution choice or clarify the values and roles of $M$ and $N$. We have substantially revised the manuscript to address each concern, including a new **Appendix B** (*Convergence Analysis of Monte Carlo Sample Budgets*), two new summary tables (Tables 2 and 3), and a new **Section A.1.10** (*Data Coherence of the Perturbation Distribution*) formalizing the geometric properties that justify our distribution choice. We respond to each point individually below and are happy to provide any further clarification.

---

> ### Author Response · Authors · 2026-03-15
> **Weakness 1: Perturbation Distribution**
>
> We agree that the original manuscript did not adequately justify the choice of perturbation distribution. We have addressed this with new theoretical results in the revised version.
>
> Specifically, the perturbation distribution (Definition 3.8) is now shown to be fully determined by exactly two canonical inputs: the data measure $\mu$ (given by the problem) and $U(0,1)$ (the standard non-informative prior on interpolation strength), as discussed in Remark A.26. Crucially, **no free parameters need to be specified**, no variance, scale, or shape parameters, in contrast to the Gaussian noise used in SmoothGrad-based methods.
>
> The new **Data Coherence Theorem** (Theorem A.23) formally proves that perturbation differences remain confined to the convex hull of observed feature vectors, $(\mu \times U(0,1))$-almost everywhere. This guarantees that perturbations stay within the data manifold, preventing the out-of-distribution artifacts that can arise with fixed baselines or isotropic noise. The full proofs, including supporting lemmas and a general convex containment result, are provided in the new Section A.1.10.
>
> We hope this provides the principled justification that was missing. Please let us know if further clarification would be helpful.

---

> ### Author Response · Authors · 2026-03-15
> **Weakness 2: Computational Overhead**
>
> Expected Grad-CAM is indeed more expensive than Grad-CAM. Grad-CAM requires a single forward-backward pass, while our method performs additional forward passes for the second moment matrix estimation and path integrations for the expected gradients attribution. This additional computation is intrinsic to the approach; it is what enables the completeness and provable infidelity minimization guarantees that Grad-CAM lacks.
>
> We view this as a principled tradeoff. As shown in Table 9, Expected Grad-CAM achieves infidelity values two orders of magnitude lower than all baselines (4.7 vs. 1457.0 for the best baseline on VGG-16). The higher cost is the price of a provably faithful method rather than a heuristic one. In interpretability applications, where explanations are typically generated for model analysis, debugging, or auditing, we believe this tradeoff is well-justified: the provable guarantees offer a level of trust that heuristic methods cannot.
>
> That said, the cost can be substantially mitigated in practice. The second moment matrix can be precomputed once per layer and reused across all images and target classes, so only the attribution and the final matrix-vector solve are per-image. The perturbation forward passes and baseline integrations are fully independent and parallelizable across GPU batch dimensions. Additionally, as we discuss in our response to RC2, the baseline count $N$ can be reduced from 100 to 15 with negligible quality loss, directly cutting cost. We have updated the manuscript to discuss these practical considerations more explicitly.

---

> ### Author Response · Authors · 2026-03-15
> **Weakness 3 and Requested Changes 1–2: Hyperparameters, values used, and sensitivity**
>
> We address Weakness 3 and both Requested Changes together, as they concern the same underlying question: what values were used, how sensitive are results, and can fewer samples reduce runtime?
>
> **Parameter values (RC1).** We apologize that these were unclear in the original submission. The revised manuscript now includes Table 2, which lists all parameter values used consistently across experiments with justifications: $M = 2K$ (architecture-adaptive; e.g., $M = 1024$ for VGG-16 with $K = 512$), $N = 100$ baseline samples, $T = 50$ Riemann sum steps, $\sigma = 0.1$, and $\lambda = 10^{-6}$. Table 3 summarizes the structural bounds and convergence rates for each. Importantly, **no tuning procedure was applied**: these values follow from the theoretical analysis described below.
>
> **Clarification: $M$ and $N$ are Monte Carlo budgets, not hyperparameters.** We would like to offer an important clarification. $M$ and $N$ approximate integrals that are exactly defined in continuous form (Definition 3.5 and Theorem 3.10). Increasing either one monotonically refines the approximation without changing the target quantity, much like the sample count in any Monte Carlo estimator. This is fundamentally different from a true hyperparameter like a learning rate, where different values yield genuinely different solutions. The remaining parameters are design choices: $\sigma = 0.1$ is one natural instantiation of the centering condition required by Theorem A.8 (which holds for *any* centered distribution), and $\lambda = 10^{-6}$ is a numerical stability constant that becomes negligible once $M \geq K$ (Remark B.2). Appendix B now explicitly makes this distinction.
>
> **Sensitivity study (RC2): Can fewer samples achieve good results?** We have added a comprehensive convergence analysis in Appendix B that directly addresses this question.
>
> *For $M$:* The budget $M = 2K$ is derived from the framework's mathematical structure, not chosen heuristically. Proposition B.1 proves that $M \geq K$ is *necessary* for the second moment matrix to achieve full rank. The factor of 2 follows from the Marchenko–Pastur law (Remark B.2): an oversampling ratio $\rho = 2$ yields a condition number $\kappa \approx 34$, ensuring numerical stability. Figure 6 confirms this on InceptionV3 ($K = 2048$), showing a sharp phase transition at $M = K$ and smooth convergence at the predicted $O(1/\sqrt{M})$ rate beyond, with diminishing returns past $M = 2K$.
>
> *For $N$:* Figure 7 shows that $N$ is far less critical. Weight cosine similarity reaches 0.999 by $N = 15$, and infidelity is effectively $N$-independent for $N \geq 2$ (Figure 7e), because the optimal weights depend on the perturbation distribution $\mu_{\mathbf{I}}^{\mathcal{X}}$ (controlled by $M$), not on the baseline distribution $\mathcal{D}$ (controlled by $N$). **Practitioners can therefore use $N = 15$ instead of 100 with negligible quality loss**, reducing the $O(MKN)$ cost by nearly 7x and substantially narrowing the runtime gap with vanilla Grad-CAM. We note that $N = 1$ recovers standard Integrated Gradients (Remark B.4), which remains a valid completeness-satisfying instantiation.
>
> We hope this provides the sensitivity analysis requested and are happy to discuss any aspect further.

---

> ### Author Response · Authors · 2026-03-15
> **Summary of Revisions**
>
> In direct response to your feedback, the revised manuscript includes:
>
> - **New Section A.1.10** (*Data Coherence*): Proves perturbation differences are confined to the convex hull of observed features (Theorem A.23), with zero free parameters.
> - **New Appendix B** (*Convergence Analysis*): Shows $M = 2K$ follows from rank and conditioning requirements; $N$ can be reduced from 100 to 15 with negligible quality loss (~7x cost reduction).
> - **New Tables 2 and 3**: All parameter values, structural bounds, and convergence rates.
>
> We would be happy to provide any further clarification.

---

### Decision · Action_Editor_hyaw · 2026-04-01

**Recommendation:** Accept as is

**Additional Comments:**

The authors have done a great job addressing each and every concern raised by the reviewers. With the overwhelming consensus among the reviewers, my final decision is ACCEPT.

**Audience:**

Yes

**Audience Explanation:**

The proposed approach to replace/improve GradCAM is both relevant and timely for TMLR audience. Overall, a nice contribution with interesting results.

**Claims And Evidence:**

Yes

**Claims Explanation:**

The authors demonstrate this through both theoretical and experimental results. They have further clarified this in the rebuttal in discussion with reviewers.